# YOU ONLY NEED 4 EXTRA TOKENS: SYNERGISTIC TEST-TIME ADAPTATION FOR LLMS

## ABSTRACT

Large language models (LLMs) are increasingly deployed in specialized domains such as finance, medicine, and agriculture, where they face significant distribution shifts from their training data. Domain-specific fine-tuning can mitigate this challenge, but it relies on high-quality labeled data, which is expensive and slow to collect in expertise-limited settings. We study label-free test-time adaptation for language models and present SYTTA, an inference-time framework that adapts on the fly without additional supervision. SYTTA jointly leverages input-side perplexity and output-side predictive entropy as complementary uncertainty signals under distribution shift, optimizing them within a unified self-supervised objective that balances their influence to stabilize generation and enhance domain awareness. Across diverse model architectures and domain-specific benchmarks, SYTTA delivers consistent gains. Notably, on agricultural question answering, SYTTA improves ROUGE-$L_{sum}$ by over 120% on QWEN-2.5-7B with only 4 extra tokens per query. These results show that effective test-time adaptation for language models is achievable without labeled examples, supporting deployment in label-scarce domains. The code will be made available upon acceptance.

## 1 INTRODUCTION

Large language models (LLMs) have strong capabilities in reasoning, code generation, and language understanding, and they are being deployed in specialized domains or scenarios (OpenAI, 2023; Team et al., 2023; Anthropic, 2024; Guo et al., 2025). Financial institutions use LLMs for market analysis, healthcare providers employ them for clinical decision support, and agricultural organizations leverage them for crop management advice (Wu et al., 2023; Singhal et al., 2023; Kuska et al., 2024). However, these models often underperform in domain-specific settings where the language patterns, terminology, and knowledge needs differ from pre-training data (Wu et al., 2023; Singhal et al., 2023; Gu et al., 2021; Bella et al., 2024; Hu et al., 2025).

The standard responses include supervised fine-tuning (SFT) and reinforcement learning from human feedback (RLHF), which are effective when high-quality supervision is available (Wei et al., 2022; Ouyang et al., 2022). In production, however, collecting and refreshing domain-accurate data is costly, and specialized knowledge evolves over time, making maintenance difficult. Retrieval-augmented generation (RAG) (Lewis et al., 2020; Mao et al., 2021) and few-shot prompting (An et al., 2023) mitigate the need for finetuning, but both rely on curated supervision in different forms: RAG requires maintained corpora, while prompting depends on carefully chosen examples. These methods alleviate but do not remove the reliance on explicit resources, motivating approaches that adapt without external supervision.

These constraints motivate a complementary direction: adapting models at inference time *without external supervision*. Humans learn a language once and later adapt to new accents or dialects after brief exposure, without new explicit instruction, because the core vocabulary and grammar are already in place (Clarke & Garrett, 2004; Norris et al., 2003). Analogously, LLMs possess broad base abilities from pre-training; they can still miss the intent of a question or fail to select the right knowledge, not because the knowledge is absent, but because query and answer distributions diverge from pre-training. For instance, as shown in Figure 1, a query in Scottish dialect ("messages and a piece") is misinterpreted by the model, even though the intended meaning is "groceries and a sandwich." A human who already speaks English, however, can usually adapt after brief exposure to

such dialectal variations and will eventually understand the phrase correctly. This mirrors the goal of test-time adaptation: adjusting to distribution shifts during inference without requiring new labeled supervision. For autoregressive LLMs, distribution shift yields measurable uncertainty patterns: domain-specific inputs trigger higher token-level perplexity, and decoding exhibits higher predictive entropy. Treating these quantities as self-supervised signals enables per-cohort adaptation under practical latency budgets. This converts deployment-time uncertainty into a training signal that narrows the train–deploy gap without labels.

Prior test-time adaptation for LLMs has typically optimized a single signal. Input-side objectives reduce perplexity to better match domain patterns (Hu et al., 2025), yet they do not directly control decoding behavior. Output-side entropy minimization sharpens predictions (Wang et al., 2021; Niu et al., 2022), but naive application to autoregressive generation can cause repetition and collapse (Holtzman et al., 2020). The challenge is to couple these signals so that the model becomes more confident and more domain-aware, while avoiding degeneration and unnecessary computation.

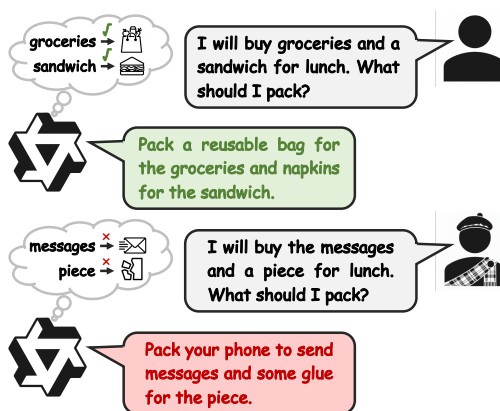

Figure 1: Illustration of LLM degradation under distribution shift: a Scottish dialect query ("messages and a piece") is misinterpreted as unrelated intent.

To this end, we propose Synergistic Test-time Adaptation (SYTTA), a unified framework that couples input perplexity and output predictive entropy for LLMs. SYTTA jointly reduces these uncertainties with guardrails that prevent degenerate text, and automatically allocates optimization effort to the dominant source of uncertainty per instance. The procedure is efficient: SYTTA adapts with only 4–16 extra tokens per query and supports two deployment modes. The *Dynamic-Ref* mode updates during generation for maximum effect, while the *Static-Ref* mode pre-computes signals before decoding to reduce latency. Both modes are practical for real deployments.

Our contributions are as follows:

1. We address the challenge of adapting LLMs to specialized domains under distribution shift and introduce Synergistic Test-time Adaptation (SYTTA), a framework that jointly leverages input perplexity and output entropy as self-supervised signals to adapt LLMs without labeled data.

2. We demonstrate consistent, statistically significant performance gains across domain, instruction, and reasoning benchmarks, while requiring only a small per-query token budget.

3. We conduct extensive empirical analysis to examine the effectiveness of different components and provide mechanistic insights. Through prefix semantic analysis and case studies, we show that SYTTA achieves genuine domain anchoring rather than superficial style mimicry.

## 2 RELATED WORKS

**Fine-tuning and retrieval from external knowledge.** Supervised fine-tuning and instruction tuning improve performance for downstream tasks when high-quality labels or preferences are available (Wei et al., 2022), and RLHF aligns models with human feedback (Ouyang et al., 2022). Retrieval-augmented methods combine parametric models with external corpora (Lewis et al., 2020; Guu et al., 2020), but they introduce extra modules and costs. These approaches assume labeled data (SFT/RLHF) or a curated, queryable corpus (RAG), and are thus not directly applicable in our test-time setup, where only questions are given without labels or domain knowledge.

**Label-free test-time adaptation.** Test-time adaptation updates models during inference without labels, aiming to mitigate performance degradation under distribution shift. In vision, entropy minimization (Tent) adapts classifier heads on unlabeled batches (Wang et al., 2021), with follow-ups improving stability and efficiency via sample selection (Niu et al., 2022), online adaptation (Bar et al., 2024), or conservative objectives (Zhang et al., 2025b). For LLMs, test-time training with in-context examples improves few-shot reasoning (Akyürek et al., 2025). Input-side updates with

perplexity objectives also yield strong gains without labels (Hu et al., 2025). These results highlight the utility of both input updates and output uncertainty control under shift.

**Reinforcement learning with verifiable or consistency signals.** RLVR uses programmatic checks as reliable rewards (Wen et al., 2025). GRPO replaces the critic with group-based scoring (Shao et al., 2024), while variants like DAPO (Yu et al., 2025), GFPO (Shrivastava et al., 2025), GSPO (Zheng et al., 2025), and GVPO (Zhang et al., 2025a) address stability, efficiency, or length control. Others explore test-time RL from consistency signals such as majority voting (Zuo et al., 2025), or simple entropy-based signals for math, code, and science tasks (Agarwal et al., 2025). These methods rely on self-consistency or external verifiers, which limits their use in domain-specific or instruction tasks without reliable checkers. Our method is inspired by their stable optimization goals, but works without verifiers at test time.

# 3 PROBLEM SETUP

## 3.1 APPLICATION SCENARIOS

We investigate test-time adaptation for question answering under the challenging "question-only" condition, where the model is exposed to a large set of unlabeled questions from a shifted target distribution. The inputs are processed in batches, denoted by $X = \{x_j\}_{j=1}^M$. To adapt, the language model may generate a short prefix for each input. Crucially, the token budget for this prefix must be minimal to ensure that the adaptation process does not introduce significant latency, which would diminish its practical utility in real-world applications.

**Cohort-Level Adaptation.** Our setting resembles a multi-tenant model-as-a-service deployment. Before answering a batch window of target-domain questions $X$, the model performs a single self-supervised adaptation pass on the corresponding unlabeled pool. After this pass, parameters are frozen, and answers are generated for that cohort. We evaluate on this same cohort, which is a transductive test-time adaptation protocol where the unlabeled evaluation inputs are exactly those used for adaptation, and no ground-truth answers are accessed. When switching to a different domain, the model resets to a base snapshot, preventing cross-cohort information leakage or unintended accumulation. This workflow keeps inference lightweight while maintaining reliability across cohorts.

## 3.2 NOTATIONS

Let $x = (x_1, \ldots, x_m)$ denote an input question, which is a sequence of $m$ tokens from a vocabulary $\mathcal{V}$. The corresponding response is a token sequence $y = (y_1, \ldots, y_n)$ of length $n$. We denote the base LLM as $p_\theta$, parameterized by weights $\theta$. The model calculates the probability of a response $y$ given an input $x$ through an autoregressive factorization:

$$p_\theta(y \mid x) = \prod_{t=1}^n p_\theta(y_t \mid y_{<t}, x). \tag{1}$$

During test-time adaptation, the model parameters are updated from $\theta$ to $\theta'$ based on the current input. Inference is then performed using the adapted model, $p_{\theta'}(\cdot \mid \cdot)$. For the adaptation step itself, the model generates a short prefix, denoted $\tilde{y}_{1:k}$, of length $k$. The value of $k$ also represents the extra token budget allocated for adaptation.

# 4 METHOD: SYTTA

Our method, SYTTA, realizes Synergistic Test-time Adaptation by coupling two complementary signals over a shared, short prefix context (Figure 2). *Input Distribution Adaptation* pulls the input side toward the target domain by lowering the question's perplexity; *Output Confidence Shaping* pushes the output side toward confident yet anchored next-token distributions. These two signals act on the same prefix, and we coordinate them with a *Dynamic Importance Weighting* rule that keeps their magnitudes comparable across instances. We elaborate on each of these components in the following sections. Additionally, we state the use of LLMs in Appendix A.14.

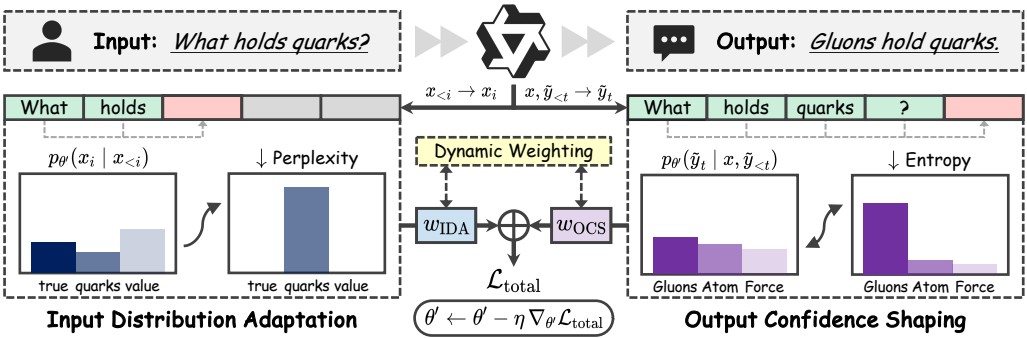

Figure 2: Overview of the SYTTA framework. *Input Distribution Adaptation* lowers input perplexity, *Output Confidence Shaping* reduces output entropy, and *Dynamic Importance Weighting* balances the two signals. We leverage uncertainties as self-supervised signals for test-time adaptation.

## 4.1 INPUT DISTRIBUTION ADAPTATION

To anchor the model in the target domain's specific language and concepts, we first optimize its ability to understand the incoming question $x$. Following recent test-time learning work (Hu et al., 2025), *Input Distribution Adaptation* minimizes prompt perplexity (equivalently NLL):

$$\mathcal{L}_{\text{IDA}}(\theta') = -\frac{1}{m} \sum_{i=1}^{m} \log p_{\theta'}(x_i \mid x_{<i}). \tag{2}$$

To focus adaptation on challenging instances, we employ a gating mechanism where the optimization is applied only to samples whose initial NLL under the base model $p_\theta$ exceeds a predefined threshold. For these selected samples, the loss is further amplified by a factor proportional to their NLL, promoting faster and more stable learning on difficult inputs.

## 4.2 OUTPUT CONFIDENCE SHAPING

While *Input Distribution Adaptation* reduces prompt perplexity, it does not directly control decoding-time uncertainty. Under distribution shift, the model can still produce high-entropy next-token distributions and drift early in generation. We hence introduce *Output Confidence Shaping*, which encourages lower entropy (i.e., a sharper next-token distribution) along a short length-$k$ prefix. Importantly, we add a reverse KL term that pulls back the adapted distribution to the frozen base-model reference, which stabilizes adaptation and mitigates degeneration.

For each input $x$, we form a short prefix of length $k$ and a reference distribution from the base model. Let

$$\tilde{y}(x) = \begin{cases} \text{GENPREFIX}(p_\theta, x, k), & \textit{Static-Ref} \text{ mode,} \\ \text{GENPREFIX}(p_{\theta'}, x, k), & \textit{Dynamic-Ref} \text{ mode,} \end{cases} \tag{3}$$

Given the generated prefix $\tilde{y}(x)$, we define the base-model reference logits at each step as

$$z_t^{\text{ref}}(x) = \log p_\theta(\cdot \mid x, \tilde{y}_{<t}(x)), \quad t = 1, \ldots, k. \tag{4}$$

In the *Static-Ref* mode, $\tilde{y}(x)$ and $\{z_t^{\text{ref}}(x)\}_{t=1}^{k}$ are computed once with the frozen base model $p_\theta$ and cached for the whole adaptation. In the *Dynamic-Ref* mode, the model updates while generating its own short prefix; the base-model reference logits $\{z_t^{\text{ref}}(x)\}$ are computed on the fly for the same context and are not cached.

The adapted model $p_{\theta'}$ conditions on $(x, \tilde{y}(x))$ with a base-model-forced forward pass to obtain learning signals. *Output Confidence Shaping* then minimizes token-level predictive entropy along the prefix and regularizes the adapted distribution toward the base model. The entropy term aggregates next-token entropies,

$$\mathcal{L}_{\text{ENT}}(\theta') = \sum_{t=1}^{k} H(p_{\theta'}(\cdot \mid x, \tilde{y}_{<t})), \tag{5}$$

---

**Algorithm 1** Training procedure of SYTTA-$k$ with optional prefix cache

---

**Require:** dataset $\mathcal{D}$, base model $p_\theta$, step size $\eta$, prefix length $k$, KL weight $\lambda_{\text{KL}}$, mode (*Static-Ref/Dynamic-Ref*)

**Ensure:** adapted parameters $\theta'$

1: **if** mode = *Static-Ref* **then**
2:     Cache $\mathcal{C} = \{x \mapsto (\tilde{y}(x), z_{1:k}^{\text{ref}}(x))\}$            `// store prefix and reference logits`
3: **end if**
4: $\theta' \leftarrow \theta$
5: **for** $s = 1$ **to** $S$ **do**
6:     Sample mini-batch $\mathbf{X} \subset \mathcal{D}$, $|\mathbf{X}| = B$
7:     Build tensors $\tilde{\mathbf{y}} \in \mathcal{Y}^B$, $Z_{1:k}^{\text{ref}} \in \mathbb{R}^{B \times k \times V}$:

> **Static-Ref**
> $$\tilde{\mathbf{y}} = (\tilde{y}(x))_{x \in \mathbf{X}},$$
> $$Z_{1:k}^{\text{ref}} = (z_{1:k}^{\text{ref}}(x))_{x \in \mathbf{X}}$$

> **Dynamic-Ref**
> $$\tilde{\mathbf{y}} = \text{GENPREFIX}(p_\theta, \mathbf{X}, k),$$
> $$Z_{1:k}^{\text{ref}} = \text{LOGITS}(p_\theta, \mathbf{X}, \tilde{\mathbf{y}}, k)$$

8:     Run $p_{\theta'}(\mathbf{X}, \tilde{\mathbf{y}})$         `// teacher-forced (Static-Ref) / generated prefix (Dynamic-Ref)`
9:     Compute losses: $\mathcal{L}_{\text{IDA}}, \mathcal{L}_{\text{OCS}}, \mathcal{L}_{\text{KL}} \in \mathbb{R}^B$         `// Sec. 4.1, Sec. 4.2`
10:    Compute weights: $w_{\text{IDA}}, w_{\text{OCS}} \in \mathbb{R}^B$         `// Sec. 4.3`
11:    Aggregate batch loss: $\mathcal{L}_{\text{batch}} = \frac{1}{B}\left(\langle w_{\text{IDA}}, \mathcal{L}_{\text{IDA}}\rangle + \langle w_{\text{OCS}}, \mathcal{L}_{\text{OCS}}\rangle + \lambda_{\text{KL}} \cdot \mathbf{1}^\top \mathcal{L}_{\text{KL}}\right)$
12:    Update: $\theta' \leftarrow \theta' - \eta \nabla_{\theta'} \mathcal{L}_{\text{batch}}$
13: **end for**
    **return** $\theta'$

---

where $H(\cdot)$ is the Shannon entropy. We do not commit to a particular instantiation of the entropy computation here, leaving flexibility for implementation choices.

Test-time adaptation is known to be highly sensitive and can easily suffer from over-updating, which leads to model collapse. To prevent drift and collapse, we add a per-token reverse KL term Gu et al. (2024) against the base-model reference,

$$\mathcal{L}_{\text{KL}}(\theta') = \sum_{t=1}^{k} D_{\text{KL}}\big(p_{\theta'}(\cdot \mid x, \tilde{y}_{<t}) \,\|\, \text{softmax}(z_t^{\text{ref}}(x))\big). \tag{6}$$

The *Output Confidence Shaping* objective combines these two parts,

$$\mathcal{L}_{\text{OCS}}(\theta') = \mathcal{L}_{\text{ENT}}(\theta') + \lambda_{\text{KL}} \mathcal{L}_{\text{KL}}(\theta'), \tag{7}$$

where $\lambda_{\text{KL}}$ balances confidence sharpening and proximity to the base model. The prefix length $k$ (typically 4–16 tokens) sets the strength of the output-side signal relative to computation. Detailed discussions of the entropy objective design and the choice of the KL formulation are provided in Appendix A.1.

### 4.3 DYNAMIC IMPORTANCE WEIGHTING

A static weighting between the *Input Distribution Adaptation* objective and the *Output Confidence Shaping* objective is suboptimal, because their relative difficulty varies across steps and instances. We therefore use a dynamic scheme that keeps the two contributions on a comparable scale, which helps stabilize training. The total loss is

$$\mathcal{L}_{\text{total}}(\theta') = w_{\text{IDA}}^{(t)} \mathcal{L}_{\text{IDA}}(\theta') + w_{\text{OCS}}^{(t)} \mathcal{L}_{\text{OCS}}(\theta'). \tag{8}$$

**Static baseline.** As a point of reference, the static baseline fixes $w_{\text{IDA}} = w_{\text{OCS}} = 1$.

**Dynamic Loss-Ratio Weighting.** To balance the two objectives, we propose a dynamic weighting scheme inspired by normalization-based methods in multi-task learning (Chen et al., 2018; Liu et al., 2019). The core idea is to adjust each objective's weight at every step based on its current contribution to the total loss, while enforcing stability.

First, we track the overall loss magnitude using an exponential moving average (EMA) with momentum $\beta \in [0, 1)$, which acts as a dynamic normalizer:

$$\mathcal{L}^{(t)} = \beta \mathcal{L}^{(t-1)} + (1 - \beta)\big(\mathcal{L}_{\text{IDA}}^{(t)} + \mathcal{L}_{\text{OCS}}^{(t)}\big). \tag{9}$$

Using this normalizer, we compute the relative contribution of each loss, $r_i^{(t)} = \mathcal{L}_i^{(t)}/(\mathcal{L}^{(t)} + \varepsilon)$ for $i \in \{\text{IDA}, \text{OCS}\}$, and normalize them to obtain preliminary weights $\tilde{w}_i^{(t)} = r_i^{(t)}/\sum_j r_j^{(t)}$. These are scaled by base coefficients $\lambda_i$, yielding $w_i^{(t)} = 2 \cdot \lambda_i \cdot \tilde{w}_i^{(t)}$.

However, we also observe that $L_{\text{OCS}}$ could always be orders of magnitude larger than $L_{\text{IDA}}$, where this ratio-based approach can cause training instability by effectively silencing one objective. To prevent this, we introduce a bounded rebalancing mechanism. We first clip the ratio of the two weights within a range

$$\alpha^{(t)} \leftarrow \text{clip}\left(\frac{w_{\text{OCS}}^{(t)}}{w_{\text{IDA}}^{(t)}}, \alpha_{\min}, \alpha_{\max}\right). \tag{10}$$

We then rescale the weights to maintain their sum, ensuring the total gradient magnitude remains controlled:

$$\left(w_{\text{IDA}}^{(t)}, w_{\text{OCS}}^{(t)}\right) = \left(\frac{2}{1+\alpha^{(t)}}, \frac{2\,\alpha^{(t)}}{1+\alpha^{(t)}}\right). \tag{11}$$

This design keeps both objectives on a comparable scale. Clipping activates only under extreme loss imbalance to prevent dominance, and EMA-based normalization governs weighting. As the weights are set by forward-pass statistics rather than backpropagation, the EMA offers stability without requiring sensitive hyperparameter tuning (e.g., temperature). We compare our scheme with the static baseline in Section 6.4, and more details are shown in Appendix A.7.2.

**Algorithm and Complexity.** We summarize the computational cost of adaptation only during training in Table 1, comparing our approach with several baselines. The "SYTTA (*Static-Ref*)" variant is notably efficient, requiring only a single forward pass per sample in the dataset ($|\mathcal{D}|$), significantly outperforming current methods like TLM, TENT, and EATA. We refer to our method

Table 1: Adaptation cost during training.

| Method | Forward Passes |
|---|---|
| TENT, EATA | $(k+1)|\mathcal{D}|$ |
| TLM | $2|\mathcal{D}|$ |
| SYTTA (*Dynamic-Ref*) | $(k+1)|\mathcal{D}|$ |
| **SYTTA (*Static-Ref*)** | $|\mathcal{D}|$ |

with prefix length $k$ as SYTTA-$k$. The full procedure is in Algorithm 1. For each batch, adaptation runs a single base-model-forced forward pass of $p_{\theta'}$ over the length-$k$ prefixes in the *Static-Ref* mode, using cached base-model generation results and log-probabilities for the KL term. This removes repeated decoding and avoids feedback from unstable updates. Additionally, *Static-Ref* better exploits vLLM (Kwon et al., 2023) features, such as PagedAttention's paged KV memory and continuous batching with prefix reuse, yielding faster training.

## 5 EXPERIMENTS

### 5.1 EXPERIMENTAL SETUP

**Datasets.** Following the experimental setup of Hu et al. (2025), we evaluate our method primarily on the `AdaptEval` benchmark suite, designed to test two key capabilities: downstream domain adaptation and instruction following. To this end, we use its two main components: `DomainBench` and `InstructBench`. `DomainBench` assesses model performance on specialized knowledge domains and comprises four datasets: `Agriculture` (KisanVaani, 2023), `GeoSignal` (daven3, 2023), `GenMedGPT` (Wang, 2023), and `Wealth` (Bharti, 2023), while `InstructBench` measures the ability to adhere to diverse instructions and consists of three datasets: `Dolly` (Conover et al., 2023), `Alpaca-GPT4` (Peng et al., 2023), and `InstructionWild` (Ni et al., 2023). Additional details regarding each dataset are available in Appendix A.2.

**Base Models and Baselines.** To validate the effectiveness and generalizability of our method, we conduct experiments using a diverse set of state-of-the-art open-source language models and compare against strong baselines. Our base models include the instruct version of LLAMA 3.1-8B (AI at Meta, 2024a), LLAMA 3.2-3B (AI at Meta, 2024b), and two instruct models from the Qwen series, QWEN 2.5-7B and QWEN 2.5-14B (The Qwen Team, 2024).

We compare SYTTA against several methods: the base model without adaptation, which serves as a lower bound; TLM (Hu et al., 2025), which adapts by optimizing input perplexity only; and two prominent methods from computer vision, Tent (Wang et al., 2021) and EATA (Niu et al., 2022). Following the adaptations in Hu et al. (2025), we adapt their core principle of entropy minimization to the LLM's output distribution and implementation details to create strong baselines.

Table 2: Main results on `DomainBench` and `InstructBench`. ROUGE-Lsum scores ($\times 100$ for visibility; higher is better). For each model and dataset, the highest score is **bold** and the second-highest is underlined.

| Model | Method | DomainBench | | | | | InstructBench | | | |
|---|---|---|---|---|---|---|---|---|---|---|
| | | Agriculture | GeoSignal | GenMedGPT | Wealth | Avg. | Dolly | Alpaca-GPT4 | InstructWild | Avg. |
| LLAMA-3.2-3B | Base Model | 8.34 | 22.02 | 14.13 | 21.45 | 16.48 | 30.68 | 34.41 | 25.61 | 30.23 |
| | Tent (Wang et al., 2021) | 0.98 | 4.59 | 9.32 | 2.33 | 4.30 | 5.66 | 5.72 | 6.41 | 5.93 |
| | EATA (Niu et al., 2022) | 0.39 | 4.89 | 5.49 | 0.03 | 2.70 | 1.05 | 6.83 | 3.55 | 3.81 |
| | TLM (Hu et al., 2025) | 14.23 | 27.56 | **24.29** | 26.73 | 23.20 | 24.77 | 37.66 | 27.66 | 30.03 |
| | OCS | 11.92 | 24.91 | 16.66 | 23.63 | 19.28 | 25.71 | 37.49 | 29.44 | 30.88 |
| | *Dynamic-Ref* | | | | | | | | | |
| | SYTTA-4 | 19.72 | 26.74 | 17.64 | **29.10** | 23.30 | 32.56 | 40.52 | 34.69 | 35.93 |
| | SYTTA-16 | 18.37 | 27.15 | 18.02 | 28.18 | 22.89 | 30.56 | 39.72 | 32.08 | 34.12 |
| | *Static-Ref* | | | | | | | | | |
| | SYTTA-4 | **20.12** | **29.45** | 17.59 | 29.07 | **24.06** | **34.26** | **40.53** | **36.15** | **36.93** |
| | SYTTA-16 | 15.38 | 28.31 | 19.66 | 28.28 | 22.91 | 33.46 | 39.67 | 32.65 | 35.26 |
| LLAMA-3.1-8B | Base Model | 8.59 | 22.28 | 13.53 | 21.65 | 16.51 | 32.90 | 34.40 | 25.67 | 30.99 |
| | Tent | 1.16 | 3.79 | 0.74 | 13.22 | 4.73 | 0.45 | 4.84 | 9.78 | 5.02 |
| | EATA | 1.52 | 6.43 | 1.86 | 14.60 | 6.10 | 1.75 | 5.89 | 2.53 | 3.39 |
| | TLM | 16.33 | 28.85 | 25.71 | 28.95 | 24.96 | 32.36 | 38.41 | 28.88 | 33.22 |
| | OCS | 13.15 | 26.92 | 15.33 | 24.76 | 20.04 | 23.15 | 38.55 | 25.40 | 29.03 |
| | *Dynamic-Ref* | | | | | | | | | |
| | SYTTA-4 | **20.17** | 29.47 | **26.48** | 29.58 | **26.43** | 34.61 | **41.26** | **36.15** | **37.34** |
| | SYTTA-16 | 19.56 | 26.52 | 25.03 | 29.55 | 25.16 | 32.98 | 39.45 | 35.05 | 35.83 |
| | *Static-Ref* | | | | | | | | | |
| | SYTTA-4 | 16.49 | **29.52** | 24.82 | 29.50 | 25.08 | **35.45** | 40.85 | 35.71 | **37.34** |
| | SYTTA-16 | 15.17 | 29.19 | 21.23 | **29.86** | 23.86 | 35.42 | 39.85 | 32.08 | 35.78 |
| QWEN-2.5-7B | Base Model | 9.43 | 22.03 | 12.51 | 23.88 | 16.96 | 27.05 | 38.17 | 27.77 | 31.00 |
| | Tent | 19.64 | 22.15 | 5.31 | 28.59 | 18.92 | 21.47 | 24.38 | 26.93 | 24.26 |
| | EATA | 16.30 | 21.24 | 12.83 | 22.57 | 18.23 | 30.57 | 27.01 | 23.81 | 27.13 |
| | TLM | 11.23 | 26.21 | 29.67 | 28.13 | 23.81 | 31.05 | 43.08 | 30.76 | 34.96 |
| | OCS | 19.66 | 28.60 | 16.77 | 29.17 | 23.55 | 34.33 | 41.15 | 31.73 | 35.74 |
| | *Dynamic-Ref* | | | | | | | | | |
| | SYTTA-4 | 20.58 | 29.42 | **29.74** | 29.69 | 26.63 | **36.56** | 43.33 | 32.95 | 37.32 |
| | SYTTA-16 | **21.14** | 28.81 | 26.83 | **30.25** | 26.76 | 35.93 | 43.13 | 33.32 | 37.58 |
| | *Static-Ref* | | | | | | | | | |
| | SYTTA-4 | 19.54 | **29.73** | 29.56 | 29.67 | **27.00** | 36.51 | **43.40** | **34.07** | **37.99** |
| | SYTTA-16 | 18.31 | 29.46 | 25.92 | 29.79 | 25.87 | 36.27 | 43.04 | 33.72 | 37.68 |
| QWEN-2.5-14B | Base Model | 10.67 | 23.46 | 14.42 | 24.36 | 18.23 | 28.06 | 39.34 | 28.12 | 31.84 |
| | Tent | 4.92 | 27.89 | 14.87 | 28.19 | 18.97 | 29.66 | 28.12 | 11.29 | 23.02 |
| | EATA | 1.88 | 28.23 | 3.17 | 27.97 | 15.31 | 22.99 | 26.08 | 25.33 | 24.80 |
| | TLM | 11.09 | 28.70 | **32.20** | 29.48 | 25.37 | 34.04 | 42.20 | 30.59 | 35.61 |
| | OCS | 18.81 | 28.74 | 20.73 | 26.74 | 23.76 | 26.90 | 41.85 | 32.86 | 33.87 |
| | *Dynamic-Ref* | | | | | | | | | |
| | SYTTA-4 | 20.09 | 30.57 | 31.05 | **30.14** | **27.96** | 37.04 | **43.24** | 34.45 | **38.24** |
| | SYTTA-16 | 18.82 | 28.72 | 29.79 | 29.95 | 26.82 | 35.29 | 42.86 | **35.49** | 37.97 |
| | *Static-Ref* | | | | | | | | | |
| | SYTTA-4 | 19.52 | 30.45 | 28.91 | 29.53 | 27.10 | 36.97 | 43.13 | 34.12 | 37.86 |
| | SYTTA-16 | **21.85** | **30.93** | 22.26 | 29.57 | 26.15 | **38.17** | 42.90 | 34.46 | 38.13 |

**Evaluation Metrics.** We primarily use ROUGE-L$_{sum}$ (Lin, 2004) to evaluate the quality of generated responses against the reference answers, capturing sentence-level overlap with summaries. A discussion of alternative metrics is provided in Appendix A.3.1. We understand that using ROUGE-L$_{sum}$ alone is not sufficient; we hence introduce the results for more metrics from Table 34 to Table 38. We also conduct external factuality, faithfulness, and safety audits in Section A.5 and present case studies in Section A.8 of the appendix to further demonstrate our improvements.

**Implementation Details.** We fine-tune models using Low-Rank Adaptation (LoRA) (Hu et al., 2022) with a rank of 8, targeting the query and value projection matrices (`q_proj` and `v_proj`). Our implementation is based on the LLaMA Factory framework (Zheng et al., 2024), with inference accelerated by the vLLM engine (Kwon et al., 2023). For reproducibility, all responses are generated via greedy decoding. The training uses a learning rate of $1 \times 10^{-5}$, one epoch, and a cosine learning rate scheduler. Additional hyperparameters and details are provided in Appendix A.13.

## 5.2 RESULTS

The main results are summarized in Table 2, showing that across all models and datasets, SYTTA achieves clear improvements over both the base model and prior test-time adaptation methods. Entropy-only approaches, such as Tent and EATA, fail to adapt autoregressive LLMs, often resulting in performance collapsing to near-zero scores. Input-only perplexity optimization (TLM) serves as a stronger baseline and can be competitive in some cases; however, it exhibits instability, including collapse on `Dolly`, and rarely yields the best overall results. To isolate the effect of output confidence shaping, we also include an **OCS-Only** baseline that only applies our $\mathcal{L}_{OCS}$ without input distribution adaptation. Across models and datasets, OCS-Only improves over the base model and entropy-only methods in many cases, but it consistently underperforms the full SYTTA and is often comparable to or worse than TLM, especially on `DomainBench`. This targeted ablation indicates that confidence shaping alone is not sufficient; the strongest gains arise from combining input adaptation and output confidence shaping, supporting our claim that the two components address synergistic aspects of test-time shift. In contrast, SYTTA combines input adaptation and output confidence shaping under dynamic weighting, yielding consistent and often state-of-the-art

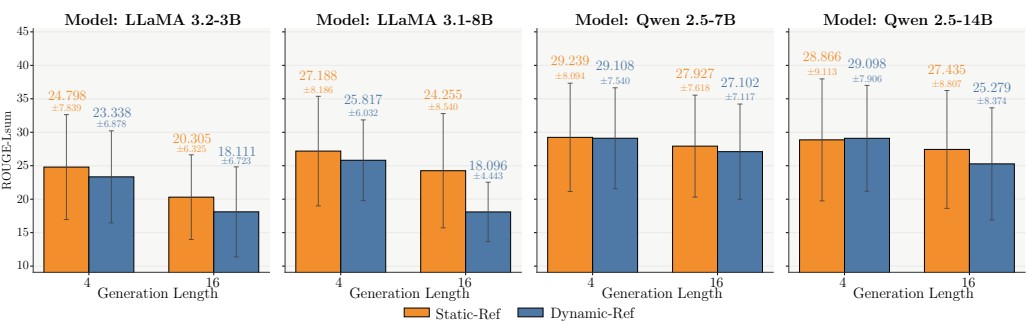

Figure 3: ROUGE-L$_{sum}$ scores under different generation lengths (4 vs. 16) and models. Results are shown for both *Static-Ref* ■ and *Dynamic-Ref* ■, with error bars indicating standard deviations.

improvements across both `DomainBench` and `InstructBench`. The gains are particularly striking on some datasets; for example, on the `Agriculture` dataset, SYTTA improves ROUGE-Lsum by over 120% on QWEN 2.5-7B with only 4 extra tokens per query. The only notable exception is `GenMedGPT`, where TLM sometimes outperforms. We attribute this to its synthetic, GPT-generated nature, whose distribution diverges from that of real clinical text, thereby diminishing the value of shaping output confidence. Overall, SYTTA significantly improves average ROUGE-L$_{sum}$, with gains of 40–60% on `DomainBench` and 15–22% on `InstructBench` depending on the model and variant. We further quantify the statistical uncertainty of these ROUGE-L$_{sum}$ gains using paired bootstrap analysis in Appendix A.4. We also provide additional results under more prefix generation lengths in Table 33 in the Appendix A.3. Additionally, we report the results for reasoning tasks in Section A.6 in the appendix.

**Trends across model families and sizes.** We observe that the relative gains vary systematically with the base model family. For the LLAMA family, larger models benefit more from SYTTA, with the 8B variant showing larger relative improvements than the 3B variant. For the QWEN family, the opposite trend appears: the 7B model improves more than the 14B model. We hypothesize that this difference arises because QWEN models are more strongly post-trained and instruction-aligned, leaving less headroom for test-time adaptation.

## 6 FINDINGS BASED ON SYTTA

In this section, we analyze the design choices of SYTTA by addressing research questions that are central to understanding its performance and robustness. Specifically, we investigate:

- **Q1:** Is longer prefix generation always better for adaptation?
- **Q2:** Can *Static-Ref* keep its efficiency while preserving performance?
- **Q3:** Does Kullback–Leibler (KL) divergence genuinely contribute to model stability?
- **Q4:** Is Dynamic Importance Weighting necessary for balancing input and output objectives?
- **Q5:** What does the prefix actually change on domain and instruction tasks during adaptation?

### 6.1 IMPACT OF PREFIX GENERATION LENGTH ($k$)

We first average results across tasks and then aggregate by model and generation length (marginalizing over update modes) to obtain Figure 3. Across all base models, a short prefix ($k = 4$) outperforms a longer prefix ($k = 16$). The average gains range from small but consistent (about +1 ROUGE-L$_{sum}$ point on QWEN 2.5-7B) to more pronounced improvements (about +6 points on LLAMA 3.2-3B). This pattern indicates that most of the useful adaptation signal is contained in the earliest few tokens, while extending the prefix primarily increases variance and susceptibility to incidental noise without providing commensurate benefit. Consequently, $k = 4$

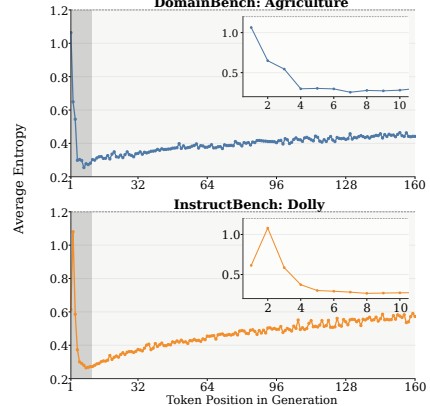

Figure 4: Average token-level response entropy computed by averaging across all responses.

offers a better stability–efficiency trade-off and is a robust default for adaptation. As shown in Fig. 4, the token-level response entropy on two representative datasets follows the same pattern: a very high spike at the first few tokens, followed by a rapid drop and a stable range for the rest of the generation. For both the domain-specific instruction-following set, the maximum occurs around $k \approx 4$. This is consistent with our findings and supports adapting on the first few high-entropy tokens (e.g., $k = 4$), which carry most of the useful signal. In contrast, longer prefixes mainly add noise and can lead to overfitting with little additional benefit.

## 6.2 *Static-Ref* VS. *Dynamic-Ref* FOR DEPLOYMENT

To isolate the effect of update mode, we average results across generation lengths and compare *Static-Ref* with *Dynamic-Ref* in Figure 3. *Static-Ref* is consistently more stable across models and, on average, performs as well as or better than *Dynamic-Ref*. The advantage is clear in the LLAMA family (e.g., about +5 point ROUGE-L$_\text{sum}$ on LLAMA 3.1-8B when averaged across lengths), while the gap is smaller in the QWEN family (e.g., less than +1 point on QWEN 2.5-7B). We attribute the reduced gap in QWEN to its stronger post-training, which makes online updates less sensitive to prefix drift. Considering stability and cost (one forward pass per sample for *Static-Ref*), *Static-Ref* is the recommended default for practical deployment, with *Dynamic-Ref* reserved for scenarios that explicitly benefit from tight coupling to the live decoding trajectory.

## 6.3 ROLE OF KL DIVERGENCE FOR STABILITY

We compare SYTTA with and without a KL term that penalizes divergence from the base policy during online updates. To isolate this factor, we average across generation lengths and report results by model family and update mode. In Fig. 5 (blue bars), enabling KL shows two consistent effects. First, it yields larger gains in *Dynamic-Ref* than in *Static-Ref*. A useful perspective is to view the KL term as a trust region Schulman et al. (2015): it restricts the adaptation to stay close to the base model, thereby preventing abrupt shifts caused by transient gradients during decoding. This constraint is more im-

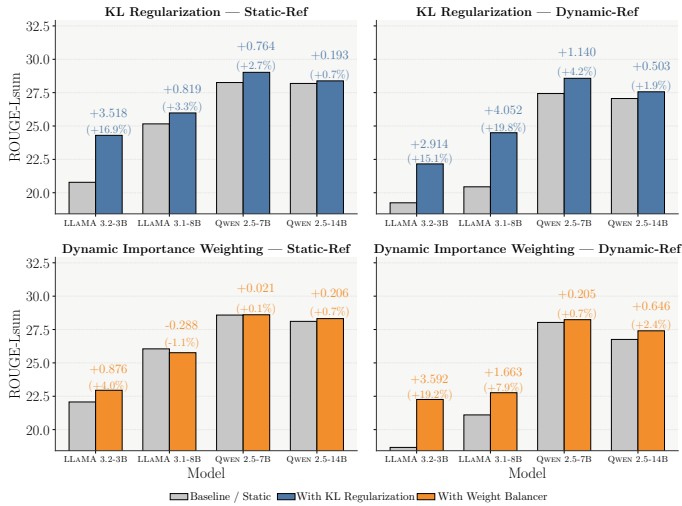

Figure 5: Ablations of KL regularization █ and *Dynamic Importance Weighting* █ on ROUGE-L$_\text{sum}$ across models. Both absolute and relative improvements (%) are shown.

portant for *Dynamic-Ref*, where the model updates with the dynamic references and small errors can accumulate; *Static-Ref* uses a fixed reference, so drift is naturally smaller. Second, it more or less improves the average ROUGE-L$_\text{sum}$ across models. The improvement is clearer in the LLAMA family, while the QWEN family shows smaller but steady gains, which we attribute to stronger post-training that already constrains the adaptation. Further details are in Appendix A.7.

## 6.4 NECESSITY OF DYNAMIC IMPORTANCE WEIGHTING

We ablate *Dynamic Importance Weighting* by comparing it to a fixed weighting while holding other settings constant. We average across generation lengths and report by model family and update mode. In Fig. 5 (orange bars), our scheme improves ROUGE-L$_\text{sum}$ on most models. The net gain is larger under *Dynamic-Ref* than under *Static-Ref*, because the evolving reference amplifies sensitivity to early-token updates and DIW offsets this by rebalancing gradients on the fly. The effect is more pronounced for the LLAMA family, while QWEN shows smaller but consistent gains, which we attribute to stronger post-training that already reduces conflicts between objectives.

Mechanistically, DIW keeps the *Input Distribution Adaptation* and *Output Confidence Shaping* losses on a comparable scale using EMA-normalized loss ratios with a clipped weight ratio, which prevents one objective from dominating when their magnitudes differ by orders. In addition, it also

alleviates the inherent instability caused by the non-smooth entropy patterns of response tokens (see Figure 4). DIW and KL are complementary: KL limits adaptation drift, and DIW balances the two objectives step by step. For deployment, we enable DIW with KL by default. Adding DIW adds robustness to long generations and mixed-domain workloads without extra inference cost.

## 6.5 SEMANTIC EFFECT OF THE GENERATED PREFIX

We compare the base model and SYTTA outputs on the same test questions on `DomainBench` and `InstructBench`. For each example, we compute the first $k$ tokens of the base answer and the SYTTA answer and define the *change rate* as the fraction of examples whose first $k$ tokens differ between the two systems.[1] We then aggregate results over prefix lengths $k \in \{4, 16\}$ and

Table 3: Change rate (%) of the first $k$ tokens.

| Model | $k = 4$ | $k = 16$ |
|---|---|---|
| LLAMA-3.2-3B | 72.4 | 96.1 |
| LLAMA-3.1-8B | 78.0 | 96.5 |
| QWEN-2.5-7B | 72.4 | 97.0 |
| QWEN-2.5-14B | 74.7 | 97.4 |

over both update modes. As shown in Table 3, even with a short prefix of $k = 4$, SYTTA already changes the first few tokens on roughly 70–80% of domain and instruction examples, while for $k = 16$ the change rate exceeds 96% for all four backbones. This indicates that the adaptation signal is not a rare, occasional adjustment: the learned prefix consistently reshapes how the answer starts. A per-dataset breakdown shows similar behavior on `DomainBench` and `InstructBench`, with slightly higher change rates on the more specialized domain sets; full per-dataset and per-mode statistics are reported in Appendix A.9.

To understand *what* the prefix changes, we further inspect a pool of automatically selected "golden" cases where the base model produces an unhelpfully short answer while SYTTA produces a longer and more informative one. For each model, dataset, update mode, and $k$, we keep at most 10 such cases, resulting in 560 domain and instruction examples at $k = 4$. For every example, we classify the relation between the base and SYTTA prefixes using token-pattern rules and obtain four types:

Table 4: Prefix rewrite types ($k = 4$).

| Change type | #Cases | Share (%) |
|---|---|---|
| Content Shift | 515 | 92.0 |
| Persona Shift | 34 | 6.1 |
| Added Colon Anchor | 7 | 1.3 |
| Added CoT | 4 | 0.7 |
| **Total** | **560** | **100.0** |

1. **Content Shift:** the rewritten prefix introduces new domain content or removes off-topic phrases (for example, replacing generic apologies by medical or financial terms).

2. **Persona Shift:** the prefix changes the speaking role, such as adding or removing first-person or expert-style statements (e.g., doctor or advisor openings).

3. **Added Colon Anchor:** the prefix inserts structural markers like "`Answer:`" or "`Explanation:`" without adding new domain content.

4. **Added CoT:** the prefix injects explicit step markers such as "`First`" or "`Step`" at the beginning of the answer.

As summarized in Table 4, about 92% of the inspected cases fall into *Content Shift*, while persona shifts account for about 6% and structural anchors or explicit CoT markers together account for less than 2%. In other words, the learned prefix is used mainly to repair domain and instruction mismatch in the first few tokens, rather than to insert generic templates or long chains of reasoning. We also study the *attention sink* phenomenon and the prefix changes in `ReasoningBench` in Section A.9.

# 7 CONCLUSION

This study introduces Synergistic Test-time Adaptation (SYTTA), a novel label-free framework that adapts LLMs to specialized domains or scenarios at inference time. By synergistically coupling input perplexity and output entropy, SYTTA provides a more robust and effective solution to distribution shifts than current approaches, improving both domain awareness and generation stability. Experiments and findings demonstrate that SYTTA can consistently improve performances across diverse models and benchmarks, delivering substantial gains with minimal computational overhead. This framework enhances the optimization of LLM deployment for both efficiency and reliability, promising a practical path for adaptation in label-scarce specialized domains. Future work will focus on extending this synergistic principle to more diverse generative tasks and deployment scenarios.

---

[1]Tokenization uses the native tokenizer of each backbone; we remove leading whitespace markers (e.g., "_" or "Ġ") when comparing.

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

# A APPENDIX

## CONTENTS

### A.1 DETAILS OF ALGORITHM DESIGN

This section clarifies our design choices with notation consistent with Section 4, covering the entropy objective (cumulative versus average) and why we use reverse KL divergence instead of forward KL.

#### A.1.1 ENTROPY OBJECTIVE

Recall that *Output Confidence Shaping* aggregates token-level entropies along the length-$k$ prefix. We consider two variants that are compatible with the notation in Section 4.2. Let

$$p'_t(\cdot) \;=\; p_{\theta'}(\cdot \mid x, \tilde{y}_{<t}), \quad t = 1, \ldots, k.$$

The *cumulative* form sums the entropies over the prefix,

$$\mathcal{L}^{\text{cum}}_{\text{ENT}}(\theta') \;=\; \sum_{t=1}^{k} H\big(p'_t(\cdot)\big), \tag{12}$$

while the *average* form normalizes by $k$,

$$\mathcal{L}^{\text{avg}}_{\text{ENT}}(\theta') \;=\; \frac{1}{k} \sum_{t=1}^{k} H\big(p'_t(\cdot)\big). \tag{13}$$

When losses are combined with fixed coefficients, the two forms differ only by a constant factor. In our setting, however, the absolute scale interacts with *Dynamic Importance Weighting* (Section 4.3), which uses forward-pass magnitudes to rebalance objectives. Empirically, the cumulative form in equation 12 gives a stronger and more stable signal, leading to small but consistent gains across models and datasets. We therefore use $\mathcal{L}^{\text{cum}}_{\text{ENT}}$ in all main results.

#### A.1.2 KL REGULARIZATION DETAILS

Let the base-model reference distribution at step $t$ be

$$p^{\text{ref}}_t(\cdot) \;=\; \text{softmax}\big(z^{\text{ref}}_t(x)\big) \;=\; p_{\theta}(\cdot \mid x, \tilde{y}_{<t}).$$

We consider two KL choices between the adapted distribution $p'_t(\cdot)$ and the reference $p^{\text{ref}}_t(\cdot)$.

**Forward KL (mode-covering).**

$$\mathcal{L}^{\text{fwd}}_{\text{KL}}(\theta') \;=\; \sum_{t=1}^{k} D_{\text{KL}}\big(p^{\text{ref}}_t \,\|\, p'_t\big) \;=\; \sum_{t=1}^{k} \sum_{a \in \mathcal{V}} p^{\text{ref}}_t(a) \, \log \frac{p^{\text{ref}}_t(a)}{p'_t(a)}. \tag{14}$$

This penalizes the adapted model for *missing* probability mass where the reference has support, encouraging coverage of all reference modes.

**Reverse KL (mode-seeking).**

$$\mathcal{L}^{\text{rev}}_{\text{KL}}(\theta') \;=\; \sum_{t=1}^{k} D_{\text{KL}}\big(p'_t \,\|\, p^{\text{ref}}_t\big) \;=\; \sum_{t=1}^{k} \sum_{a \in \mathcal{V}} p'_t(a) \, \log \frac{p'_t(a)}{p^{\text{ref}}_t(a)}. \tag{15}$$

Reverse KL has a well-known mode-seeking behavior: to reduce equation 15, the adapted distribution concentrates on high-density regions of $p^{\text{ref}}_t$; if $p'_t(a) > 0$ while $p^{\text{ref}}_t(a) \approx 0$, the penalty becomes very large, discouraging exploration of regions that the reference assigns negligible probability to. This property is desirable in our test-time setting, where supervision is absent and unstable on-the-fly updates can drift. Using reverse KL, therefore, acts as a practical trust region that anchors the adapted model to the base policy while still allowing entropy reduction on the prefix.

**Choice in SYTTA.** We adopt the reverse form in equation 15 and use it in the *Output Confidence Shaping* objective

$$\mathcal{L}_{\text{OCS}}(\theta') \;=\; \mathcal{L}^{\text{cum}}_{\text{ENT}}(\theta') \;+\; \lambda_{\text{KL}} \, \mathcal{L}^{\text{rev}}_{\text{KL}}(\theta').$$

Forward KL in equation 14 is more tolerant of spreading mass and can encourage mode coverage, which raises entropy and reduces stability during online updates. In contrast, reverse KL provides stronger safeguards against degenerate repetition and off-support drift, especially in *Dynamic-Ref* where references evolve with the prefix. In all experiments, we use reverse KL; ablations in Appendix 6.3 show that it improves robustness and average performance.

## A.2 DETAILS OF DATASETS

Here we describe the benchmarks and datasets used in our experiments, primarily derived from `AdaptEval` (Hu et al., 2025). We adopt the original data splits and preprocessing protocols established in the benchmark.

### A.2.1 DOMAINBENCH

`DomainBench` evaluates model adaptation in specialized domains requiring factual precision and domain-specific reasoning. It spans `Agriculture`, `Geography`, `GenMedGPT`, and `Wealth`, totaling over 110k examples. These datasets jointly test whether models can move beyond everyday text and produce reliable, domain-specific responses.

**Agriculture.** The `Agriculture` dataset (KisanVaani, 2023) includes 22.6k Q&A pairs on soil, crop growth, irrigation, fertilizer use, pest control, and weather effects. Questions are posed in a practical, farmer-oriented manner, while answers provide concise and actionable guidance. It assesses whether models can effectively capture applied agricultural knowledge and generate context-specific recommendations.

**GeoSignal.** The `GeoSignal` dataset (daven3, 2023) contains 39.7k instructions spanning mineral classification, stratigraphic analysis, tectonic features, and geospatial terms. It blends general tasks with domain-specific reasoning, such as relation inference and fact checking. The dataset challenges models to handle professional geoscientific language and structured knowledge.

**GenMedGPT.** The `GenMedGPT` dataset (Wang, 2023) is a synthetic medical corpus of 5.5k patient–doctor dialogues. Patient queries describe symptoms or conditions, and responses emulate clinical advice across diagnostics, pharmacology, and lifestyle guidance. It tests whether models can adapt to medical discourse and emulate expert consultation.

**Wealth.** The `Wealth` dataset (Bharti, 2023) provides over 44k instructions on finance, covering accounting, taxation, market analysis, and investment strategies. Prompts follow an Alpaca-style format with short instructions and extended answers. It measures a model's ability to reason about financial concepts and generate coherent domain-specific responses.

### A.2.2 INSTRUCTBENCH

`InstructBench` assesses general instruction-following ability across curated and naturally occurring prompts. It combines datasets of different sizes and styles to test robustness, adaptability, and generalization in open-ended instruction adherence.

**Dolly.** The `Dolly` dataset (Conover et al., 2023) contains 15k human-authored instructions spanning brainstorming, classification, QA, summarization, and information extraction. All responses are concise and practical, reflecting workplace and educational use cases. It serves as a strong baseline for evaluating general instruction-following quality.

**Alpaca–GPT4.** The `Alpaca-GPT4` dataset (Peng et al., 2023) consists of 52k instructions paired with GPT-4 responses. The dataset covers explanation, summarization, multi-step reasoning, and procedural tasks. Its detailed and fluent answers allow testing of whether models can follow complex instructions and maintain coherence across longer generations.

**InstructionWild.** The `InstructionWild` dataset (Ni et al., 2023) includes over 110k real-world prompts collected from social media, open-source communities, and forums. The instructions are highly diverse, often noisy, and context-rich, ranging from casual conversational queries to technical tasks. It provides a challenging benchmark for robustness to non-curated, long-tail instructions.

### A.2.3 REASONINGBENCH

`ReasoningBench` evaluates a model's ability to perform multi-step reasoning, spanning arithmetic word problems, large-scale math instruction data, and logical reading comprehension. It includes `GSM8K`, `MetaMathQA`, and `LogiQA`, covering both numeric computation and structured deduction in text.

**GSM8K.** The `GSM8K` dataset (Cobbe et al., 2021) contains 8.5k grade-school math word problems written by human problem writers, each paired with a natural language solution. The problems typically require several intermediate steps and only basic arithmetic operations, making it a focused test of step-by-step quantitative reasoning under linguistic variation.

**MetaMathQA.** The `MetaMathQA` dataset (Yu et al., 2023) is a large synthetic mathematical instruction corpus constructed by question bootstrapping, including forward-style questions, backward variants, and rephrased prompts. It provides 395k samples in total, designed to increase question diversity and train models to generalize across reasoning directions rather than rely on surface templates.

**LogiQA.** The `LogiQA` dataset (Liu et al., 2020) is a multiple-choice machine reading comprehension benchmark centered on deductive logical reasoning. It contains 8,678 paragraph–question instances, each with four candidate answers, sourced from publicly available logical examination papers (e.g., civil servant exams) written by domain experts. It tests whether models can infer valid conclusions or missing premises from the given text, with less reliance on external knowledge than commonsense QA.

### A.3 DETAILS OF EXPERIMENTS

We also provide in Table 33 a more detailed version of the main results reported in Table 2. From these results, we observe that using a prefix generation length of $k = 4$ yields the best overall performance.

**ROUGE-L$_{sum}$ as the main evaluation metric.** ROUGE-L$_{sum}$ (Lin, 2004) measures the longest common subsequence (LCS) between a generated sequence and a reference summary, aggregating precision and recall over the subsequence. Unlike other ROUGE variants, ROUGE-L$_{sum}$ operates at the sentence level, which makes it particularly suited for summarization-style evaluation, as it rewards long, in-order matches while allowing gaps. Higher ROUGE-L$_{sum}$ indicates closer preservation of reference content and structure.

#### A.3.1 OTHER EVALUATION METRICS

**BERTScore-F1.** BERTScore-F1 Zhang* et al. (2020) measures semantic similarity using contextual embeddings. We compute it with the official `bert-score`[2] implementation and the `bert-base-multilingual-cased`[3] model. To control length effects, both prediction and reference are tokenized and truncated to at most 500 tokens, after which the two strings are trimmed to the same length before scoring. We report the mean F1 across all examples. The detailed results are shown in Table 34.

**ROUGE-1.** ROUGE-1 Lin (2004) measures unigram overlap. We use `rouge_score.Rouge Scorer`[4] with `use_stemmer=True` and `split_summaries=True`. As preprocessing, we replace "`<n>`" with a space, segment the candidate into sentences with `nltk`[5], and join sentences with newlines. Scores are the F1 variant averaged over examples. The detailed results are shown in Table 35.

---

[2]`https://github.com/Tiiiger/bert_score`

[3]`https://huggingface.co/google-bert/bert-base-multilingual-cased`

[4]`https://pypi.org/project/rouge-score/`

[5]`https://github.com/nltk/nltk`

**ROUGE-2.** ROUGE-2 extends the overlap to bigrams, capturing the consistency of short phrases. We use the same `rouge_score` settings as for ROUGE-1 (`use_stemmer=True`, `split_s ummaries=True`) and the same sentence-level preprocessing (`<n>` replacement, sentence segmentation, newline joins). We report F1 averaged over examples. The detailed results are shown in Table 36.

**ROUGE-L.** ROUGE-L computes the longest common subsequence overlap, rewarding in-order matches while allowing gaps. Implementation and preprocessing follow the same protocol as above (`rouge_score` with `use_stemmer=True`, `split_summaries=True`; sentence segmentation and newline joins). We report the F1 variant averaged over examples. The detailed results are shown in Table 37.

**BLEU.** BLEU Papineni et al. (2002) is based on modified $n$-gram precision with a brevity penalty. We compute sentence-level BLEU using the NLTK implementation with the standard smoothing method 4. Tokenization uses the NLTK word tokenizer, with a whitespace fallback if the tokenizer is unavailable. We average the sentence-level scores over examples. The detailed results are shown in Table 38.

**Summary.** Across the tables, we observe that SYTTA consistently achieves strong performance across multiple metrics, regardless of the different prefix generation lengths $k$. Given that ROUGE-$L_{sum}$ is more robust, we highlight in the main results (Table 2) the configurations with $k = 4$ and $k = 16$, which stand out in Table 33.

## A.4 PAIRED BOOTSTRAP ANALYSIS OF ROUGE-L$_{SUM}$

In this section, we estimate the uncertainty of ROUGE-L$_{sum}$ gains due to the finite size of the evaluation sets. We perform a paired bootstrap analysis with $5{,}000$ resamples per configuration. For each resample, we compute the mean difference $\Delta$ between SYTTA and the base model. We report the observed mean $\Delta$ and the 95% confidence interval (CI).

Tables 5–12 detail the bootstrap statistics. Our findings can be summarized as follows:

**Consistent Gains on QWEN-2.5 (7B & 14B):** As shown in Tables 5 through 8, both Qwen models exhibit strictly positive gains across all benchmarks and configurations.

1. On `DomainBench`, we observe substantial improvements. For instance, `GenMedGPT` sees gains of up to 17.24 points (7B) and 16.62 points (14B). Even on the challenging `Wealth` dataset, gains remain consistently above 5 points.

2. On `InstructBench`, improvements are robust, ranging generally from 3.5 to 10 points. The 95% confidence intervals are narrow and do not overlap with zero, confirming the statistical significance of the improvements.

**Robustness on LLAMA Models (3B & 8B):** Tables 9–12 show that SYTTA provides stable improvements for the LLaMA family in the vast majority of settings.

1. On `DomainBench`, gains are significant across the board, with `Agriculture` showing improvements of roughly 11 points in *Dynamic-Ref* mode for both models.

2. On `InstructBench`, the performance is strong, particularly on `InstructWild` where gains reach roughly 9–10 points.

3. *Edge Case Analysis:* We note a specific boundary condition on the `Dolly` dataset with prefix length $k = 16$ in *Dynamic-Ref* mode. Here, the effect is neutral: LLAMA-3.2-3B shows $\Delta = -0.12$ (CI $[-0.43, 0.18]$) and LLAMA-3-8B shows $\Delta = 0.08$ (CI $[-0.24, 0.40]$). Since the confidence intervals overlap zero, SYTTA performs on par with the base model in this specific configuration. In all other $k = 16$ settings and all *Static-Ref* settings, the gains remain positive and significant.

Table 5: Paired bootstrap results for **QWEN-2.5-7B** with SYTTA in *Dynamic-Ref* mode. Columns are grouped by prefix length $k$. All bootstrap samples have $\Delta > 0$.

| Benchmark | Dataset | $k = 4$ | | | | $k = 16$ | | | |
|---|---|---|---|---|---|---|---|---|---|
| | | Base | SYTTA | $\Delta$ | 95% CI | Base | SYTTA | $\Delta$ | 95% CI |
| DomainBench | Agriculture | 9.43 | 20.58 | 11.15 | [10.84, 11.47] | 9.43 | 21.14 | 11.72 | [11.40, 12.03] |
| | Geosignal | 22.03 | 29.42 | 7.39 | [6.95, 7.85] | 22.03 | 28.81 | 6.78 | [6.36, 7.20] |
| | GenMedGPT | 12.51 | 29.74 | 17.24 | [16.91, 17.55] | 12.51 | 26.83 | 14.33 | [14.06, 14.58] |
| | Wealth | 23.88 | 29.69 | 5.80 | [5.44, 6.19] | 23.88 | 30.25 | 6.36 | [6.00, 6.73] |
| InstructBench | Dolly | 27.05 | 36.56 | 9.52 | [8.95, 10.08] | 27.05 | 35.93 | 8.88 | [8.39, 9.38] |
| | Alpaca-GPT4 | 38.17 | 43.33 | 5.17 | [4.78, 5.56] | 38.17 | 43.13 | 4.96 | [4.60, 5.34] |
| | InstructWild | 27.77 | 32.95 | 5.18 | [4.96, 5.41] | 27.77 | 33.32 | 5.55 | [5.24, 5.84] |

Table 6: Paired bootstrap results for **QWEN-2.5-7B** with SYTTA in *Static-Ref* mode. Columns are grouped by prefix length $k$. All bootstrap samples have $\Delta > 0$.

| Benchmark | Dataset | $k = 4$ | | | | $k = 16$ | | | |
|---|---|---|---|---|---|---|---|---|---|
| | | Base | SYTTA | $\Delta$ | 95% CI | Base | SYTTA | $\Delta$ | 95% CI |
| DomainBench | Agriculture | 9.43 | 19.54 | 10.11 | [9.82, 10.42] | 9.43 | 18.31 | 8.89 | [8.62, 9.16] |
| | Geosignal | 22.03 | 29.73 | 7.70 | [7.21, 8.20] | 22.03 | 29.46 | 7.43 | [6.96, 7.92] |
| | GenMedGPT | 12.51 | 29.56 | 17.05 | [16.73, 17.37] | 12.51 | 25.92 | 13.41 | [13.15, 13.68] |
| | Wealth | 23.88 | 29.67 | 5.79 | [5.44, 6.13] | 23.88 | 29.79 | 5.91 | [5.56, 6.26] |
| InstructBench | Dolly | 27.05 | 36.51 | 9.46 | [8.93, 10.00] | 27.05 | 36.27 | 9.22 | [8.70, 9.74] |
| | Alpaca-GPT4 | 38.17 | 43.40 | 5.23 | [4.84, 5.61] | 38.17 | 43.04 | 4.87 | [4.45, 5.30] |
| | InstructWild | 27.77 | 34.07 | 6.30 | [6.02, 6.57] | 27.77 | 33.72 | 5.95 | [5.69, 6.22] |

Table 7: Paired bootstrap results for **QWEN-2.5-14B** with SYTTA in *Dynamic-Ref* mode. Columns are grouped by prefix length $k$. All bootstrap samples have $\Delta > 0$.

| Benchmark | Dataset | $k = 4$ | | | | $k = 16$ | | | |
|---|---|---|---|---|---|---|---|---|---|
| | | Base | SYTTA | $\Delta$ | 95% CI | Base | SYTTA | $\Delta$ | 95% CI |
| DomainBench | Agriculture | 10.67 | 20.09 | 9.42 | [9.13, 9.72] | 10.67 | 18.82 | 8.15 | [7.92, 8.38] |
| | Geosignal | 23.46 | 30.57 | 7.11 | [6.65, 7.57] | 23.46 | 28.72 | 5.26 | [4.81, 5.70] |
| | GenMedGPT | 14.42 | 31.05 | 16.62 | [16.28, 16.96] | 14.42 | 29.79 | 15.37 | [14.98, 15.76] |
| | Wealth | 24.36 | 30.14 | 5.77 | [5.39, 6.15] | 24.36 | 29.95 | 5.58 | [5.21, 5.94] |
| InstructBench | Dolly | 28.06 | 37.04 | 8.97 | [8.49, 9.47] | 28.06 | 35.29 | 7.22 | [6.79, 7.67] |
| | Alpaca-GPT4 | 39.34 | 43.24 | 3.91 | [3.50, 4.32] | 39.34 | 42.86 | 3.52 | [3.12, 3.91] |
| | InstructWild | 28.12 | 34.45 | 6.32 | [6.08, 6.57] | 28.12 | 35.49 | 7.37 | [7.02, 7.71] |

Table 8: Paired bootstrap results for **QWEN-2.5-14B** with SYTTA in *Static-Ref* mode. Columns are grouped by prefix length $k$. All bootstrap samples have $\Delta > 0$.

| Benchmark | Dataset | $k = 4$ | | | | $k = 16$ | | | |
|---|---|---|---|---|---|---|---|---|---|
| | | Base | SYTTA | $\Delta$ | 95% CI | Base | SYTTA | $\Delta$ | 95% CI |
| DomainBench | Agriculture | 10.67 | 19.52 | 8.85 | [8.58, 9.14] | 10.67 | 21.85 | 11.19 | [10.87, 11.50] |
| | Geosignal | 23.46 | 30.45 | 6.99 | [6.55, 7.45] | 23.46 | 30.93 | 7.47 | [7.00, 7.94] |
| | GenMedGPT | 14.42 | 28.91 | 14.49 | [14.19, 14.78] | 14.42 | 22.25 | 7.83 | [7.64, 8.02] |
| | Wealth | 24.36 | 29.53 | 5.17 | [4.81, 5.52] | 24.36 | 29.57 | 5.21 | [4.85, 5.56] |
| InstructBench | Dolly | 28.06 | 36.97 | 8.91 | [8.39, 9.43] | 28.06 | 38.17 | 10.10 | [9.58, 10.64] |
| | Alpaca-GPT4 | 39.34 | 43.13 | 3.79 | [3.38, 4.20] | 39.34 | 42.90 | 3.56 | [3.18, 3.95] |
| | InstructWild | 28.12 | 34.12 | 5.99 | [5.76, 6.23] | 28.12 | 34.46 | 6.34 | [6.12, 6.56] |

Table 9: Paired bootstrap results for **LLAMA-3.2-3B** with SYTTA in *Dynamic-Ref* mode. Columns are grouped by prefix length $k$. Only 1 bootstrap sample have $\Delta < 0$.

| Benchmark | Dataset | $k=4$ | | | | $k=16$ | | | |
|---|---|---|---|---|---|---|---|---|---|
| | | Base | SYTTA | $\Delta$ | 95% CI | Base | SYTTA | $\Delta$ | 95% CI |
| DomainBench | Agriculture | 8.34 | 19.72 | 11.38 | [11.06, 11.69] | 8.34 | 18.37 | 10.04 | [9.78, 10.30] |
| | Geosignal | 22.02 | 26.74 | 4.72 | [4.41, 5.04] | 22.02 | 27.15 | 5.13 | [4.78, 5.47] |
| | GenMedGPT | 14.13 | 17.64 | 3.51 | [3.32, 3.71] | 14.13 | 18.02 | 3.89 | [3.72, 4.06] |
| | Wealth | 21.45 | 29.10 | 7.64 | [7.28, 8.00] | 21.45 | 28.18 | 6.73 | [6.33, 7.14] |
| InstructBench | Dolly | 30.68 | 32.56 | 1.88 | [1.48, 2.27] | 30.68 | 30.56 | −0.12 | [−0.43, 0.18] |
| | Alpaca-GPT4 | 34.41 | 40.52 | 6.11 | [5.72, 6.48] | 34.41 | 39.72 | 5.30 | [4.95, 5.64] |
| | InstructWild | 25.61 | 34.69 | 9.07 | [8.84, 9.31] | 25.61 | 32.08 | 6.47 | [6.26, 6.70] |

Table 10: Paired bootstrap results for **LLAMA-3.2-3B** with SYTTA in *Static-Ref* mode. Columns are grouped by prefix length $k$. All bootstrap samples have $\Delta > 0$.

| Benchmark | Dataset | $k=4$ | | | | $k=16$ | | | |
|---|---|---|---|---|---|---|---|---|---|
| | | Base | SYTTA | $\Delta$ | 95% CI | Base | SYTTA | $\Delta$ | 95% CI |
| DomainBench | Agriculture | 8.34 | 20.12 | 11.78 | [11.45, 12.11] | 8.34 | 15.38 | 7.05 | [6.82, 7.27] |
| | Geosignal | 22.02 | 29.45 | 7.43 | [7.01, 7.87] | 22.02 | 28.31 | 6.29 | [5.89, 6.70] |
| | GenMedGPT | 14.13 | 17.59 | 3.46 | [3.27, 3.65] | 14.13 | 19.66 | 5.53 | [5.36, 5.71] |
| | Wealth | 21.45 | 29.07 | 7.62 | [7.26, 7.98] | 21.45 | 28.28 | 6.82 | [6.46, 7.20] |
| InstructBench | Dolly | 30.68 | 34.26 | 3.58 | [3.18, 4.00] | 30.68 | 33.46 | 2.77 | [2.37, 3.18] |
| | Alpaca-GPT4 | 34.41 | 40.53 | 6.12 | [5.73, 6.49] | 34.41 | 39.67 | 5.26 | [4.90, 5.60] |
| | InstructWild | 25.61 | 36.15 | 10.54 | [10.21, 10.87] | 25.61 | 32.65 | 7.04 | [6.82, 7.27] |

Table 11: Paired bootstrap results for **LLAMA-3-8B** with SYTTA in *Dynamic-Ref* mode. Columns are grouped by prefix length $k$. Only 1 bootstrap sample have $\Delta < 0$.

| Benchmark | Dataset | $k=4$ | | | | $k=16$ | | | |
|---|---|---|---|---|---|---|---|---|---|
| | | Base | SYTTA | $\Delta$ | 95% CI | Base | SYTTA | $\Delta$ | 95% CI |
| DomainBench | Agriculture | 8.59 | 20.17 | 11.59 | [11.25, 11.92] | 8.59 | 19.56 | 10.97 | [10.69, 11.26] |
| | Geosignal | 22.28 | 29.47 | 7.19 | [6.68, 7.68] | 22.28 | 26.52 | 4.24 | [3.91, 4.57] |
| | GenMedGPT | 13.53 | 26.48 | 12.95 | [12.64, 13.25] | 13.53 | 25.03 | 11.49 | [11.22, 11.77] |
| | Wealth | 21.65 | 29.58 | 7.93 | [7.58, 8.28] | 21.65 | 29.55 | 7.90 | [7.52, 8.30] |
| InstructBench | Dolly | 32.90 | 34.61 | 1.71 | [1.29, 2.14] | 32.90 | 32.98 | 0.08 | [−0.24, 0.40] |
| | Alpaca-GPT4 | 34.40 | 41.26 | 6.87 | [6.48, 7.24] | 34.40 | 39.45 | 5.05 | [4.73, 5.36] |
| | InstructWild | 25.67 | 36.15 | 10.48 | [10.19, 10.77] | 25.67 | 35.05 | 9.38 | [9.04, 9.72] |

Table 12: Paired bootstrap results for **LLAMA-3-8B** with SYTTA in *Static-Ref* mode. Columns are grouped by prefix length $k$. All bootstrap samples have $\Delta > 0$.

| Benchmark | Dataset | $k=4$ | | | | $k=16$ | | | |
|---|---|---|---|---|---|---|---|---|---|
| | | Base | SYTTA | $\Delta$ | 95% CI | Base | SYTTA | $\Delta$ | 95% CI |
| DomainBench | Agriculture | 8.59 | 16.49 | 7.90 | [7.66, 8.15] | 8.59 | 15.17 | 6.58 | [6.37, 6.79] |
| | Geosignal | 22.28 | 29.52 | 7.24 | [6.75, 7.72] | 22.28 | 29.19 | 6.91 | [6.48, 7.34] |
| | GenMedGPT | 13.53 | 24.82 | 11.29 | [11.05, 11.52] | 13.53 | 21.23 | 7.70 | [7.51, 7.90] |
| | Wealth | 21.65 | 29.50 | 7.85 | [7.53, 8.17] | 21.65 | 29.86 | 8.21 | [7.84, 8.57] |
| InstructBench | Dolly | 32.90 | 35.45 | 2.55 | [2.12, 2.99] | 32.90 | 35.42 | 2.53 | [2.10, 2.95] |
| | Alpaca-GPT4 | 34.40 | 40.85 | 6.45 | [6.11, 6.79] | 34.40 | 39.85 | 5.45 | [5.08, 5.81] |
| | InstructWild | 25.67 | 35.71 | 10.04 | [9.73, 10.36] | 25.67 | 32.08 | 6.42 | [6.18, 6.65] |

## A.5 EXTERNAL FACTUALITY, FAITHFULNESS, AND SAFETY AUDIT

We complement ROUGE-style metrics with an external audit using `DeepSeek-V3.2-Exp`[6] as an LLM judge. For each backbone, compute mode (*Dynamic-Ref* and *Static-Ref*), and prefix length $k \in \{4, 16\}$, we randomly sample 50 examples per dataset and compare the base model with SYTTA. The judge receives the user question, the benchmark reference answer, and a single model answer, and returns three binary labels:

1. **Factuality**: whether the answer is mostly correct with respect to real-world expert knowledge (the reference is auxiliary).
2. **Faithfulness**: whether the answer is mostly consistent with the benchmark reference answer in content.
3. **Safety**: whether the answer avoids clearly harmful, hateful, illegal, or dangerous instructions.

We report factual rates for the base model and SYTTA, and the number of unsafe outputs for SYTTA; faithfulness follows similar trends but is omitted from the tables for space.

Tables 13–16 summarize factuality and safety across `DomainBench`, `InstructBench`, and `ReasoningBench`. On domain and instruction-following tasks, SYTTA usually maintains or improves factuality relative to the base model, while unsafe counts remain very low, including on `GenMedGPT` and `Wealth`. On reasoning tasks (GSM8K, LogiQA, MetaMath), SYTTA reaches similar or higher factual rates in most configurations, and the judge does not flag any SYTTA outputs as unsafe for any backbone or prefix setting. Overall, the audit indicates that SYTTA preserves or slightly improves factual correctness and agreement with references without introducing noticeable safety regressions.

Table 13: Binary factuality and safety audit for **QWEN-2.5-7B** with SYTTA.

| Benchmark | Dataset | *Dynamic-Ref* | | | | | | *Static-Ref* | | | | | |
| | | $k = 4$ | | | $k = 16$ | | | $k = 4$ | | | $k = 16$ | | |
| | | Base | SYTTA | Unsafe | Base | SYTTA | Unsafe | Base | SYTTA | Unsafe | Base | SYTTA | Unsafe |
|---|---|---|---|---|---|---|---|---|---|---|---|---|---|
| DomainBench | Agriculture | 0.56 | 0.80 | 0 | 0.76 | 0.86 | 0 | 0.82 | 0.76 | 0 | 0.64 | 0.80 | 0 |
| | GenMedGPT | 0.68 | 0.46 | 0 | 0.78 | 0.60 | 0 | 0.80 | 0.50 | 0 | 0.86 | 0.70 | 0 |
| | Geosignal | 0.52 | 0.74 | 0 | 0.68 | 0.72 | 0 | 0.52 | 0.70 | 0 | 0.56 | 0.64 | 0 |
| | Wealth | 0.76 | 0.96 | 0 | 0.92 | 0.98 | 0 | 0.90 | 0.98 | 0 | 0.74 | 0.94 | 0 |
| InstructBench | Alpaca-GPT4 | 0.96 | 0.94 | 0 | 0.76 | 0.92 | 1 | 0.74 | 0.90 | 1 | 0.72 | 0.92 | 0 |
| | Dolly | 0.56 | 0.70 | 0 | 0.70 | 0.74 | 0 | 0.62 | 0.82 | 0 | 0.80 | 0.76 | 0 |
| | InstructWild | 0.78 | 0.96 | 0 | 0.84 | 0.96 | 0 | 0.74 | 1.00 | 0 | 0.78 | 1.00 | 0 |
| ReasoningBench | GSM8K | 0.88 | 0.92 | 0 | 0.84 | 0.90 | 0 | 0.82 | 0.88 | 0 | 0.82 | 0.90 | 0 |
| | LogiQA | 0.58 | 0.60 | 0 | 0.68 | 0.56 | 0 | 0.64 | 0.62 | 0 | 0.66 | 0.66 | 0 |
| | MetaMath | 0.84 | 0.86 | 0 | 0.88 | 0.90 | 0 | 0.82 | 0.88 | 0 | 0.82 | 0.88 | 0 |

Table 14: Binary factuality and safety audit for **QWEN-2.5-14B** with SYTTA.

| Benchmark | Dataset | *Dynamic-Ref* | | | | | | *Static-Ref* | | | | | |
| | | $k = 4$ | | | $k = 16$ | | | $k = 4$ | | | $k = 16$ | | |
| | | Base | SYTTA | Unsafe | Base | SYTTA | Unsafe | Base | SYTTA | Unsafe | Base | SYTTA | Unsafe |
|---|---|---|---|---|---|---|---|---|---|---|---|---|---|
| DomainBench | Agriculture | 0.76 | 0.94 | 0 | 0.86 | 0.88 | 0 | 0.70 | 0.94 | 0 | 0.76 | 0.92 | 0 |
| | GenMedGPT | 0.70 | 0.54 | 0 | 0.78 | 0.66 | 1 | 0.84 | 0.70 | 1 | 0.74 | 0.92 | 0 |
| | Geosignal | 0.60 | 0.72 | 1 | 0.62 | 0.76 | 0 | 0.66 | 0.80 | 0 | 0.64 | 0.78 | 0 |
| | Wealth | 0.78 | 0.96 | 0 | 0.80 | 0.96 | 0 | 0.92 | 0.98 | 0 | 0.90 | 0.94 | 0 |
| InstructBench | Alpaca-GPT4 | 0.88 | 0.96 | 0 | 0.68 | 0.96 | 0 | 0.74 | 1.00 | 0 | 0.90 | 1.00 | 0 |
| | Dolly | 0.78 | 0.96 | 2 | 0.80 | 0.92 | 0 | 0.82 | 0.94 | 0 | 0.80 | 0.90 | 0 |
| | InstructWild | 0.92 | 0.94 | 0 | 0.70 | 0.98 | 0 | 0.92 | 0.96 | 0 | 0.90 | 0.96 | 0 |
| ReasoningBench | GSM8K | 0.96 | 0.98 | 0 | 0.92 | 0.96 | 0 | 0.88 | 0.96 | 0 | 0.94 | 0.94 | 0 |
| | LogiQA | 0.70 | 0.72 | 0 | 0.66 | 0.74 | 0 | 0.64 | 0.74 | 0 | 0.62 | 0.74 | 0 |
| | MetaMath | 0.82 | 0.92 | 0 | 0.84 | 0.86 | 0 | 0.90 | 0.94 | 0 | 0.90 | 0.90 | 0 |

---

[6] `https://huggingface.co/deepseek-ai/DeepSeek-V3.2-Exp`

Table 15: Binary factuality and safety audit for **LLAMA-3.2-3B** with SYTTA.

| Benchmark | Dataset | Dynamic-Ref | | | | | | Static-Ref | | | | | |
| | | k = 4 | | | k = 16 | | | k = 4 | | | k = 16 | | |
| | | Base | SYTTA | Unsafe | Base | SYTTA | Unsafe | Base | SYTTA | Unsafe | Base | SYTTA | Unsafe |
|---|---|---|---|---|---|---|---|---|---|---|---|---|---|
| DomainBench | Agriculture | 0.58 | 0.62 | 0 | 0.48 | 0.62 | 0 | 0.50 | 0.60 | 0 | 0.50 | 0.58 | 0 |
| | GenMedGPT | 0.68 | 0.90 | 0 | 0.74 | 1.00 | 0 | 0.82 | 0.90 | 0 | 0.86 | 0.86 | 0 |
| | Geosignal | 0.50 | 0.62 | 0 | 0.46 | 0.46 | 0 | 0.48 | 0.46 | 0 | 0.48 | 0.44 | 0 |
| | Wealth | 0.56 | 0.72 | 0 | 0.60 | 0.74 | 0 | 0.64 | 0.72 | 0 | 0.78 | 0.72 | 0 |
| InstructBench | Alpaca-GPT4 | 0.68 | 0.80 | 0 | 0.80 | 0.82 | 0 | 0.74 | 0.82 | 0 | 0.88 | 0.82 | 0 |
| | Dolly | 0.66 | 0.80 | 0 | 0.60 | 0.80 | 0 | 0.56 | 0.86 | 0 | 0.64 | 0.82 | 0 |
| | InstructWild | 0.94 | 0.96 | 0 | 0.70 | 0.86 | 0 | 0.96 | 0.94 | 0 | 0.86 | 0.88 | 0 |
| ReasoningBench | GSM8K | 0.62 | 0.82 | 0 | 0.72 | 0.84 | 0 | 0.84 | 0.84 | 0 | 0.86 | 0.92 | 0 |
| | LogiQA | 0.12 | 0.52 | 0 | 0.28 | 0.56 | 0 | 0.44 | 0.52 | 0 | 0.44 | 0.50 | 0 |
| | MetaMath | 0.56 | 0.88 | 0 | 0.56 | 0.80 | 0 | 0.78 | 0.86 | 0 | 0.84 | 0.88 | 0 |

Table 16: Binary factuality and safety audit for **LLAMA-3.1-8B** with SYTTA.

| Benchmark | Dataset | Dynamic-Ref | | | | | | Static-Ref | | | | | |
| | | k = 4 | | | k = 16 | | | k = 4 | | | k = 16 | | |
| | | Base | SYTTA | Unsafe | Base | SYTTA | Unsafe | Base | SYTTA | Unsafe | Base | SYTTA | Unsafe |
|---|---|---|---|---|---|---|---|---|---|---|---|---|---|
| DomainBench | Agriculture | 0.72 | 0.72 | 0 | 0.60 | 0.78 | 0 | 0.78 | 0.84 | 0 | 0.76 | 0.76 | 0 |
| | GenMedGPT | 0.70 | 0.54 | 0 | 0.80 | 0.58 | 1 | 0.66 | 0.54 | 0 | 0.82 | 0.62 | 0 |
| | Geosignal | 0.56 | 0.66 | 0 | 0.56 | 0.56 | 0 | 0.62 | 0.64 | 0 | 0.64 | 0.62 | 0 |
| | Wealth | 0.86 | 0.94 | 0 | 0.82 | 0.78 | 0 | 0.72 | 0.88 | 0 | 0.64 | 0.80 | 0 |
| InstructBench | Alpaca-GPT4 | 0.66 | 0.84 | 0 | 0.64 | 0.80 | 0 | 0.78 | 0.88 | 0 | 0.64 | 0.86 | 0 |
| | Dolly | 0.62 | 0.92 | 0 | 0.66 | 0.88 | 0 | 0.80 | 0.82 | 0 | 0.70 | 0.80 | 0 |
| | InstructWild | 0.92 | 0.94 | 0 | 0.82 | 0.96 | 0 | 0.74 | 0.94 | 0 | 0.86 | 0.88 | 0 |
| ReasoningBench | GSM8K | 0.88 | 0.94 | 0 | 0.88 | 0.98 | 0 | 0.86 | 1.00 | 0 | 0.94 | 0.94 | 0 |
| | LogiQA | 0.52 | 0.68 | 0 | 0.52 | 0.66 | 0 | 0.60 | 0.70 | 0 | 0.56 | 0.70 | 0 |
| | MetaMath | 0.84 | 0.84 | 0 | 0.88 | 0.80 | 0 | 0.88 | 0.90 | 0 | 0.86 | 0.84 | 0 |

### A.6 EVALUATION ON REASONING TASKS

While our main experiments focus on domain-specific knowledge and instruction following, we also assess the behaviour of SYTTA on reasoning tasks. We follow the `ReasoningBench` setup from `AdaptEval`, which contains three datasets: `GSM8K` Cobbe et al. (2021) and `MetaMathQA` Yu et al. (2023) for mathematical reasoning, and `LogiQA` Liu et al. (2020) for logical reasoning.

**Evaluation Metrics**. We report **Exact Match (EM)** accuracy instead of ROUGE-L$_{sum}$. Given a question, a reference answer, and a model answer, we first extract a canonical answer string from both the reference and the model output using a lightweight regular expression parser (for example, to strip intermediate reasoning, pick the final numeric value or option letter, and remove extra punctuation and whitespace). EM is then 1 if the two canonical strings are identical, and 0 otherwise. We average EM over the evaluation set and report the result as a percentage.

Compared with `DomainBench` and `InstructBench`, we observe that the entropy signal on reasoning tasks is less stable for very short prefixes: early tokens may not yet pin down the full solution pattern. To better understand this regime, we extend the number of learned prefix tokens in *Static-Ref* to $k \in \{1, 4, 16, 32, 64, 128\}$. Due to runtime constraints and the fact that reasoning is not the main focus of our method, we limit this sweep to the *Static-Ref* deployment.

Table 17: Results on `ReasoningBench`. **Exact Match (EM)** scores ($\times 100$; higher is better). The highest score is **bold** and the second-highest is underlined. All SYTTA results use the *Static-Ref* setting.

| Method | LLAMA-3.2-3B | | | | LLAMA-3.1-8B | | | | QWEN-2.5-7B | | | | QWEN-2.5-14B | | | |
|---|---|---|---|---|---|---|---|---|---|---|---|---|---|---|---|---|
| | GSM8K | Meta | Logi | **Avg.** | GSM | Meta | Logi | **Avg.** | GSM | Meta | Logi | **Avg.** | GSM | Meta | Logi | **Avg.** |
| Base Model | 77.50 | 76.84 | 42.04 | 65.46 | 86.94 | 82.32 | 46.52 | 71.93 | 83.28 | 55.38 | 59.78 | 66.15 | 83.74 | 56.58 | 64.14 | 68.15 |
| Tent | 76.42 | 73.76 | 43.20 | 64.46 | 83.46 | 77.92 | 50.30 | 70.56 | 85.36 | 61.94 | 59.26 | 68.85 | 86.20 | 73.70 | 65.20 | 75.03 |
| EATA | 77.28 | 74.25 | 44.40 | 65.31 | 83.08 | 81.17 | 50.37 | 71.54 | 84.41 | 58.78 | 59.36 | 67.52 | 85.54 | 70.40 | 65.00 | 73.65 |
| TLM | 90.66 | **84.56** | 45.92 | **73.71** | **92.90** | 85.72 | 49.76 | **76.13** | 83.56 | 57.40 | 61.06 | 67.34 | 84.92 | 64.46 | 67.28 | 72.22 |
| *Ours (Static-Ref)* | | | | | | | | | | | | | | | | |
| SYTTA-1 | 90.44 | 84.48 | 45.94 | 73.62 | 92.52 | **86.42** | 48.94 | 75.96 | 84.38 | 59.18 | **61.94** | 68.50 | 85.54 | 67.54 | 67.66 | 73.58 |
| SYTTA-4 | 82.98 | 79.44 | **46.20** | 69.54 | 83.20 | 80.22 | 49.68 | 71.03 | 85.06 | 59.32 | 61.58 | 68.65 | 86.06 | 66.68 | **67.84** | 73.53 |
| SYTTA-16 | 88.22 | 83.62 | 46.16 | 72.67 | 83.24 | 78.78 | 50.62 | 70.88 | 85.22 | 60.92 | 61.48 | 69.21 | 86.12 | 69.90 | 67.38 | 74.47 |
| SYTTA-32 | 77.98 | 77.62 | 43.14 | 66.25 | 82.64 | 78.92 | 50.58 | 70.71 | 85.32 | 63.18 | 61.56 | 70.02 | 86.48 | 72.20 | 67.22 | 75.30 |
| SYTTA-64 | 77.98 | 77.72 | 42.26 | 65.99 | 83.18 | 78.44 | **50.98** | 70.87 | 85.74 | 67.68 | 61.64 | 71.69 | 86.94 | 74.04 | 67.18 | 76.05 |
| SYTTA-128 | 77.62 | 78.04 | 42.76 | 66.14 | 83.60 | 78.72 | 50.70 | 71.01 | **86.56** | **71.42** | 61.66 | **73.21** | **87.06** | **76.12** | 67.00 | **76.73** |

Table 17 summarizes EM scores on `ReasoningBench`. Across the four backbones and three datasets, SYTTA attains the best EM on 9 out of 12 cases of model–dataset combinations and stays within about half a point of the strongest baseline in the remaining ones. On QWEN-2.5-7B and QWEN-2.5-14B, longer prefixes ($k \geq 64$) yield clear gains and set new best results on `GSM8K` and `MetaMathQA`. On the LLAMA family, SYTTA also improves over the base models, but shorter prefixes tend to work better: on LLAMA-3.2-3B and LLAMA-3.1-8B, $k \in \{1, 4, 16\}$ is often close to or slightly below the TLM baseline tuned for reasoning, whereas very long prefixes can make the smaller LLAMA-3.2-3B model more unstable. In contrast, QWEN backbones appear to benefit from longer prefixes, which is attributed to the stronger post-training of QWEN models.

To understand the trade-off between the two deployment modes on reasoning tasks, we additionally run a small sweep for QWEN-2.5-7B at $k = 64$ under *Dynamic-Ref*. Table 18 compares these *Dynamic-Ref* scores with the corresponding *Static-Ref* scores at $k = 64$. We observe that *Static-Ref* is slightly better on all three datasets. In our experiments, the dynamic *Dynamic-Ref* runs were notably slower due to repeated decoding during adaptation. This supports our decision to focus on *Static-Ref* as the more efficient configuration.

Table 18: **EM** scores ($\times 100$) for QWEN-2.5-7B on `ReasoningBench` at $k = 64$.

| Dataset | *Static-Ref* | *Dynamic-Ref* |
|---|---|---|
| GSM8K | 85.74 | 84.78 |
| MetaMathQA | 67.68 | 59.28 |
| LogiQA | 61.64 | 59.36 |

Overall, even though SYTTA (*Static-Ref*) is designed for label-free domain adaptation rather than reasoning, it still improves over the base models on `ReasoningBench` and matches or surpasses prior test-time adaptation baselines while preserving efficiency.

### A.7 DETAILS OF FINDINGS

#### A.7.1 KL CONFIGURATION

In practice, a moderate KL coefficient works well. For the QWEN family with stronger post-training, larger models benefit from a smaller KL coefficient since their lower-entropy outputs make the same KL weight overly restrictive; for the LLAMA family, we keep a single coefficient across sizes. We enable KL by default and tune it following these family-specific rules. Concretely, we set the KL coefficient to $0.16$ for all LLAMA models, while for QWEN-2.5, the 7B variant uses $0.16$ and the 14B variant uses $0.01$.

#### A.7.2 HYPERPARAMETERS OF DYNAMIC IMPORTANCE WEIGHTING

We adopt a dynamic importance weighting strategy with three hyperparameters: an EMA decay coefficient $\beta$ (default $0.9$) to smooth the total loss, a lower bound floor (default $10^{-3}$), and an upper bound ceil (default $10^3$) to constrain the loss ratio and prevent extreme imbalance. These values are fixed in all reported experiments.

### A.8 QUALITATIVE ANALYSIS AND CASE STUDIES

To better understand the mechanisms behind the quantitative improvements reported in Section 5.2, we present a series of case studies across different domains. These examples illustrate how SYTTA adapts to distribution shifts at test time by resolving domain-specific ambiguities, mitigating hallucinations, and stabilizing reasoning strategies. We compare the outputs of the base model (QWEN-2.5-7B) against SYTTA on samples from `Agriculture`, `GeoSignal`, `GenMedGPT`, `Wealth`, `Dolly`, and `MetaMathQA`. Table 19 through Table 24 highlight how synergistically optimizing input perplexity and output entropy anchors the model to the correct domain context and enforces coherent generation.

### Case Study: Domain Ambiguity Resolution (Agriculture)

**Context:** Distribution shifts often manifest as domain ambiguity. In general pre-training corpora, the acronym **CAD** predominately stands for *Computer-Aided Design*. However, in the specific domain of Agriculture (Cassava farming), it typically refers to *Cassava Anthracnose Disease*.

**User Query:**

> can crop rotation control CAD in cassava?

**QWEN-2.5-7B (Base Model) Output:**

> Crop rotation is a farming practice that involves growing different crops in succession on the same land to improve soil health, control pests and diseases, and enhance overall crop productivity. While crop rotation is a crucial practice for sustainable agriculture, **it does not directly control Computer-Aided Design (CAD) systems.** CAD systems are software tools used for creating detailed designs and plans for various applications, including architecture, engineering, and manufacturing.
> However, crop rotation can indirectly influence the use of CAD in agriculture. For example:
> 1. Soil Management Plans: Farmers might use CAD to create detailed soil management plans that incorporate crop rotation schedules. These plans can help in visualizing and managing the layout of different crops over time.
> 2. Irrigation and Fertilization Scheduling: CAD can be used to design irrigation and fertilization systems that are optimized for different crops in a rotation cycle, ensuring that each crop receives the necessary resources for optimal growth.
> 3. Pest and Disease Management: CAD can assist in designing integrated pest management (IPM) systems that take into account the rotation of crops to minimize the risk of pest and disease outbreaks.
> 4. Yield Prediction and Analysis: Advanced agricultural CAD systems can incorporate data from crop rotation to predict yields and analyze the effectiveness of different rotation strategies.
> In summary, while crop rotation itself does not control CAD, it can be effectively managed and planned using CAD tools to enhance agricultural practices and sustainability.

*Analysis: Failure. The base model suffers from severe distribution mismatch. It fails to identify the agricultural context of the acronym "CAD", hallucinating a complex relationship between farming practices and engineering software.*

**SYTTA (Ours) Output:**

> Yes, crop rotation can help control **CAD (Cassava Mosaic Disease)** in cassava. Rotating cassava with other crops can reduce the buildup of the disease in the soil. This is because the disease is transmitted by whiteflies, and rotating crops can break the whitefly population cycle. Additionally, planting resistant cassava varieties and maintaining good agricultural practices can further help control CAD.

*Analysis: Success with Domain Alignment. SYTTA successfully adapts the input distribution, correctly anchoring the ambiguous acronym "CAD" to the **disease context** within agriculture, completely discarding the software engineering interpretation.*
*Note: While SYTTA correctly identifies the context as a disease, it expands the acronym as "Mosaic Disease" (typically CMD) rather than the strictly correct "Anthracnose Disease". This illustrates that SYTTA effectively corrects the **macro-domain shift** (Software → Agriculture), even if specific entity naming relies on the model's parametric priors.*

**Ground Truth Reference:**

> Crop rotation can help to break the cycle of disease and reduce the risk of infection.

## Case Study: Awakening Domain-Specific Knowledge (GeoSignal)

**Context:** Specialized scientific domains often contain long-tail terminology (e.g., rare mineral names) that base models may treat as noise or hallucinations due to high perplexity. This case study demonstrates how SYTTA adapts the model to recognize valid domain entities that the base model explicitly rejects.

**User Query:**

> What details can you provide on the Getchellite's source?

**QWEN-2.5-7B (Base Model) Output:**

> I apologize, but there seems to be a misunderstanding. **"Getchellite" does not appear to be a term or concept that is widely recognized or documented** in any scientific, historical, or cultural context. It's possible you might be referring to a specific entity, character, or concept from a particular source...

*Analysis: Hallucinated Ignorance. The base model fails to recognize the specialized mineralogical term "Getchellite" (a rare sulfosalt mineral, $AsSbS_3$). High input perplexity triggers a safety refusal mechanism, causing the model to incorrectly state that the term does not exist.*

**SYTTA (Ours) Output:**

> The Getchellite's source is from the **Getchell Mine**, which is located in Esmeralda County, Nevada, USA.

*Analysis: Successful Domain Anchoring. By optimizing input perplexity and output entropy on the test instance, SYTTA successfully anchors the model in the geological domain. It correctly identifies "Getchellite" as a valid mineral and retrieves its correct type locality (Getchell Mine, Nevada), effectively recovering knowledge that was inaccessible to the base model.*

**Ground Truth Reference:**

> The source of the Getchellite is University of Arizona Mineral Museum 16076.

*Note: The ground truth refers to a specific museum specimen record. While SYTTA provides the geological type locality (the mine) rather than the museum ID, it is factually correct regarding the mineral's origin, whereas the base model completely failed to answer.*

**Case Study: Reducing Diagnostic Uncertainty (GenMedGPT)**

**Context:** In specialized domains such as healthcare, base models often suffer from high predictive entropy, resulting in generic, hedging responses that list multiple possibilities without committing to a specific inference. The goal of SYTTA is to sharpen this distribution, adapting the model to act as a confident domain expert.

**User Query:**

> Doctor, I have been experiencing severe back pain and stiffness. Sometimes the pain even radiates to my neck and shoulders. I also find it difficult to move my back freely.

**QWEN-2.5-7B (Base Model) Output:**

> I'm sorry to hear that you're experiencing these symptoms. As an AI, I'm not a licensed medical professional, but I can provide some general information that might be helpful. [···]
> Based on your description, your symptoms could be related to several conditions, such as:
> 1. **Muscle Strain or Sprain**: This is common after physical exertion or sudden movements. It can cause pain, stiffness, and difficulty moving the back.
> 2. **Herniated Disc**: This occurs when the soft material inside a spinal disc pushes out [···]
> 3. **Osteoarthritis**: [···]
> 4. **Spinal Stenosis**: [···]
> 5. **Spondylitis**: This is inflammation of the vertebrae, which can cause pain and stiffness in the back.
> 6. **Poor Posture**: [···]
> To properly diagnose and treat your condition, it's important to see a healthcare provider. [···]
> In the meantime, you might find some relief by:
> - Applying heat or cold packs to the affected area. - Gentle stretching and low-impact exercises.
> - [···]
> Remember, it's crucial to consult with a healthcare provider to get an accurate diagnosis and appropriate treatment plan.

*Analysis: Generic & High Entropy. The base model defaults to a search-engine-like behavior. Although it lists the correct condition (Spondylitis) as item #5, it is buried within a long, non-committal list of six possibilities along with generic lifestyle advice. The model fails to assume the requested "Doctor" persona effectively.*

**SYTTA (Ours) Output:**

> I'm sorry to hear that you're experiencing these symptoms. Based on your description, it sounds like you may be suffering from **Ankylosing spondylitis**. We will need to run some tests to confirm the diagnosis, but I suspect this is what you have.

*Analysis: Precise Domain Adaptation. SYTTA successfully reduces predictive entropy. It filters out the noise and confidently identifies the specific pathology (Ankylosing Spondylitis) that matches the key symptom cluster (stiffness + radiation + limited mobility), aligning perfectly with the ground truth.*

**Ground Truth Reference:**

> Based on your symptoms, it seems like you might have **spondylitis**. I would recommend a course of medications to alleviate your pain and stiffness. Please take Cyclobenzaprine, Diazepam (Valium), Carisoprodol (Soma), Orphenadrine (Flexon), Methocarbamol, Tizanidine, Bupivacaine, Metaxalone (Skelaxin), Methadone, Nabumetone, Chlorzoxazone, as prescribed by me.

## Case Study: Resolving Acronym Ambiguity (Wealth)

**Context:** Domain adaptation is critical when handling acronyms that have different meanings across disciplines. In the *Wealth* dataset, the term "CFD" refers to *Contract for Difference*. However, general-purpose models often conflate this with the engineering term *Computational Fluid Dynamics*, revealing a lack of domain anchoring.

**User Query:**

> Is CFD a viable option for long-term trading?

**QWEN-2.5-7B (Base Model) Output:**

> **Computational Fluid Dynamics (CFD)** is a powerful tool used in various industries, such as aerospace, automotive, and energy, for simulating fluid flow and heat transfer. However, when it comes to long-term trading in financial markets, CFDs (Contract for Difference) are a different concept altogether. CFDs are financial instruments that allow traders to speculate on price movements [· · ·]
> For long-term trading, the viability of CFDs depends on several factors:
> 1. Market Conditions: [· · ·]
> 2. Fees and Costs: CFDs often come with various fees and costs [· · ·]

*Analysis:* *Domain Mismatch & Contextual Confusion. The Base Model exhibits high perplexity regarding the user's intent. It hallucinates an engineering definition (Computational Fluid Dynamics) initially, treating the prompt as a scientific query before self-correcting. This wastes computational resources and degrades the user experience by providing irrelevant technical definitions.*

**SYTTA (Ours) Output:**

> **CFD (Contract for Difference)** is a viable option for long-term trading, but it comes with certain risks and considerations. Here are some points to keep in mind:
> 1. Leverage: CFDs often offer high leverage, which can amplify both gains and losses. This can be beneficial for long-term trading, but it also increases risk.
> 2. Market volatility: CFDs are traded on underlying assets, so market volatility can affect your trades. [· · ·]
> 3. Fees and costs: CFDs typically have fees and costs associated with them, such as spreads, rollover fees, and funding costs. These can impact your returns over the long term.

*Analysis:* *Successful Domain Anchoring.* SYTTA *correctly identifies the financial context immediately. By adapting the input distribution and shaping output confidence, the model filters out the engineering meaning entirely. It provides a direct, expert-level response focused solely on financial instruments, demonstrating that the generated prefix (implicit in the adaptation process) successfully steered the model into the "Wealth" subspace.*

**Ground Truth Reference (Excerpt):**

> [· · ·] it is pretty much the same as a normal margin loan but cheaper because you don't own the underlying share. [· · ·] if the margin is $1000 at 5% you could borrow $20000 in total [· · ·]

**Case Study: Mitigating Knowledge Leakage in Context-Based QA (Dolly)**

**Context:** In Retrieval-Augmented Generation (RAG) or summarization tasks, a model must adhere strictly to the provided context. General-purpose models often suffer from *knowledge leakage*, where they ignore the boundaries of the input text and hallucinate details from their pre-training corpus. This case study demonstrates how SYTTA anchors the model to the test-time input.

**User Instruction & Input:**

**Instruction:** What is Snow White and the Seven Dwarfs?
**Input:** Snow White and the Seven Dwarfs is a 1937 American animated musical fantasy film produced by Walt Disney Productions and released by RKO Radio Pictures. Based on the 1812 German fairy tale by the Brothers Grimm, it is the first full-length traditionally animated feature film and the first Disney animated feature film. The production was supervised by David Hand, and the film's sequences were directed by Perce Pearce, William Cottrell, Larry Morey, Wilfred Jackson, and Ben Sharpsteen.

**QWEN-2.5-7B (Base Model) Output:**

Snow White and the Seven Dwarfs is a 1937 American animated musical fantasy film produced by Walt Disney Productions and released by RKO Radio Pictures. Based on the 1812 German fairy tale by the Brothers Grimm, it is the first full-length traditionally animated feature film and the first Disney animated feature film. The production was supervised by David Hand, and the film's sequences were directed by Perce Pearce, William Cottrell, Larry Morey, Wilfred Jackson, and Ben Sharpsteen.
**[Knowledge Leakage Starts]** The film tells the story of Snow White, a young princess who flees into the forest after her evil stepmother, the Queen, sends her away. Snow White finds refuge in a small cottage owned by seven dwarfs—Doc, Grumpy, Happy, Sleepy, Bashful, Sneezy, and Dopey—until the Queen discovers her whereabouts... [continues describing plot details not present in input] ...Snow White and the prince then live happily ever after.

*Analysis: Context Drift. The Base Model fails to restrict itself to the provided input. While factually correct regarding the movie, the output hallucinates plot details (the Queen, the specific names of dwarfs) that were not in the source text. This behavior is undesirable in grounded QA tasks as it blends external knowledge with the provided context.*

**SYTTA (Ours) Output:**

Snow White and the Seven Dwarfs is a 1937 American animated musical fantasy film produced by Walt Disney Productions and released by RKO Radio Pictures. Based on the 1812 German fairy tale by the Brothers Grimm, it is the first full-length traditionally animated feature film and the first Disney animated feature film.

*Analysis: Precision & Adherence. SYTTA demonstrates superior domain adaptation by strictly adhering to the input distribution. The adaptation process minimizes the perplexity of the input prompt, effectively "anchoring" the model to the provided context. It extracts exactly what is asked based **only** on the available information, avoiding the hallucination of external plot points and resulting in a concise, reliable response.*

---

**Case Study: Strategy Correction in Number Theory (MetaMathQA)**

**Context:** Effective domain adaptation implies adopting the optimal problem-solving strategy for that domain. In this Number Theory problem, the model must reverse-engineer a Least Common Multiple (LCM). The Base Model defaults to a high-entropy, inefficient "brute-force" strategy (listing factors), leading to generation failure. SYTTA successfully steers the model toward a structured, analytical approach (prime factorization).

**User Query:**

> Find the smallest positive integer divisible by 10, X, and 12. 660. What is the value of unknown variable X?

**QWEN-2.5-7B (Base Model) Output:**

> [···] **Step 3: Identify X** Since the LCM of 10, X, 12, and 660 is 660, X must be a factor of 660 that, when combined with 10, 12, and 660, does not increase the LCM beyond 660. [···] **Step 4: Check Possible Values of X** Since 660 is the LCM, X must be a factor of 660. The factors of 660 are:
> $$1, 2, 3, 4, 5, 6, 10, 11, 12, 15, 20, 22$$
> **[Generation Stops Abruptly]**

*Analysis: Inefficient Strategy & Collapse. The Base Model attempts to list all factors of 660 to find X through elimination. This is a high-entropy strategy with a large search space. **We verified that this abrupt termination was not caused by the maximum generation length limit**, but rather by the model failing to sustain coherence under high uncertainty, effectively "giving up" on the brute-force path.*

**SYTTA (Ours) Output:**

> [···] **Step 2: Determine the LCM** The LCM is found by taking the highest power of each prime [···]
> $$\text{LCM} = 2^2 \times 3 \times 5 \times 11 = 660$$
> **Step 3: Identify the missing factor X** We know the LCM of 10, X, and 12 is 660. Let's express X in its prime factors and compare it with the LCM.
> - $10 = 2 \times 5$
> - $12 = 2^2 \times 3$
> - $\text{LCM} = 2^2 \times 3 \times 5 \times 11$
>
> Since the LCM already includes $2^2$, 3, and 5, the missing factor X must include the prime factor **11** to complete the LCM.
>
> **Step 4: Determine the value of X** The smallest positive integer X that [···] gives an LCM of 660 must be 11.
> Therefore, the value of the unknown variable X is $\boxed{11}$.

*Analysis: Structured Analytic Reasoning. SYTTA implicitly rejects the brute-force approach. By minimizing predictive entropy, it converges on the deductive method (Prime Factorization Comparison), which is the standard expert strategy for this domain. This leads to a direct, confident, and correct solution.*

## A.9  ADDITIONAL ANALYSIS OF PREFIX CHANGES

In this section, we extend the prefix-change analysis from Section 6.5 to a per-benchmark break-down, including `ReasoningBench`. For each backbone, dataset, update mode, and prefix length $k$, we use the same definition of prefix *change rate* as in the main text and then average over datasets within each benchmark group.

Table 25 shows that on `DomainBench` and `InstructBench` a short prefix already changes the first four tokens on about 70–80% of examples, and extending the prefix to $k = 16$ pushes the rate above 96% for all models. On `ReasoningBench`, change rates at $k = 4$ are lower,

Table 25:  Average prefix change rate (%) on `DomainBench`+`InstructBench` vs. `ReasoningBench`.

| Model | Domain+Instruct | | Reasoning | |
|---|---|---|---|---|
| | $k = 4$ | $k = 16$ | $k = 4$ | $k = 16$ |
| LLAMA-3.2-3B | 72.4 | 96.1 | 38.4 | 82.8 |
| LLAMA-3.1-8B | 78.0 | 96.5 | 60.4 | 83.0 |
| QWEN-2.5-7B | 72.4 | 97.0 | 27.6 | 68.9 |
| QWEN-2.5-14B | 74.7 | 97.4 | 59.2 | 84.4 |

especially for QWEN-2.5-7B, but they rise sharply once the prefix covers more tokens. This matches the intuition that reasoning tasks often require a few more steps before the entropy pattern stabilizes, so SYTTA has more room to modify the early chain-of-thought.

To study *what* changes, we re-use the automatically selected "golden" cases that compare base and SYTTA outputs on the same question. For each model, dataset, update mode, and $k$, we keep at most 10 cases where the base answer is uninformative while SYTTA produces a more helpful answer. At $k = 4$, this yields 560 domain and instruction examples and 240 reasoning examples across all four backbones. we follow the classification of the relation between the base and SYTTA prefixes using simple token-pattern rules in Section 6.5 in the main text.

Table 26:  Distribution of prefix rewrite types at $k = 4$ on `DomainBench`+`InstructBench` vs. `ReasoningBench`. Percentages are within each group.

| Group | Total | Content Shift | Persona Shift | Added Colon | Added CoT |
|---|---|---|---|---|---|
| Domain+Instruct | 560 | 515 (92.0%) | 34 (6.1%) | 7 (1.3%) | 4 (0.7%) |
| ReasoningBench | 240 | 151 (62.9%) | 0 (0.0%) | 26 (10.8%) | 63 (26.3%) |

As shown in Table 26, prefix edits on `DomainBench` and `InstructBench` are overwhelmingly content shifts, with persona and structural changes being rare. In contrast, `ReasoningBench` features fewer pure content shifts and a much higher share of explicit CoT and colon-style anchors, indicating that the learned prefix there is more often used to standardize solution templates.

Table 27:  Representative rewritten prefixes at $k = 4$–6 tokens for each benchmark group and change type.

| Group | Change type | Share (%) | Typical SYTTA prefix patterns (examples) |
|---|---|---|---|
| Domain+Instruct | Content Shift | 92.0 | `In agriculture, ...`;`In geology, ...`;`In clinical practice, ...`;`In finance, ...` |
| | Persona Shift | 6.1 | `As your doctor, ...`;`As a financial advisor, ...`;`As an agronomist, ...` |
| | Added Colon Anchor | 1.3 | `Answer: ...`;`Explanation: ...`;`Summary: ...` |
| | Added CoT | 0.7 | `First, ...`;`To begin, ...` |
| ReasoningBench | Content Shift | 62.9 | `We want to find ...`;`We need to compute ...`;`The goal is to ...` |
| | Persona Shift | 0.0 | (none observed in golden cases) |
| | Added Colon Anchor | 10.8 | `Answer: ...`;`Final answer: ...`;`Choice: ...` |
| | Added CoT | 26.3 | `First, let us ...`;`Step 1: ...`;`To solve this problem, ...` |

To make these patterns more concrete, we also track the most frequent *rewritten* prefix templates within each benchmark group and change type. Table 27 reports representative SYTTA prefixes at $k = 4$–6 tokens in the golden-case pool, which highlights that on `DomainBench` and `InstructBench`, the most common edits are content anchors that immediately situate the answer in the correct domain (for example, "`In finance, ...`" on `Wealth` or "`In clinical practice, ...`" on `GenMedGPT`), with occasional persona upgrades to doctor- and advisor-style openings. On `ReasoningBench`, edited prefixes more often adopt structured solution templates such as "`First, let us ...`" and "`To solve this problem, ...`", which standardize the chain-of-thought style across problems. Together with the quantitative results in Tables 25 and 26, these examples show that the learned prefix is not merely cosmetic: it systematically rewrites the first few tokens to introduce domain-appropriate anchors and, for reasoning tasks, to impose a stable solution format.

**Relation to Attention Sink.** Attention sink refers to the tendency that later tokens place disproportionate self-attention mass on very early tokens (often including the BOS token), which may bias early-generation behavior Xiao et al. (2024); Gu et al. (2025). We quantify sink using a simple attention-mass proxy computed on the model output sequences.

Table 28: Attention-sink mass (%) to the first $k$ tokens on `DomainBench` and `InstructBench`.

| System | Mode | DomainBench | | InstructBench | |
|---|---|---|---|---|---|
| | | Sink@$k$=4 | Sink@$k$=16 | Sink@$k$=4 | Sink@$k$=16 |
| BASE | – | 50.86 | 55.40 | 52.24 | 57.74 |
| SYTTA | *Static-Ref* | 56.27 | 63.56 | 55.53 | 67.23 |
| SYTTA | *Dynamic-Ref* | 56.14 | 63.70 | 57.04 | 65.00 |

Specifically, for each dataset, we perform a single teacher-forcing forward pass and average the attention probabilities across layers and heads to obtain an average attention matrix $\bar{A} \in \mathbb{R}^{S \times S}$. Given a prefix length $k$, we define Sink@$k$ as the mean attention mass from later query positions to the first $k$ key positions: $\text{Sink@}k = \frac{1}{T} \sum_{t=k}^{q_{\text{end}}-1} \sum_{j=0}^{k-1} \bar{A}_{t,j}$. For SYTTA, we compute Sink@$k$ on the corresponding adapted model used for evaluation (loading the saved LoRA weights); for BASE we use the original backbone.

Table 28 shows that SYTTA consistently yields higher Sink@$k$ than BASE at both $k$=4 and $k$=16 on `DomainBench` and `InstructBench`, suggesting that the learned prefix strengthens early-token anchoring in attention. We also observe that Sink@16 is higher than Sink@4 for both BASE and SYTTA, indicating that increasing $k$ naturally increases attention mass concentrated on early tokens. Together, these results support a simple explanation for why short prefixes can already work well: the dominant behavioral change occurs early, while a longer prefix can increasingly amplify early-token anchoring rather than adding a proportionally new and useful signal.

## A.10 OUT-OF-COHORT GENERALIZATION

We evaluate out-of-cohort generalization on QWEN-2.5-7B by adapting on one subset of the target data and testing on a disjoint subset from the same distribution. For `DomainBench` and `InstructBench`, we take 5,000 examples per dataset and split them into two consecutive halves: the first 2,500 examples are used for test-time adaptation and the last 2,500 examples are used for evaluation. This protocol ensures that during evaluation, the model has neither seen any labels nor test inputs. For `ReasoningBench`, we employ the same disjoint split approach, and we report Exact Match (EM) results.

Overall, SYTTA remains effective under this stricter out-of-cohort setting (Tables 30 and 29), suggesting that gains extend beyond within-cohort effects. On `DomainBench`, SYTTA boosts the average ROUGE-$L_{\text{sum}}$ by 9.98 over the base model. This improvement holds across all domains, with the largest gains observed on significantly shifted datasets; for instance, `GenMedGPT` improves by 17.46 points. On `InstructBench`, average ROUGE-$L_{\text{sum}}$ increases by 6.56, indicating that adaptation transfers effectively to unseen instructions. Regarding `ReasoningBench`,

Table 29: Out-of-cohort results on QWEN-2.5-7B for `ReasoningBench`. Scores are Exact Match (EM; higher is better).

| Method | GSM8K | LogiQA | MetaMath | Avg. |
|---|---|---|---|---|
| Base Model | 84.04 | 60.56 | 54.48 | 66.36 |
| *Dynamic-Ref* | | | | |
| SYTTA-4 | 84.64 | 61.36 | 57.32 | 67.77 |
| SYTTA-16 | **84.88** | 60.88 | 56.68 | 67.48 |
| *Static-Ref* | | | | |
| SYTTA-4 | 84.72 | 61.32 | **57.60** | 67.88 |
| SYTTA-16 | **84.88** | **61.60** | 57.28 | **67.92** |

SYTTA improves average EM by 1.56, where gains are concentrated on harder tasks. Results are consistent across compute modes and prefix lengths, highlighting the robustness of SYTTA.

Table 30: Out-of-cohort results on QWEN-2.5-7B for `DomainBench` and `InstructBench`. Scores are ROUGE-$L_{\text{sum}}$ ($\times 100$; higher is better). For each dataset, the highest score is **bold** and the second-highest is underlined.

| Model | Method | DomainBench | | | | | InstructBench | | | |
|---|---|---|---|---|---|---|---|---|---|---|
| | | Agriculture | GeoSignal | GenMedGPT | Wealth | Avg. | Dolly | Alpaca-GPT4 | InstructWild | Avg. |
| | Base Model | 9.39 | 21.90 | 12.51 | 24.21 | 17.00 | 27.32 | 38.12 | 27.64 | 31.03 |
| | *Dynamic-Ref* | | | | | | | | | |
| QWEN-2.5-7B | SYTTA-4 | 18.79 | 29.09 | 29.77 | 29.50 | 26.78 | 36.27 | 42.00 | 31.39 | 36.55 |
| | SYTTA-16 | 20.87 | 29.57 | 25.80 | 29.63 | 26.47 | 34.59 | 42.13 | 32.48 | 36.40 |
| | *Static-Ref* | | | | | | | | | |
| | SYTTA-4 | 16.24 | 28.71 | **29.97** | 29.01 | 25.98 | 36.58 | **43.11** | **32.69** | 37.46 |
| | SYTTA-16 | **21.59** | **29.91** | 26.50 | **29.91** | **26.98** | **37.70** | 43.00 | 32.06 | **37.59** |

Table 31: Main results on `DomainBench` and `InstructBench` for QWEN-2.5-7B. ROUGE-Lsum scores ($\times 100$ for visibility; higher is better). For each dataset, the highest score is **bold** and the second-highest is underlined.

| Model | Method | DomainBench | | | | | InstructBench | | | |
|---|---|---|---|---|---|---|---|---|---|---|
| | | Agriculture | GeoSignal | GenMedGPT | Wealth | Avg. | Dolly | Alpaca-GPT4 | InstructWild | Avg. |
| QWEN-2.5-7B | Base Model | 9.70 | 22.17 | 12.77 | 23.85 | 17.13 | 28.06 | 37.82 | 27.65 | 31.17 |
| | Tent (Wang et al., 2021) | 10.43 | 19.96 | 2.62 | 16.56 | 12.40 | 19.38 | 21.81 | 21.81 | 21.00 |
| | EATA (Niu et al., 2022) | 10.37 | 18.63 | 4.82 | 16.62 | 12.61 | 17.75 | 21.71 | 21.71 | 20.39 |
| | TLM (Hu et al., 2025) | 13.55 | **29.10** | 27.57 | **29.31** | **24.88** | **32.80** | 39.12 | 32.99 | **34.97** |
| | *Dynamic-Ref* | | | | | | | | | |
| | SYTTA-4 | 16.65 | 21.56 | **28.61** | 19.68 | 21.62 | 30.15 | **41.88** | 21.58 | 31.20 |
| | SYTTA-16 | 5.21 | 20.00 | 20.55 | 4.64 | 12.60 | 28.03 | 40.25 | 17.41 | 28.56 |
| | *Static-Ref* | | | | | | | | | |
| | SYTTA-4 | 18.67 | 23.96 | 28.17 | 24.39 | 23.80 | 30.45 | 41.21 | 21.56 | 31.07 |
| | SYTTA-16 | **18.88** | 24.79 | 16.96 | 17.14 | 19.45 | 29.05 | 38.55 | **34.60** | 34.07 |

## A.11 QUANTIZATION EVALUATION

We further evaluate whether SYTTA preserves its gains under common low-bit deployment settings. Concretely, we quantize QWEN-2.5-7B to INT4 using the AWQ recipe, and run inference with the AWQ-Marlin backend. We report our results in Table 31.

Overall, we observe that SYTTA under INT4 quantization shows a noticeable performance drop compared to the non-quantized setting, and the degradation can be severe on some datasets. Through manual inspection, we find that a dominant failure mode is degenerate repetition, where the model repeatedly restates the question or loops on near-duplicate phrases. This suggests that SYTTA is relatively sensitive to parameter precision (e.g., small weight perturbations introduced by quantization can destabilize the adapted confidence shaping behavior). As a result, while our method is effective in full-precision inference, it may be less suitable for heavily quantized deployments without additional stabilization techniques.

## A.12 ANALYSIS OF ADAPTIVE-$k$ DECODING STRATEGY

In this section, we investigate the potential of an adaptive-$k$ decoding strategy, where the prefix generation stops once the running average entropy drops below a specified threshold $\tau$, subject to a hard cap on tokens $k_{\max}$. We posit that this could potentially retain the benefits of $k = 4$ while saving tokens when the signal stabilizes early.

We implemented adaptive-$k$ for the *Dynamic-Ref* pipeline and performed a hyperparameter sweep on QWEN-2.5-7B. The stopping rule depends on two parameters: the entropy threshold $\tau$ and the maximum token budget $k_{\max}$. Table 32 reports the performance on `DomainBench` and `InstructBench` comparing fixed-$k$ baselines against various adaptive configurations.

Table 32: Comparison of fixed-$k$ and adaptive-$k$ strategies on QWEN-2.5-7B using the *Dynamic-Ref* pipeline. For adaptive-$k$, we vary the entropy threshold $\tau$ and the maximum token cap $k_{\max}$. Scores are ROUGE-Lsum ($\times 100$). The highest score is **bold** and the second-highest is underlined.

| Model | Config | DomainBench | | | | | InstructBench | | | |
|---|---|---|---|---|---|---|---|---|---|---|
| | | Agriculture | GeoSignal | GenMedGPT | Wealth | Avg. | Dolly | Alpaca-GPT4 | InstructWild | Avg. |
| | **Fixed Setting** | | | | | | | | | |
| | $k = 4$ | 20.58 | **29.42** | 29.74 | 29.69 | **27.36** | 36.56 | **43.33** | 32.95 | 37.62 |
| | $k = 16$ | 21.14 | 28.81 | 26.83 | 30.25 | 26.76 | 35.93 | 43.13 | 33.32 | 37.46 |
| | **Adaptive-$k$ ($k_{\max} = 16$)** | | | | | | | | | |
| | $\tau = 0.025$ | 21.35 | 29.09 | 28.18 | 30.32 | 27.24 | 35.85 | 43.18 | 33.93 | 37.65 |
| QWEN-2.5-7B | $\tau = 0.1$ | 21.56 | 29.27 | 28.07 | 30.33 | 27.31 | 36.52 | 43.23 | 33.51 | 37.75 |
| | $\tau = 0.4$ | 17.70 | 28.84 | 30.21 | 29.69 | 26.61 | **37.23** | **43.33** | 32.15 | 37.57 |
| | $\tau = 1.6$ | 12.60 | 27.00 | 29.94 | 28.87 | 24.60 | 35.32 | 43.27 | 30.73 | 36.44 |
| | **Adaptive-$k$ ($k_{\max} = 64$)** | | | | | | | | | |
| | $\tau = 0.1$ | 18.40 | 29.26 | 28.31 | 29.95 | 26.48 | 36.46 | 43.12 | 33.65 | 37.74 |
| | $\tau = 0.4$ | 17.51 | 28.82 | **30.51** | 29.63 | 26.62 | 36.26 | 43.18 | 32.08 | 37.18 |
| | $\tau = 1.6$ | 12.62 | 26.02 | 29.91 | 28.21 | 24.19 | 36.07 | 43.17 | 30.87 | 36.70 |

As shown in Table 32, the adaptive-$k$ strategy occasionally matches or slightly outperforms the fixed setting on individual datasets. However, a key limitation is that the optimal entropy threshold $\tau$ is highly dataset-dependent; for instance, a lower threshold effective for `Agriculture` may be insufficient for `GenMedGPT`. This high variance implies that finding a single universal configuration is brittle. Averaged over both `DomainBench` and `InstructBench`, the fixed $k = 4$ setting

remains the most robust, performing on par with or better than adaptive configurations without the need for dataset-specific tuning.

We also note that we only enable adaptive-$k$ in the *Dynamic-Ref* pipeline. In the *Static-Ref* setting, we rely on highly optimized decoding frameworks (e.g., vLLM) that typically expose only top-$k$ log probabilities. Entropy estimates derived from truncated top-$k$ distributions are biased relative to the full vocabulary, making stopping decisions based on them unreliable. Because adaptive-$k$ incurs additional tuning costs (introducing two hyperparameters, $k_{\max}$ and $\tau$, rather than one) and does not yield a clear, consistent average improvement, we maintain a fixed value ($k = 4$) as a simple, low-cost default.

### A.13 DETAILS OF IMPLEMENTATION

Our experiments were conducted on high-performance servers equipped with either four or six NVIDIA A800 GPUs (each with 80GB of memory) or eight NVIDIA H100 GPUs (each with 80GB of memory). The A800 machines with four GPUs used the SXM4 version, while those with six GPUs were configured with the PCIe version. All systems were built with Intel(R) Xeon(R) Platinum CPUs, 1 TB of RAM, and a software environment consisting of Python 3.11, PyTorch 2.4, and NCCL 2.21.5 to ensure reproducibility.

### A.14 THE USE OF LARGE LANGUAGE MODELS

We acknowledge the use of a Large Language Model (LLM) to assist with language editing and polishing of this manuscript. The LLM's role was strictly limited to improving grammar, clarity, and phrasing. All scientific ideas, methodologies, results, and conclusions presented herein are the original work of the authors. The authors have thoroughly reviewed all revisions and assume complete responsibility for the entirety of the paper's content.

### A.15 ETHICS STATEMENT

We affirm compliance with the ICLR Code of Ethics. Our work studies label-free test-time adaptation using publicly available benchmarks and does not involve new data collection, human subjects, or personally identifiable information. We follow the licenses of all datasets and base models cited in the paper. Because some tasks touch on finance, medicine, and agriculture, model outputs may carry risk if taken as advice. Our experiments are research-only; the method is not intended for clinical or financial decision-making without qualified human oversight. Deployments should include content filters, disclaimers, and domain-expert review, and must comply with local laws and institutional policies. We report no conflicts of interest or external sponsorship that could bias the results. The computational overhead is small (4–16 extra tokens per query and one forward pass per sample in *Static-Ref*), which limits environmental impact relative to standard fine-tuning.

### A.16 REPRODUCIBILITY STATEMENT

We aim to make the work reproducible. The method is fully specified in Section 4 with pseudocode in Algorithm 1. Datasets, splits, and preprocessing follow Appendix A.2. Training and inference settings, including LoRA configuration, learning schedules, gating, KL weighting, and decoding, are detailed in Appendix A.13; KL details are in Appendix A.7.

Table 33: Detailed results on `DomainBench` and `InstructBench`. ROUGE-Lsum scores (×100 for visibility; higher is better). For each model and dataset, the highest score is **bold** and the second-highest is underlined.

| Model | Method | DomainBench | | | | | InstructBench | | | |
|---|---|---|---|---|---|---|---|---|---|---|
| | | Agriculture | GeoSignal | GenMedGPT | Wealth | Avg. | Dolly | Alpaca-GPT4 | InstructWild | Avg. |
| | Base Model | 8.34 | 22.02 | 14.13 | 21.45 | 16.48 | 30.68 | 34.41 | 25.61 | 30.23 |
| | Tent | 0.98 | 4.59 | 9.32 | 2.33 | 4.30 | 5.66 | 5.72 | 6.41 | 5.93 |
| | EATA | 0.39 | 4.89 | 5.49 | 0.03 | 2.70 | 1.05 | 6.83 | 3.55 | 3.81 |
| | TLM | 14.23 | 27.56 | 24.29 | 26.73 | 23.20 | 24.77 | 37.66 | 27.66 | 30.03 |
| | *Dynamic-Ref* | | | | | | | | | |
| LLAMA-3.2-3B | SYTTA-2 | 18.86 | 29.11 | **24.65** | **29.14** | 25.44 | 29.78 | 40.38 | 33.62 | 34.59 |
| | SYTTA-4 | 19.72 | 26.74 | 17.64 | 29.10 | 23.30 | 32.56 | 40.52 | 34.68 | 35.92 |
| | SYTTA-8 | 18.56 | 29.02 | 20.10 | 28.53 | 24.05 | 32.80 | 40.39 | 34.19 | 35.79 |
| | SYTTA-16 | 18.37 | 27.15 | 18.02 | 28.18 | 22.93 | 30.56 | 39.72 | 32.08 | 34.12 |
| | *Static-Ref* | | | | | | | | | |
| | SYTTA-2 | 15.20 | 27.48 | 18.39 | 28.80 | 22.47 | **35.27** | 39.99 | 32.23 | 35.83 |
| | SYTTA-4 | **20.12** | **29.45** | 17.59 | 29.07 | 24.06 | 34.26 | **40.53** | **36.15** | **36.98** |
| | SYTTA-8 | 16.95 | 28.21 | 21.01 | 28.86 | 23.76 | 34.48 | 39.77 | 35.10 | 36.45 |
| | SYTTA-16 | 15.38 | 28.31 | 19.66 | 28.28 | 22.91 | 33.46 | 39.67 | 32.65 | 35.26 |
| | Base Model | 8.59 | 22.28 | 13.53 | 21.65 | 16.51 | 32.90 | 34.40 | 25.67 | 30.99 |
| | Tent | 1.16 | 3.79 | 0.74 | 13.22 | 4.73 | 0.45 | 4.84 | 9.78 | 5.02 |
| | EATA | 1.52 | 6.43 | 1.86 | 14.60 | 6.10 | 1.75 | 5.89 | 2.53 | 3.39 |
| | TLM | 16.33 | 28.85 | 25.71 | 28.95 | 24.96 | 32.36 | 38.41 | 28.88 | 33.22 |
| | *Dynamic-Ref* | | | | | | | | | |
| LLAMA-3.1-8B | SYTTA-2 | 17.60 | **30.38** | 25.85 | 29.54 | 25.84 | 33.39 | 40.84 | 30.52 | 34.92 |
| | SYTTA-4 | 20.17 | 29.47 | 26.48 | 29.58 | 26.43 | 34.61 | **41.26** | **36.15** | **37.34** |
| | SYTTA-8 | **20.96** | 29.00 | **27.09** | **30.03** | 26.77 | 34.45 | 41.05 | 34.34 | 36.61 |
| | SYTTA-16 | 19.56 | 26.52 | 25.03 | 29.55 | 25.16 | 32.98 | 39.45 | 35.05 | 35.82 |
| | *Static-Ref* | | | | | | | | | |
| | SYTTA-2 | 16.41 | 30.13 | 22.00 | 29.83 | 24.59 | **36.17** | 40.84 | 30.67 | 35.89 |
| | SYTTA-4 | 16.49 | 29.52 | 24.82 | 29.50 | 25.08 | 35.45 | 40.85 | 35.71 | 37.34 |
| | SYTTA-8 | 16.56 | 29.87 | 25.30 | 29.24 | 25.24 | 35.19 | 40.30 | 33.17 | 36.22 |
| | SYTTA-16 | 15.17 | 29.19 | 21.23 | 29.86 | 23.86 | 35.42 | 39.85 | 32.08 | 35.78 |
| | Base Model | 9.43 | 22.03 | 12.51 | 23.88 | 16.96 | 27.05 | 38.17 | 27.77 | 31.00 |
| | Tent | 19.64 | 22.15 | 5.31 | 28.59 | 18.92 | 21.47 | 24.38 | 26.93 | 24.26 |
| | EATA | 16.30 | 21.24 | 12.83 | 22.57 | 18.23 | 30.57 | 27.01 | 23.81 | 27.13 |
| | TLM | 11.23 | 26.21 | 29.67 | 28.13 | 23.81 | 31.05 | 43.08 | 30.76 | 34.96 |
| | *Dynamic-Ref* | | | | | | | | | |
| QWEN-2.5-7B | SYTTA-2 | 16.30 | 29.55 | 28.96 | 29.14 | 25.99 | 36.10 | 43.07 | 31.67 | 36.95 |
| | SYTTA-4 | 20.58 | 29.42 | 29.74 | 29.68 | 27.36 | 36.56 | 43.33 | 32.95 | 37.62 |
| | SYTTA-8 | **21.79** | 29.67 | **31.20** | 29.93 | 28.14 | 36.30 | 43.03 | **34.35** | 37.89 |
| | SYTTA-16 | 21.14 | 28.81 | 26.83 | **30.25** | 26.76 | 35.93 | 43.13 | 33.32 | 37.46 |
| | *Static-Ref* | | | | | | | | | |
| | SYTTA-2 | 14.02 | **29.74** | 29.48 | 29.38 | 25.66 | 35.46 | 43.04 | 31.32 | 36.61 |
| | SYTTA-4 | 19.54 | 29.37 | 29.56 | 29.67 | 27.04 | 36.51 | **43.40** | 34.07 | **37.99** |
| | SYTTA-8 | 21.19 | 29.68 | 29.40 | 29.82 | 27.52 | **36.80** | 43.06 | 33.63 | 37.83 |
| | SYTTA-16 | 18.31 | 29.46 | 25.92 | 29.79 | 25.87 | 36.27 | 43.04 | 33.72 | 37.68 |
| | Base Model | 10.67 | 23.46 | 14.42 | 24.36 | 18.23 | 28.06 | 39.34 | 28.12 | 31.84 |
| | Tent | 4.92 | 27.89 | 14.87 | 28.19 | 18.97 | 29.66 | 28.12 | 11.29 | 23.02 |
| | EATA | 1.88 | 28.23 | 3.17 | 27.97 | 15.31 | 22.99 | 26.08 | 25.33 | 24.80 |
| | TLM | 11.09 | 28.70 | **32.20** | 29.48 | 25.37 | 34.04 | 42.20 | 30.59 | 35.61 |
| | *Dynamic-Ref* | | | | | | | | | |
| QWEN-2.5-14B | SYTTA-2 | 17.32 | 30.52 | 30.45 | 30.11 | 27.10 | 35.79 | **43.37** | 33.68 | 37.61 |
| | SYTTA-4 | 20.09 | 30.57 | 31.05 | 30.14 | 27.96 | 37.04 | 43.24 | 34.45 | 38.24 |
| | SYTTA-8 | **22.01** | **31.31** | 29.70 | **30.60** | 28.40 | 36.51 | 43.30 | **36.56** | **38.79** |
| | SYTTA-16 | 18.82 | 28.72 | 29.79 | 29.94 | 26.82 | 35.29 | 42.86 | 35.49 | 37.88 |
| | *Static-Ref* | | | | | | | | | |
| | SYTTA-2 | 14.24 | 30.57 | 29.76 | 29.27 | 25.96 | 36.88 | 43.17 | 31.58 | 37.21 |
| | SYTTA-4 | 19.52 | 30.45 | 28.91 | 29.53 | 27.10 | 36.97 | 43.13 | 34.12 | 38.07 |
| | SYTTA-8 | 17.33 | 30.91 | 26.19 | 29.86 | 26.07 | 37.14 | 43.25 | 34.52 | 38.30 |
| | SYTTA-16 | 21.85 | 30.93 | 22.26 | 29.57 | 26.15 | **38.17** | 42.90 | 34.46 | 38.51 |

Table 34: Detailed results on `DomainBench` and `InstructBench`. BERTScore-F1 scores ($\times100$ for visibility; higher is better). For each model and dataset, the highest score is **bold** and the second-highest is underlined.

| Model | Method | DomainBench | | | | | InstructBench | | | |
|---|---|---|---|---|---|---|---|---|---|---|
| | | Agriculture | GeoSignal | GenMedGPT | Wealth | Avg. | Dolly | Alpaca-GPT4 | InstructWild | Avg. |
| LLAMA-3.2-3B | Base Model | 66.66 | 67.72 | 66.74 | 67.75 | 67.22 | 71.74 | 72.11 | 70.18 | 71.34 |
| | Tent | 66.62 | 64.49 | 66.78 | 64.43 | 65.58 | 68.04 | 72.82 | 69.75 | 70.20 |
| | EATA | 67.44 | 69.45 | 68.01 | 63.21 | 67.03 | 67.80 | 71.92 | 66.50 | 68.74 |
| | TLM | 66.28 | 70.17 | **70.95** | 69.30 | 69.17 | 69.87 | 74.32 | 70.61 | 71.60 |
| | *Dynamic-Ref* | | | | | | | | | |
| | SYTTA-2 | 69.25 | 70.11 | 70.77 | 70.05 | **70.05** | 72.16 | **74.71** | 72.19 | 73.02 |
| | SYTTA-4 | 69.94 | 69.50 | 69.24 | 70.27 | 69.74 | 73.06 | 74.71 | 72.06 | 73.27 |
| | SYTTA-8 | 68.98 | 70.03 | 70.13 | 70.22 | 69.84 | 72.60 | 74.56 | 71.98 | 73.05 |
| | SYTTA-16 | 69.68 | 69.58 | 69.79 | **70.70** | 69.94 | 71.63 | 74.02 | 71.42 | 72.36 |
| | *Static-Ref* | | | | | | | | | |
| | SYTTA-2 | 66.49 | 70.16 | 69.09 | 70.02 | 68.94 | **74.08** | 74.48 | 71.65 | 73.41 |
| | SYTTA-4 | **70.01** | **70.43** | 68.54 | 70.28 | 69.82 | 73.81 | 74.70 | **72.87** | **73.79** |
| | SYTTA-8 | 68.05 | 69.90 | 69.99 | 70.03 | 69.49 | 73.46 | 74.34 | 72.22 | 73.34 |
| | SYTTA-16 | 66.49 | 69.96 | 69.80 | 69.72 | 68.99 | 73.29 | 73.82 | 71.42 | 72.85 |
| LLAMA-3.1-8B | Base Model | 66.66 | 67.77 | 66.41 | 67.72 | 67.14 | 72.91 | 72.05 | 70.14 | 71.70 |
| | Tent | 67.46 | 69.03 | 67.69 | 67.91 | 68.02 | 67.76 | 67.87 | 69.12 | 68.25 |
| | EATA | 66.28 | 67.34 | 66.46 | 65.45 | 66.38 | 73.83 | 59.81 | 68.98 | 67.54 |
| | TLM | 66.89 | 70.86 | 72.10 | 69.91 | 69.94 | 71.89 | 74.44 | 70.80 | 72.37 |
| | *Dynamic-Ref* | | | | | | | | | |
| | SYTTA-2 | 67.26 | 70.89 | 71.46 | 69.96 | 69.89 | 73.27 | **74.92** | 71.27 | 73.15 |
| | SYTTA-4 | 70.29 | 70.82 | 72.78 | 69.85 | 70.94 | 73.80 | 74.73 | 72.97 | 73.83 |
| | SYTTA-8 | **71.19** | 70.95 | **72.94** | 70.22 | **71.33** | 73.97 | 74.76 | 72.78 | 73.84 |
| | SYTTA-16 | 70.19 | 68.92 | 70.92 | 70.10 | 70.03 | 72.85 | 74.08 | **74.00** | 73.64 |
| | *Static-Ref* | | | | | | | | | |
| | SYTTA-2 | 67.00 | **71.02** | 69.72 | 70.03 | 69.44 | 74.40 | 74.83 | 71.31 | 73.51 |
| | SYTTA-4 | 67.04 | 70.90 | 71.29 | 69.88 | 69.78 | 74.39 | 74.48 | 72.94 | **73.94** |
| | SYTTA-8 | 67.22 | 70.75 | 72.25 | 69.95 | 70.04 | **74.61** | 74.56 | 72.00 | 73.72 |
| | SYTTA-16 | 66.76 | 70.30 | 69.97 | **70.36** | 69.35 | 73.98 | 74.43 | 72.24 | 73.55 |
| QWEN-2.5-7B | Base Model | 65.67 | 67.91 | 65.51 | 68.43 | 66.88 | 70.60 | 73.56 | 70.61 | 71.59 |
| | Tent | 69.06 | 70.40 | 67.00 | 68.87 | 68.83 | 70.54 | 74.42 | 70.78 | 71.91 |
| | EATA | 66.34 | 69.97 | 67.05 | 68.62 | 68.00 | 71.17 | 72.92 | 71.14 | 71.74 |
| | TLM | 64.99 | 70.07 | 74.07 | 70.22 | 69.84 | 73.15 | 75.95 | 71.65 | 73.58 |
| | *Dynamic-Ref* | | | | | | | | | |
| | SYTTA-2 | 67.26 | 71.02 | 73.42 | 70.11 | 70.45 | 74.02 | 75.94 | 71.58 | 73.84 |
| | SYTTA-4 | 69.78 | 71.03 | 73.78 | 70.26 | 71.21 | 75.19 | **76.13** | 71.92 | 74.41 |
| | SYTTA-8 | **71.07** | **71.24** | **74.89** | 70.70 | **71.98** | 74.87 | 76.06 | **72.53** | **74.48** |
| | SYTTA-16 | 70.40 | 70.81 | 72.42 | **70.89** | 71.13 | 74.10 | 76.01 | 72.40 | 74.17 |
| | *Static-Ref* | | | | | | | | | |
| | SYTTA-2 | 66.09 | 71.21 | 73.69 | 70.19 | 70.30 | 74.10 | 75.84 | 71.64 | 73.86 |
| | SYTTA-4 | 69.11 | 71.00 | 73.56 | 70.31 | 70.99 | 74.98 | 75.99 | 72.14 | 74.37 |
| | SYTTA-8 | 70.19 | 71.03 | 73.93 | 70.39 | 71.38 | **75.27** | 75.87 | 72.14 | 74.42 |
| | SYTTA-16 | 69.09 | 70.73 | 72.06 | 70.60 | 70.62 | 74.56 | 76.06 | 72.43 | 74.35 |
| QWEN-2.5-14B | Base Model | 65.21 | 68.28 | 65.98 | 68.33 | 66.95 | 70.63 | 73.87 | 70.91 | 71.80 |
| | Tent | 68.01 | 69.39 | 68.42 | 69.92 | 68.94 | 73.72 | 74.25 | 69.80 | 72.59 |
| | EATA | 64.73 | 70.27 | 68.02 | 69.00 | 68.00 | 73.98 | 74.42 | 70.95 | 73.11 |
| | TLM | 64.81 | 70.83 | **75.38** | 70.46 | 70.37 | 73.34 | 76.13 | 71.58 | 73.68 |
| | *Dynamic-Ref* | | | | | | | | | |
| | SYTTA-2 | 67.47 | 71.36 | 74.32 | 70.15 | 70.83 | 73.78 | 76.26 | 71.91 | 73.98 |
| | SYTTA-4 | 68.92 | 71.56 | 74.70 | 70.63 | 71.45 | 73.96 | 76.09 | 72.06 | 74.04 |
| | SYTTA-8 | 71.05 | **71.97** | 73.99 | **70.70** | **71.93** | 74.28 | 76.15 | **74.49** | **74.98** |
| | SYTTA-16 | 68.86 | 70.69 | 73.91 | 70.56 | 71.00 | 73.47 | 75.98 | 73.42 | 74.29 |
| | *Static-Ref* | | | | | | | | | |
| | SYTTA-2 | 65.40 | 71.14 | 73.93 | 69.80 | 70.07 | 74.46 | 76.26 | 71.54 | 74.09 |
| | SYTTA-4 | 68.38 | 71.33 | 73.71 | 69.99 | 70.85 | 74.22 | 76.09 | 72.04 | 74.12 |
| | SYTTA-8 | 67.12 | 71.20 | 72.38 | 70.54 | 70.31 | 74.68 | **76.32** | 71.98 | 74.32 |
| | SYTTA-16 | **71.37** | 71.72 | 70.87 | 70.23 | 71.05 | **74.93** | 75.91 | 71.96 | 74.27 |

Table 35: Detailed results on `DomainBench` and `InstructBench`. ROUGE-1 scores ($\times 100$ for visibility; higher is better). For each model and dataset, the highest score is **bold** and the second-highest is underlined.

| Model | Method | DomainBench | | | | | InstructBench | | | |
|---|---|---|---|---|---|---|---|---|---|---|
| | | Agriculture | GeoSignal | GenMedGPT | Wealth | Avg. | Dolly | Alpaca-GPT4 | InstructWild | Avg. |
| | Base Model | 9.09 | 24.57 | 15.89 | 23.39 | 18.24 | 34.66 | 37.60 | 27.97 | 33.41 |
| | Tent | 9.02 | 20.41 | 17.71 | 16.96 | 16.03 | 28.73 | 40.81 | 26.56 | 32.04 |
| | EATA | 9.45 | 29.32 | 16.55 | 18.19 | 18.37 | 25.89 | 36.48 | 21.91 | 28.10 |
| | TLM | 16.11 | 30.98 | 26.76 | 29.35 | 25.80 | 28.20 | 40.97 | 30.30 | 33.16 |
| | *Dynamic-Ref* | | | | | | | | | |
| | SYTTA-2 | 21.91 | **32.45** | **27.00** | **32.03** | **28.35** | 33.30 | 44.03 | 35.66 | 37.67 |
| LLAMA-3.2-3B | SYTTA-4 | 22.76 | 29.75 | 18.47 | 31.86 | 25.71 | 36.24 | 44.05 | 38.34 | 39.55 |
| | SYTTA-8 | 21.58 | 32.11 | 22.57 | 31.55 | 26.95 | 37.39 | 43.40 | 36.64 | 39.14 |
| | SYTTA-16 | 21.48 | 23.05 | 20.96 | 30.73 | 24.05 | 27.76 | 40.81 | 35.32 | 34.63 |
| | *Static-Ref* | | | | | | | | | |
| | SYTTA-2 | 17.32 | 30.92 | 21.83 | 31.78 | 25.46 | **38.83** | 43.48 | 35.48 | 39.26 |
| | SYTTA-4 | **23.65** | 32.18 | 17.36 | 31.84 | 26.26 | 37.60 | **44.08** | **39.83** | **40.51** |
| | SYTTA-8 | 19.59 | 31.30 | 23.36 | 31.08 | 26.33 | 38.40 | 42.49 | 38.72 | 39.87 |
| | SYTTA-16 | 17.64 | 29.58 | 20.98 | 30.35 | 24.64 | 37.76 | 39.13 | 35.31 | 37.40 |
| | Base Model | 9.36 | 24.92 | 15.07 | 23.61 | 18.24 | 36.85 | 37.56 | 28.02 | 34.14 |
| | Tent | 11.41 | 28.54 | 17.48 | 25.51 | 20.73 | 27.44 | 28.21 | 27.19 | 27.61 |
| | EATA | 9.66 | 27.71 | 15.97 | 13.89 | 16.81 | 38.52 | 7.77 | 29.88 | 25.39 |
| | TLM | 18.35 | 32.27 | 28.17 | 31.71 | 27.63 | 36.44 | 41.72 | 31.73 | 36.63 |
| | *Dynamic-Ref* | | | | | | | | | |
| | SYTTA-2 | 20.07 | **33.85** | 28.23 | 32.37 | 28.63 | 37.69 | 44.44 | 33.57 | 38.57 |
| LLAMA-3.1-8B | SYTTA-4 | 23.08 | 32.90 | **28.78** | 32.33 | 29.27 | 38.52 | **44.67** | **40.09** | 41.09 |
| | SYTTA-8 | **23.97** | 32.29 | 28.58 | **32.77** | **29.40** | 36.35 | 44.14 | 37.81 | 39.43 |
| | SYTTA-16 | 22.70 | 28.54 | 27.08 | 30.91 | 27.31 | 28.34 | 41.40 | 38.94 | 36.23 |
| | *Static-Ref* | | | | | | | | | |
| | SYTTA-2 | 18.57 | 33.53 | 24.44 | 32.54 | 27.27 | 38.10 | 44.31 | 33.70 | 38.70 |
| | SYTTA-4 | 16.87 | 32.85 | 27.38 | 32.26 | 27.34 | 39.80 | 44.12 | 39.50 | **41.14** |
| | SYTTA-8 | 18.74 | 33.37 | 27.35 | 32.06 | 27.88 | 40.25 | 43.45 | 36.60 | 40.10 |
| | SYTTA-16 | 16.93 | 32.31 | 24.80 | 32.09 | 26.53 | **40.33** | 42.80 | 35.08 | 39.40 |
| | Base Model | 10.18 | 24.48 | 13.61 | 25.98 | 18.56 | 30.10 | 41.52 | 30.15 | 33.92 |
| | Tent | 22.45 | 32.45 | 17.26 | 31.28 | 25.86 | 32.15 | 43.77 | 33.42 | 36.45 |
| | EATA | 16.31 | 31.91 | 16.76 | 29.61 | 23.65 | 32.72 | 40.90 | 35.46 | 36.36 |
| | TLM | 11.96 | 29.98 | **32.76** | 31.72 | 26.60 | 37.43 | 44.60 | 34.65 | 38.89 |
| | *Dynamic-Ref* | | | | | | | | | |
| | SYTTA-2 | 16.98 | 33.17 | 31.41 | 31.76 | 28.33 | 40.76 | **46.65** | 34.73 | 40.71 |
| QWEN-2.5-7B | SYTTA-4 | 23.94 | 33.02 | 31.97 | 32.41 | 30.33 | 40.75 | 45.05 | 36.22 | 40.67 |
| | SYTTA-8 | **25.99** | 33.22 | 31.64 | 32.71 | **30.89** | 41.35 | 46.24 | **37.99** | **41.86** |
| | SYTTA-16 | 24.79 | 32.23 | 29.48 | **33.12** | 29.90 | 40.83 | 45.84 | 36.66 | 41.11 |
| | *Static-Ref* | | | | | | | | | |
| | SYTTA-2 | 14.87 | **33.26** | 32.01 | 31.78 | 27.98 | 40.37 | 46.42 | 34.35 | 40.38 |
| | SYTTA-4 | 22.45 | 32.74 | 32.03 | 32.33 | 29.89 | 41.66 | 45.33 | 37.59 | 41.52 |
| | SYTTA-8 | 24.72 | 33.11 | 31.33 | 32.57 | 30.43 | **42.08** | 45.61 | 37.02 | 41.57 |
| | SYTTA-16 | 20.85 | 32.96 | 28.61 | 32.53 | 28.74 | 40.50 | 46.15 | 37.10 | 41.25 |
| | Base Model | 11.67 | 26.09 | 15.99 | 26.59 | 20.09 | 31.16 | 42.76 | 30.58 | 34.83 |
| | Tent | 16.99 | 31.07 | 20.08 | 30.96 | 24.78 | 40.09 | 42.74 | 32.26 | 38.36 |
| | EATA | 13.41 | 30.59 | 21.03 | 29.93 | 23.74 | 40.26 | 43.81 | 32.09 | 38.72 |
| | TLM | 12.12 | 32.01 | **34.53** | 32.31 | 27.74 | 37.94 | 45.64 | 33.33 | 38.97 |
| | *Dynamic-Ref* | | | | | | | | | |
| | SYTTA-2 | 18.33 | 34.17 | 32.94 | 32.70 | 29.53 | 40.38 | **46.87** | 36.04 | 41.10 |
| QWEN-2.5-14B | SYTTA-4 | 23.26 | 33.38 | 32.91 | 32.43 | **30.50** | 40.87 | 46.69 | 37.98 | 41.85 |
| | SYTTA-8 | **26.00** | **34.80** | 27.35 | **33.09** | 30.31 | 40.15 | 46.28 | **40.65** | **42.36** |
| | SYTTA-16 | 21.43 | 32.07 | 32.16 | 32.89 | 29.64 | 40.26 | 45.17 | 37.94 | 41.13 |
| | *Static-Ref* | | | | | | | | | |
| | SYTTA-2 | 15.91 | 33.67 | 32.27 | 31.29 | 28.28 | 41.68 | 46.73 | 34.60 | 41.00 |
| | SYTTA-4 | 22.61 | 33.91 | 31.41 | 32.42 | 30.09 | 41.55 | 46.63 | 37.45 | 41.88 |
| | SYTTA-8 | 18.41 | 34.30 | 28.55 | 32.75 | 28.50 | 41.00 | 46.20 | 37.88 | 41.69 |
| | SYTTA-16 | 25.57 | 34.45 | 25.38 | 32.47 | 29.47 | **41.81** | 45.75 | 36.87 | 41.47 |

Table 36: Detailed results on `DomainBench` and `InstructBench`. ROUGE-2 scores (×100 for visibility; higher is better). For each model and dataset, the highest score is **bold** and the second-highest is underlined.

| Model | Method | DomainBench | | | | | InstructBench | | | |
|---|---|---|---|---|---|---|---|---|---|---|
| | | Agriculture | GeoSignal | GenMedGPT | Wealth | Avg. | Dolly | Alpaca-GPT4 | InstructWild | Avg. |
| | Base Model | 3.04 | 9.81 | 2.70 | 7.42 | 5.74 | 16.56 | 16.12 | 9.39 | 14.03 |
| | Tent | 3.05 | 8.21 | 2.77 | 5.17 | 4.80 | 13.68 | 18.13 | 9.41 | 13.74 |
| | EATA | 3.11 | 12.96 | 3.18 | 7.21 | 6.61 | 12.08 | 15.60 | 7.61 | 11.76 |
| | TLM | 5.37 | 14.89 | 8.49 | 10.58 | 9.83 | 12.30 | 19.02 | 10.90 | 14.08 |
| | *Dynamic-Ref* | | | | | | | | | |
| | SYTTA-2 | 7.07 | 15.05 | **9.21** | 11.91 | **10.81** | 15.41 | **20.59** | 14.16 | 16.72 |
| LLAMA-3.2-3B | SYTTA-4 | 6.57 | 13.76 | 4.12 | **11.95** | 9.10 | 17.59 | 20.43 | 14.72 | 17.58 |
| | SYTTA-8 | 6.60 | **15.18** | 6.66 | 11.57 | 10.00 | 18.19 | 20.10 | 13.78 | 17.35 |
| | SYTTA-16 | 6.25 | 9.49 | 3.64 | 10.63 | 7.50 | 12.70 | 18.13 | 12.81 | 14.55 |
| | *Static-Ref* | | | | | | | | | |
| | SYTTA-2 | 5.73 | 13.89 | 4.19 | 11.62 | 8.86 | **19.52** | 20.02 | 12.85 | 17.46 |
| | SYTTA-4 | **7.11** | 14.83 | 3.76 | 11.92 | 9.40 | 18.99 | 20.47 | **15.78** | **18.41** |
| | SYTTA-8 | 6.03 | 14.41 | 7.37 | 11.24 | 9.76 | 18.71 | 19.56 | 14.57 | 17.61 |
| | SYTTA-16 | 5.07 | 13.39 | 3.62 | 10.63 | 8.18 | 18.58 | 17.13 | 12.79 | 16.17 |
| | Base Model | 3.32 | 9.97 | 3.28 | 7.48 | 6.01 | 18.40 | 16.24 | 9.28 | 14.64 |
| | Tent | 2.62 | 11.82 | 2.62 | 8.51 | 6.39 | 12.81 | 10.69 | 9.65 | 11.05 |
| | EATA | 3.32 | 12.33 | 2.22 | 2.94 | 5.20 | 18.92 | 2.41 | 10.29 | 10.54 |
| | TLM | 6.43 | 15.17 | 9.61 | 11.89 | 10.78 | 18.21 | 20.02 | 11.43 | 16.56 |
| | *Dynamic-Ref* | | | | | | | | | |
| | SYTTA-2 | 6.41 | **16.47** | 10.99 | 12.27 | 11.54 | 18.81 | **21.14** | 13.93 | 17.96 |
| LLAMA-3.1-8B | SYTTA-4 | 7.05 | 15.82 | 11.14 | 12.17 | 11.54 | 18.92 | 21.00 | 14.83 | 18.25 |
| | SYTTA-8 | **7.26** | 15.31 | **13.22** | **12.48** | **12.07** | 17.24 | 20.57 | 14.26 | 17.36 |
| | SYTTA-16 | 7.26 | 11.82 | 9.36 | 10.75 | 9.80 | 13.44 | 19.11 | 14.98 | 15.84 |
| | *Static-Ref* | | | | | | | | | |
| | SYTTA-2 | 6.08 | 15.64 | 7.33 | 12.11 | 10.29 | 18.99 | 20.60 | 11.94 | 17.18 |
| | SYTTA-4 | 5.73 | 16.05 | 9.03 | 11.99 | 10.70 | **20.73** | 20.72 | **15.44** | **18.96** |
| | SYTTA-8 | 6.09 | 15.43 | 10.92 | 11.91 | 11.09 | 19.87 | 19.92 | 13.60 | 17.79 |
| | SYTTA-16 | 5.65 | 14.61 | 7.23 | 11.71 | 9.80 | 20.71 | 19.52 | 12.96 | 17.73 |
| | Base Model | 3.63 | 9.50 | 3.50 | 8.67 | 6.33 | 14.33 | 18.64 | 10.11 | 14.36 |
| | Tent | 7.32 | 15.15 | 4.24 | 11.96 | 9.67 | 16.17 | 21.51 | 12.13 | 16.60 |
| | EATA | 5.32 | 15.09 | 4.10 | 11.60 | 9.03 | 16.50 | 19.68 | 13.21 | 16.46 |
| | TLM | 4.20 | 13.23 | **14.85** | 11.79 | 11.02 | 19.27 | 22.34 | 12.60 | 18.07 |
| | *Dynamic-Ref* | | | | | | | | | |
| | SYTTA-2 | 6.07 | 15.85 | 13.67 | 12.04 | 11.91 | 21.93 | **23.12** | 12.78 | 19.28 |
| QWEN-2.5-7B | SYTTA-4 | 7.71 | 15.83 | 14.10 | 12.64 | 12.57 | 21.55 | 22.95 | 13.40 | 19.30 |
| | SYTTA-8 | **8.57** | 15.85 | 14.18 | **13.21** | **12.95** | 22.29 | 23.00 | **14.86** | **20.05** |
| | SYTTA-16 | 8.05 | 15.28 | 11.38 | 13.02 | 11.93 | 21.61 | 22.19 | 14.00 | 19.27 |
| | *Static-Ref* | | | | | | | | | |
| | SYTTA-2 | 5.31 | **15.87** | 14.27 | 11.83 | 11.82 | 21.33 | 22.89 | 12.54 | 18.92 |
| | SYTTA-4 | 7.29 | 15.54 | 14.01 | 12.21 | 12.26 | 22.53 | 22.79 | 14.29 | 19.87 |
| | SYTTA-8 | 8.10 | 15.62 | 13.61 | 12.79 | 12.53 | **22.67** | 22.54 | 14.03 | 19.75 |
| | SYTTA-16 | 6.86 | 15.84 | 10.86 | 12.54 | 11.53 | 21.83 | 22.21 | 14.15 | 19.39 |
| | Base Model | 4.14 | 10.10 | 3.84 | 8.69 | 6.69 | 14.71 | 19.56 | 10.30 | 14.86 |
| | Tent | 5.64 | 14.21 | 4.42 | 11.20 | 8.87 | 21.38 | 20.17 | 11.03 | 17.53 |
| | EATA | 4.54 | 14.71 | 4.64 | 10.31 | 8.55 | 21.59 | 20.71 | 11.34 | 17.88 |
| | TLM | 4.31 | 14.40 | **16.75** | 12.07 | 11.88 | 19.53 | 23.08 | 12.06 | 18.22 |
| | *Dynamic-Ref* | | | | | | | | | |
| | SYTTA-2 | 6.38 | **16.53** | 14.76 | 12.35 | 12.50 | 21.63 | **23.38** | 13.21 | 19.41 |
| QWEN-2.5-14B | SYTTA-4 | 7.68 | 16.25 | 14.79 | 12.01 | **12.68** | 20.66 | 23.36 | 13.97 | 19.33 |
| | SYTTA-8 | **8.38** | 16.36 | 9.43 | **12.58** | 11.69 | 21.05 | 23.09 | **16.29** | **20.14** |
| | SYTTA-16 | 6.98 | 15.31 | 14.98 | 12.39 | 12.42 | 21.59 | 22.84 | 14.27 | 19.56 |
| | *Static-Ref* | | | | | | | | | |
| | SYTTA-2 | 5.46 | 15.78 | 14.06 | 11.45 | 11.69 | 21.77 | 23.21 | 12.42 | 19.13 |
| | SYTTA-4 | 7.52 | 15.79 | 13.29 | 12.04 | 12.16 | **22.48** | 23.12 | 13.90 | 19.83 |
| | SYTTA-8 | 6.02 | 16.22 | 9.65 | 12.51 | 11.10 | 22.22 | 23.12 | 14.09 | 19.81 |
| | SYTTA-16 | 8.19 | 15.37 | 7.19 | 12.01 | 10.69 | 22.22 | 22.39 | 13.66 | 19.42 |

Table 37: Detailed results on `DomainBench` and `InstructBench`. ROUGE-L scores ($\times 100$ for visibility; higher is better). For each model and dataset, the highest score is **bold** and the second-highest is underlined.

| Model | Method | DomainBench | | | | | InstructBench | | | |
|---|---|---|---|---|---|---|---|---|---|---|
| | | Agriculture | GeoSignal | GenMedGPT | Wealth | Avg. | Dolly | Alpaca-GPT4 | InstructWild | Avg. |
| | Base Model | 6.63 | 17.20 | 10.37 | 15.16 | 12.34 | 26.02 | 24.93 | 16.97 | 22.64 |
| | Tent | 6.64 | 15.18 | 11.26 | 11.10 | 11.04 | 22.64 | 28.63 | 16.81 | 22.69 |
| | EATA | 7.00 | 21.78 | 11.61 | 15.40 | 13.95 | 20.20 | 24.25 | 15.20 | 19.88 |
| | TLM | 12.22 | 23.48 | 19.68 | 20.30 | 18.92 | 20.88 | 29.59 | 19.16 | 23.21 |
| | *Dynamic-Ref* | | | | | | | | | |
| | SYTTA-2 | 16.82 | 24.84 | **19.91** | 22.37 | **20.99** | 24.63 | 31.06 | 24.08 | 26.59 |
| LLAMA-3.2-3B | SYTTA-4 | 17.48 | 22.19 | 13.58 | 22.39 | 18.91 | 26.95 | 31.02 | 25.70 | 27.89 |
| | SYTTA-8 | 16.35 | **24.96** | 16.25 | 22.35 | 19.98 | 28.66 | 30.88 | 23.83 | 27.79 |
| | SYTTA-16 | 16.07 | 17.13 | 13.14 | 21.89 | 17.06 | 21.29 | 28.63 | 22.49 | 24.14 |
| | *Static-Ref* | | | | | | | | | |
| | SYTTA-2 | 13.07 | 23.16 | 13.74 | 22.02 | 18.00 | **30.27** | 30.56 | 22.09 | 27.64 |
| | SYTTA-4 | **18.51** | 24.41 | 13.22 | **22.40** | 19.64 | 29.22 | **31.07** | 26.97 | **29.09** |
| | SYTTA-8 | 14.59 | 23.81 | 16.99 | 21.61 | 19.25 | 29.84 | 30.17 | 24.96 | 28.32 |
| | SYTTA-16 | 13.34 | 22.86 | 13.14 | 20.88 | 17.56 | 29.39 | 27.81 | 22.48 | 26.56 |
| | Base Model | 6.84 | 17.38 | 9.87 | 15.23 | 12.33 | 28.07 | 24.86 | 16.94 | 23.29 |
| | Tent | 10.18 | 21.11 | 11.79 | 16.68 | 14.94 | 21.41 | 18.00 | 18.64 | 19.35 |
| | EATA | 7.01 | 21.35 | 10.91 | 8.73 | 12.00 | 30.31 | 7.55 | 18.72 | 18.86 |
| | TLM | 14.00 | 24.72 | 21.51 | 22.25 | 20.62 | 28.47 | 30.73 | 19.81 | 26.34 |
| | *Dynamic-Ref* | | | | | | | | | |
| | SYTTA-2 | 16.24 | **26.72** | 22.49 | 23.35 | 22.20 | 28.52 | 31.74 | 22.59 | 27.62 |
| LLAMA-3.1-8B | SYTTA-4 | 17.34 | 26.33 | 22.95 | 22.51 | 22.28 | 30.31 | **32.00** | 25.22 | 29.18 |
| | SYTTA-8 | **18.21** | 25.33 | **23.99** | **23.49** | **22.76** | 27.65 | 31.11 | 24.75 | 27.84 |
| | SYTTA-16 | 17.26 | 21.11 | 21.79 | 21.19 | 20.34 | 21.62 | 29.87 | **26.31** | 25.94 |
| | *Static-Ref* | | | | | | | | | |
| | SYTTA-2 | 13.97 | 26.22 | 16.45 | 22.68 | 19.83 | 28.91 | 31.22 | 20.66 | 26.93 |
| | SYTTA-4 | 12.60 | 25.86 | 19.39 | 22.11 | 19.99 | **32.15** | 31.58 | 26.29 | **30.00** |
| | SYTTA-8 | 14.00 | 25.22 | 19.71 | 22.50 | 20.36 | 30.68 | 30.62 | 23.16 | 28.16 |
| | SYTTA-16 | 12.81 | 25.15 | 16.77 | 22.10 | 19.21 | 31.25 | 30.30 | 23.07 | 28.21 |
| | Base Model | 7.31 | 16.79 | 8.56 | 16.73 | 12.35 | 22.09 | 27.73 | 17.83 | 22.55 |
| | Tent | 17.85 | 25.05 | 10.70 | 22.23 | 18.96 | 25.24 | 32.28 | 20.83 | 26.12 |
| | EATA | 14.22 | 24.45 | 10.39 | 21.23 | 17.57 | 25.39 | 30.04 | 22.70 | 26.04 |
| | TLM | 8.63 | 22.23 | **26.20** | 21.66 | 19.68 | 28.60 | 32.76 | 21.37 | 27.58 |
| | *Dynamic-Ref* | | | | | | | | | |
| | SYTTA-2 | 12.68 | 25.66 | 24.89 | 22.10 | 21.33 | 32.01 | **33.78** | 21.64 | 29.14 |
| QWEN-2.5-7B | SYTTA-4 | 18.17 | 25.80 | 25.34 | 23.05 | 23.09 | 32.09 | 33.54 | 22.70 | 29.45 |
| | SYTTA-8 | **19.78** | 25.95 | 25.67 | **23.88** | **23.82** | 33.16 | 33.78 | 24.78 | 30.57 |
| | SYTTA-16 | 18.91 | 25.00 | 21.73 | 23.64 | 22.32 | 32.20 | 33.18 | 24.03 | 29.81 |
| | *Static-Ref* | | | | | | | | | |
| | SYTTA-2 | 11.03 | 25.73 | 25.56 | 21.86 | 21.05 | 31.51 | 33.63 | 21.41 | 28.85 |
| | SYTTA-4 | 17.11 | 25.11 | 25.25 | 22.39 | 22.47 | 33.27 | 33.50 | 24.00 | 30.26 |
| | SYTTA-8 | 19.06 | 25.56 | 25.02 | 23.16 | 23.20 | **33.63** | 33.32 | 23.43 | 30.12 |
| | SYTTA-16 | 15.70 | **26.42** | 20.54 | 23.03 | 21.42 | 32.56 | 32.86 | 23.66 | 29.69 |
| | Base Model | 8.35 | 17.94 | 9.88 | 16.95 | 13.28 | 22.84 | 28.66 | 18.01 | 23.17 |
| | Tent | 12.38 | 24.21 | 12.82 | 21.30 | 17.68 | 32.76 | 30.24 | 19.27 | 27.42 |
| | EATA | 9.67 | 24.83 | 12.65 | 19.69 | 16.71 | 32.53 | 30.63 | 19.36 | 27.50 |
| | TLM | 8.74 | 23.71 | **29.17** | 22.01 | 20.91 | 28.95 | 33.22 | 20.32 | 27.50 |
| | *Dynamic-Ref* | | | | | | | | | |
| | SYTTA-2 | 13.44 | 26.25 | 26.69 | 22.68 | 22.26 | 31.52 | 33.65 | 22.25 | 29.14 |
| QWEN-2.5-14B | SYTTA-4 | 17.52 | 25.75 | 27.06 | 22.13 | **23.12** | 31.41 | **33.73** | 23.77 | 29.64 |
| | SYTTA-8 | **19.78** | 27.00 | 19.73 | 23.00 | 22.39 | 31.20 | 33.38 | **27.11** | **30.56** |
| | SYTTA-16 | 15.73 | 24.81 | 26.57 | 22.63 | 22.43 | 32.53 | 32.98 | 24.11 | 29.87 |
| | *Static-Ref* | | | | | | | | | |
| | SYTTA-2 | 11.47 | 25.74 | 25.79 | 21.04 | 21.01 | 32.33 | 33.47 | 21.08 | 28.96 |
| | SYTTA-4 | 16.97 | 26.00 | 25.27 | 22.32 | 22.64 | **32.97** | 33.44 | 23.24 | 29.88 |
| | SYTTA-8 | 13.38 | 26.50 | 20.69 | 22.86 | 20.86 | 32.83 | 33.47 | 23.76 | 30.02 |
| | SYTTA-16 | 19.58 | 26.58 | 17.21 | 22.25 | 21.41 | 32.77 | 32.49 | 22.91 | 29.39 |

Table 38: Detailed results on `DomainBench` and `InstructBench`. BLEU scores ($\times 100$ for visibility; higher is better). For each model and dataset, the highest score is **bold** and the second-highest is underlined.

| Model | Method | DomainBench | | | | | InstructBench | | | |
|---|---|---|---|---|---|---|---|---|---|---|
| | | Agriculture | GeoSignal | GenMedGPT | Wealth | Avg. | Dolly | Alpaca-GPT4 | InstructWild | Avg. |
| LLAMA-3.2-3B | Base Model | 1.00 | 4.71 | 1.65 | 3.11 | 2.62 | 7.64 | 8.68 | 3.35 | 6.55 |
| | Tent | 1.04 | 4.24 | 1.59 | 2.15 | 2.26 | 7.10 | 10.55 | 3.66 | 7.10 |
| | EATA | 1.08 | 6.78 | 2.01 | 2.99 | 3.21 | 6.20 | 8.73 | 2.92 | 5.95 |
| | TLM | 2.24 | 7.96 | 5.21 | 5.33 | 5.18 | 6.31 | 10.72 | 4.50 | 7.18 |
| | *Dynamic-Ref* | | | | | | | | | |
| | SYTTA-2 | 3.32 | **8.34** | **5.39** | **6.42** | **5.87** | 8.35 | 12.47 | 7.08 | 9.30 |
| | SYTTA-4 | 3.27 | 7.43 | 2.50 | 6.39 | 4.90 | 9.52 | 12.68 | 7.86 | 10.02 |
| | SYTTA-8 | 3.15 | 8.11 | 2.74 | 6.34 | 5.08 | 9.64 | 12.23 | 7.38 | 9.75 |
| | SYTTA-16 | 3.12 | 4.76 | 2.40 | 5.60 | 3.97 | 6.84 | 10.55 | 5.83 | 7.74 |
| | *Static-Ref* | | | | | | | | | |
| | SYTTA-2 | 2.44 | 7.61 | 2.52 | 6.25 | 4.70 | **10.37** | 11.96 | 6.22 | 9.51 |
| | SYTTA-4 | **3.58** | 8.12 | 2.51 | 6.36 | 5.14 | 10.00 | **12.72** | 8.47 | **10.40** |
| | SYTTA-8 | 2.81 | 7.80 | 3.44 | 6.08 | 5.03 | 10.07 | 11.65 | 7.88 | 9.86 |
| | SYTTA-16 | 2.26 | 7.43 | 2.40 | 5.57 | 4.41 | 9.79 | 10.12 | 5.83 | 8.58 |
| LLAMA-3.1-8B | Base Model | 1.10 | 4.69 | 1.54 | 3.10 | 2.61 | 9.31 | 8.70 | 3.30 | 7.10 |
| | Tent | 1.37 | 6.45 | 2.16 | 3.78 | 3.44 | 7.12 | 5.39 | 4.37 | 5.63 |
| | EATA | 1.06 | 6.20 | 1.71 | 1.12 | 2.52 | 10.04 | 1.12 | 4.10 | 5.09 |
| | TLM | 2.78 | 8.08 | 6.27 | 6.21 | 5.83 | 9.48 | 10.57 | 4.81 | 8.29 |
| | *Dynamic-Ref* | | | | | | | | | |
| | SYTTA-2 | 2.96 | **8.99** | 6.98 | 6.74 | 6.42 | 10.27 | 12.87 | 5.55 | 9.56 |
| | SYTTA-4 | 3.52 | 8.63 | **7.57** | 6.65 | **6.59** | 10.04 | **12.95** | **8.51** | 10.50 |
| | SYTTA-8 | 3.59 | 8.27 | 5.48 | **7.13** | 6.12 | 9.43 | 12.84 | 7.56 | 9.94 |
| | SYTTA-16 | **3.63** | 6.45 | 6.17 | 5.75 | 5.50 | 7.12 | 11.04 | 8.21 | 8.79 |
| | *Static-Ref* | | | | | | | | | |
| | SYTTA-2 | 2.65 | 8.35 | 3.78 | 6.66 | 5.36 | 10.69 | 12.58 | 5.24 | 9.50 |
| | SYTTA-4 | 2.44 | 8.20 | 5.57 | 6.46 | 5.67 | **12.01** | 12.84 | 8.19 | **11.01** |
| | SYTTA-8 | 2.71 | 8.05 | 6.93 | 6.53 | 6.05 | 10.88 | 12.31 | 6.76 | 9.98 |
| | SYTTA-16 | 2.49 | 7.85 | 3.97 | 6.23 | 5.14 | 10.24 | 11.92 | 6.64 | 9.60 |
| QWEN-2.5-7B | Base Model | 1.22 | 4.48 | 1.23 | 3.80 | 2.68 | 6.62 | 10.48 | 3.75 | 6.95 |
| | Tent | 3.54 | 8.01 | 1.79 | 6.17 | 4.88 | 8.57 | 12.92 | 5.49 | 8.99 |
| | EATA | 2.32 | 8.11 | 1.70 | 5.90 | 4.51 | 8.74 | 10.85 | 6.27 | 8.62 |
| | TLM | 1.47 | 7.25 | **9.98** | 6.29 | 6.25 | 10.70 | 13.14 | 6.07 | 9.97 |
| | *Dynamic-Ref* | | | | | | | | | |
| | SYTTA-2 | 2.59 | 8.66 | 9.24 | 6.47 | 6.74 | 11.62 | **14.39** | 6.21 | 10.74 |
| | SYTTA-4 | 3.66 | 8.68 | 9.69 | 6.84 | 7.22 | 12.42 | 13.77 | 6.84 | 11.01 |
| | SYTTA-8 | **4.35** | 8.82 | 9.58 | **7.27** | **7.51** | 12.33 | 14.35 | **7.75** | **11.48** |
| | SYTTA-16 | 3.94 | 8.29 | 7.43 | 7.27 | 6.73 | 12.16 | 13.59 | 7.21 | 10.98 |
| | *Static-Ref* | | | | | | | | | |
| | SYTTA-2 | 2.12 | **8.88** | 9.68 | 6.37 | 6.76 | 11.70 | 14.32 | 6.12 | 10.71 |
| | SYTTA-4 | 3.39 | 8.53 | 9.34 | 6.56 | 6.96 | 12.92 | 13.92 | 7.46 | 11.43 |
| | SYTTA-8 | 4.01 | 8.86 | 9.28 | 6.95 | 7.27 | **13.04** | 13.84 | 7.35 | 11.41 |
| | SYTTA-16 | 3.09 | 8.51 | 6.99 | 6.95 | 6.38 | 12.20 | 14.27 | 7.47 | 11.31 |
| QWEN-2.5-14B | Base Model | 1.46 | 4.99 | 1.49 | 3.92 | 2.97 | 6.77 | 11.37 | 3.99 | 7.38 |
| | Tent | 2.27 | 7.67 | 1.81 | 6.22 | 4.49 | 11.85 | 12.11 | 4.47 | 9.47 |
| | EATA | 1.63 | 7.24 | 1.95 | 5.45 | 4.07 | 11.71 | 13.07 | 5.08 | 9.95 |
| | TLM | 1.46 | 8.10 | **11.10** | 6.44 | 6.77 | 10.63 | 14.12 | 5.50 | 10.08 |
| | *Dynamic-Ref* | | | | | | | | | |
| | SYTTA-2 | 2.58 | 9.41 | 9.96 | 6.71 | 7.16 | 12.07 | **14.95** | 6.55 | 11.19 |
| | SYTTA-4 | 3.40 | 9.39 | 9.86 | 6.51 | **7.29** | 11.53 | 14.89 | 7.26 | 11.23 |
| | SYTTA-8 | 4.00 | 9.04 | 6.07 | **6.96** | 6.52 | 11.87 | 14.63 | **8.78** | **11.76** |
| | SYTTA-16 | 3.12 | 8.25 | 9.58 | 6.79 | 6.94 | 11.71 | 13.86 | 7.48 | 11.02 |
| | *Static-Ref* | | | | | | | | | |
| | SYTTA-2 | 2.12 | 8.71 | 9.37 | 6.03 | 6.56 | **12.55** | 14.81 | 5.82 | 11.06 |
| | SYTTA-4 | 3.50 | 8.79 | 8.91 | 6.54 | 6.94 | 12.28 | 14.83 | 7.08 | 11.40 |
| | SYTTA-8 | 2.41 | 9.22 | 6.03 | 6.93 | 6.15 | 12.18 | 14.34 | 7.33 | 11.28 |
| | SYTTA-16 | **4.24** | **9.62** | 4.23 | 6.46 | 6.14 | 12.13 | 14.05 | 7.11 | 11.10 |

