# OpenReview forum: "You only need 4 extra tokens: Synergistic Test-time Adaptation for LLMs"
_ICLR.cc/2026/Conference — Submitted to ICLR 2026_

### Official Review · Reviewer_9aHm · 2025-10-26

**Soundness:** 3
**Presentation:** 4
**Contribution:** 3
**Rating:** 6
**Confidence:** 4

**Summary:**

This work is in line with test-time adaptation of LLMs. It proposes a novel label-free test-time adaptation framework (SYTTA) for LLMs. It couples two complementary uncertainty signals from different positions: input-side perplexity and output-side predictive entropy. Existing works typically take only one signal into account. A novel strategy, dynamic importance weighting, is applied to incorporate these two signals effectively. The experimental results demonstrate that the proposed method significantly outperforms strong baselines, verifying its effectiveness.

**Strengths:**

- The writing and structure are excellent.
- The proposed framework is novel and interesting, especially the dynamic weighting part.
- Two different signals are well exploited via the dynamic weighting.
- Potential collapse and distribution drift are also considered; to avoid such issues, KL is applied.
- The experiments are comprehensive; a concise case study would further clarify the improvement over baselines.
- Strong and consistent gains over strong baselines across the experiments.

**Weaknesses:**

- From my point of view, no major weakness.
- The discussion of prefix token length could be deepened. More detailed analysis of the generated prefix tokens and other perspectives (e.g., attention sink) would be helpful and interesting.
- It would be beneficial to also compare the computational overhead and latency.

**Questions:**

**Questions and Suggestions**

- I am curious what the exact generated prefix tokens are. I would appreciate a discussion regarding the generated prefix tokens to analyse their contributions from a semantic perspective.
- Does attention sink also affect the prefix token length setting?
- It would be better to have a concise case study section to compare different models straightforwardly.


Lastly, I would be happy to increase the rating if my questions are addressed.

---

> ### Author Response · Authors · 2025-11-26
> **Response to Reviewer 9aHm 1/4**
>
> We thank the reviewer for the positive assessment. We are glad that the reviewer finds the writing/structure excellent, and considers the framework novel and the dynamic weighting effective. In the revised PDF, we have incorporated several changes; all modifications are highlighted in **blue**, and updated table captions are also shown in **blue** for easier inspection.
>
> > **Reviewer:** *I am curious what the exact generated prefix tokens are. I would appreciate a discussion regarding the generated prefix tokens to analyse their contributions from a semantic perspective.*
>
> **Response**: Thank you for this question. We have added a dedicated analysis to the revised PDF (Section 6, Q5: Semantic Effect of the Generated Prefix, and Appendix A.9). You are welcome to refer to these sections for more details and additional case studies. To facilitate easy review of the key points directly in this rebuttal, we summarize the most relevant results here.
>
> **How often are the first tokens changed?** We compare base vs SyTTA outputs on the same DomainBench + InstructBench questions, and define change rate as the fraction of examples whose first $k$ tokens differ (native tokenizer per backbone; ignore leading whitespace markers when comparing).
>
> | Model        | change rate @ $k=4$ (%) | change rate @ $k=16$ (%) |
> | ------------ | ----------------------: | -----------------------: |
> | LLaMA-3.2-3B |                    72.4 |                     96.1 |
> | LLaMA-3.1-8B |                    78.0 |                     96.5 |
> | Qwen-2.5-7B  |                    72.4 |                     97.0 |
> | Qwen-2.5-14B |                    74.7 |                     97.4 |
>
> Even at $k=4$, SyTTA rewrites the beginning for ~70–80% of examples, and by $k=16$, it is greater than 96% for all backbones.
>
> **What do the rewritten prefix tokens do semantically?** On a set of “golden” cases (base answer is unhelpfully short, SyTTA is longer/more informative; $560$ Domain+Instruct examples at $k=4$), we categorize prefix rewrites:
>
> | Rewrite type ($k=4$)                                                              |  #Cases | Share (%) |
> | --------------------------------------------------------------------------------- | ------: | --------: |
> | Content shift (add correct domain/instruction content; remove off-topic openings) |     515 |      92.0 |
> | Persona shift (change speaking role like doctor/advisor)                          |      34 |       6.1 |
> | Added structural anchor (e.g., “Answer:”, “Explanation:”)                         |       7 |       1.3 |
> | Added explicit CoT marker (e.g., “First”, “Step 1”)                               |       4 |       0.7 |
> | **Total**                                                                         | **560** | **100.0** |
>
> This directly addresses the semantic contribution: most rewritten prefix tokens are used to immediately anchor the answer in the correct domain/instruction (92% content shifts), rather than injecting templates or explicit chain-of-thought markers (<2% combined).
>
> **Concrete “exact token” patterns (typical examples).** The most frequent rewritten prefixes (first ~4–6 tokens) look like:
>
> * **Domain/Instruct content anchors:** “In finance, …”, “In clinical practice, …”, “In geology, …”
> * **Persona anchors (rare):** “As your doctor, …”, “As a financial advisor, …”
> * **Template/CoT markers (very rare here):** “Answer: …”, “First, …”
>
> In short, SyTTA-generated prefixes usually change the *semantic framing* of the answer right away by correcting domain/instruction mismatch, rather than adding generic formatting or explicit CoT.

---

> ### Author Response · Authors · 2025-11-26
> **Response to Reviewer 9aHm 2/4**
>
> > **Does attention sink also affect the prefix token length setting?**
>
> **Response:** Yes. We added this analysis in the revised PDF (Appendix A.9), and we also summarize the key numbers here for convenience.
>
> We measure an attention-sink proxy as the average self-attention mass from later positions to the first $k$ tokens (Sink@$k$), using a single teacher-forcing pass and averaging attention probabilities over layers and heads. The results below show two consistent patterns: (1) SyTTA increases Sink@$k$ compared to the base model at both $k=4$ and $k=16$; (2) for both base and SyTTA, Sink@$16$ is higher than Sink@$4$, meaning that increasing $k$ naturally concentrates more attention on early tokens.
>
> | System | Mode     | DomainBench Sink@$4$ | DomainBench Sink@$16$ | InstructBench Sink@$4$ | InstructBench Sink@$16$ |
> | ------ | -------- | -------------------: | --------------------: | ---------------------: | ----------------------: |
> | Base   | –        |                50.86 |                 55.40 |                  52.24 |                   57.74 |
> | SyTTA  | *Static-Ref*    |                56.27 |                 63.56 |                  55.53 |                   67.23 |
> | SyTTA  | *Dynamic-Ref* |                56.14 |                 63.70 |                  57.04 |                   65.00 |
>
> These observations suggest a simple link to the prefix-length choice: short prefixes often already capture most benefits because early tokens receive disproportionately large attention, and increasing $k$ tends to amplify early-token anchoring (higher Sink@$k$) rather than adding equally new information. This helps explain why longer prefixes may give smaller marginal gains on domain/instruction tasks.

---

> ### Author Response · Authors · 2025-11-26
> **Response to Reviewer 9aHm 3/4**
>
> > **It would be better to have a concise case study section to compare different models straightforwardly.**
>
> **Response:** Thank you for the suggestion. We have added a concise case-study section to the revised PDF (Appendix A.8), featuring side-by-side comparisons across four DomainBench datasets, one InstructBench dataset, and one ReasoningBench dataset. To facilitate comparison, all cases utilize the same backbone (Qwen-2.5-7B) and present Base vs. SyTTA under the same prompt, so the differences arise from test-time adaptation rather than changes to the backbone. You are very welcome to read Appendix A.8 for the full outputs; below is a compact snapshot of the key takeaways.
>
> | Dataset (type)                    | What Base does wrong                                                                | What SyTTA changes (main effect)                                                                                                                                                           |
> | --------------------------------- | ----------------------------------------------------------------------------------- | ------------------------------------------------------------------------------------------------------------------------------------------------------------------------------------------ |
> | Agriculture (DomainBench)         | Misreads **CAD** as *Computer-Aided Design* and answers about engineering software  | Anchors to the **agriculture disease** context early; fixes the macro domain mismatch (Software → Agriculture). (It can still miss the exact disease expansion, which we note in the PDF.) |
> | GeoSignal (DomainBench)           | Treats “Getchellite” as non-existing and refuses (“not widely recognized”)          | Anchors to geology and gives a meaningful, domain-grounded answer (recognizes it as a mineral and provides plausible provenance information).                                              |
> | GenMedGPT (DomainBench)           | High-entropy hedging: long list of many diagnoses, weak “doctor” persona            | Reduces uncertainty and gives a concise, more decisive diagnosis-style response aligned with the symptom cluster.                                                                          |
> | Wealth (DomainBench)              | Starts with *Computational Fluid Dynamics* before self-correcting to finance        | Immediately anchors **CFD** to *Contract for Difference* and stays in the financial context.                                                                                               |
> | Dolly (InstructBench; context QA) | “Knowledge leakage”: adds plot details not in the given passage                     | Stays within the provided context and avoids adding unsupported details.                                                                                                                   |
> | MetaMathQA (ReasoningBench)       | Uses an inefficient brute-force path (listing factors) and collapses mid-generation | Switches to a stable strategy (prime factorization/LCM reasoning) and completes a correct solution.                                                                                        |
>
> Across these cases, the recurring pattern is that SyTTA improves the *first-step framing* of generation: it anchors the answer in the correct domain context early (acronyms, long-tail terms, persona), improves faithfulness to the provided context when required, and stabilizes the reasoning strategy on math problems. The full outputs and analysis are in Appendix A.8 of the revised PDF.

---

> ### Author Response · Authors · 2025-11-26
> **Response to Reviewer 9aHm 4/4**
>
> > **It would be beneficial to also compare the computational overhead and latency.**
>
> **Response:** We clarify that the cost has two parts: adaptation (test-time training) and generation (deployment inference). We have also added this clarification to the revised PDF (Algorithm/Complexity paragraph and Table). We include the key table here so that it is easy to review without switching PDFs.
>
> For **adaptation**, a hardware-agnostic way to compare gradient-based TTA methods is to count the number of forward evaluations per sample, since all such methods still require a backward pass per update step, and the main difference is how many forward computations are needed to form the objective. With prefix length $k$ (we denote SyTTA-$k$), our *Dynamic-Ref* variant matches TENT/EATA at $(k+1)|\mathcal{D}|$, while our *Static-Ref* variant uses only **one** forward pass per sample, $|\mathcal{D}|$, by caching the base-model prefix outputs and the log-probabilities used in the KL term (so we avoid repeated decoding in the reference path).
>
>
> | Method | Forward passes (adaptation) |
> |---|---:|
> | TENT / EATA | $(k+1)\cdot\lvert\mathcal{D}\rvert$ |
> | TLM | $2\cdot\lvert\mathcal{D}\rvert$ |
> | SyTTA (*Dynamic-Ref*) | $(k+1)\\cdot\lvert\mathcal{D}\rvert$ |
> | **SyTTA (*Static-Ref*)** | **$\lvert\mathcal{D}\rvert$** |
>
>
> For **generation**, inference is the same as the base model except for attaching a small LoRA module, so the latency overhead is minimal. Additionally, LoRA can be merged into the base weights, making deployment inference effectively identical to that of the base model.
>
> Finally, forward-pass counting is a theoretical analysis. In addition, we measured training-time latency and memory on a single Nvidia H100 for realistic deployment settings. For this experiment we use Qwen-2.5-7B on the Agriculture dataset. Wall-clock time is the full adaptation run, and VRAM usage is measured at the 100th sample.
>
> | Method                    | Time (min:sec) | VRAM (MB) |
> | ------------------------- | -------------: | --------: |
> | TLM                       |           7:31 |     16295 |
> | SyTTA (*Dynamic-Ref*) $k$=4  |          15:29 |     16091 |
> | SyTTA (*Dynamic-Ref*) $k$=16 |          32:23 |     16209 |
> | SyTTA (*Static-Ref*) $k$=4   |           9:57 |     16091 |
> | SyTTA (*Static-Ref*) $k$=16  |          10:03 |     16205 |
>
> From these measurements, SyTTA adds only a small amount of extra training time compared with TLM when using *Static-Ref*: a rough increase for $k=4$ , while VRAM stays almost unchanged. In contrast, *Dynamic-Ref* costs grow roughly in proportion to $k$, which matches our forward-pass complexity analysis. The very similar VRAM numbers across all methods indicate that the main overhead is compute rather than memory, and that SyTTA does not require a larger memory footprint than TLM. Together, these results support *Static-Ref* as the most practical configuration: it keeps memory usage flat, keeps training latency close to TLM, and is relatively insensitive to $k$, while still providing most of the accuracy gains.
>
> > **Lastly, I would be happy to increase the rating if my questions are addressed.**
>
> **Response:** Thank you. We hope that the new Findings (Q5) and the added appendices (A.8/A.9), together with the clarified discussion of overhead and latency, address the questions and make the contributions clearer.

---

### Official Review · Reviewer_FMEh · 2025-10-29

**Soundness:** 2
**Presentation:** 2
**Contribution:** 2
**Rating:** 4
**Confidence:** 5

**Summary:**

The paper proposes SYTTA, a test-time adaptation framework that jointly (i) reduces input-side mismatch via perplexity-driven Input Distribution Adaptation and (ii) sharpens output confidence by minimizing predictive entropy with a reverse-KL constraint to the base model; a dynamic importance weighting balances the two losses.

**Strengths:**

1.	Clear and well-structured writing.
2.	The paper targets test-time learning/adaptation for LLMs, a rapidly emerging direction with clear utility for real-world deployments that face distribution shift and scarce labels.

**Weaknesses:**

1.	**Limited novelty over TLM.** The paper directly reuses input perplexity minimization from prior TLM and mainly adds an output entropy term. Conceptually, this reads as an incremental extension that couples two known self-supervised signals rather than a fundamentally new objective.
2.	**Overstated “Contributions.”** Of the three listed contributions, #2 (“consistent performance gains”) and #3 (“extensive empirical analysis/ablations”) summarize results and diagnostics rather than introduce additional methodological innovations. This weakens the crispness of the claimed contributions.
3.	**Unsubstantiated claim around L191–193.** The statement that “Input Distribution Adaptation reduces input perplexity but does not ensure coherent or confident generation; models may still exhibit high predictive entropy or drift” is plausible but lacks a formal theoretical justification or a targeted empirical test explicitly verifying this claim.
4.	**Benchmark coverage is narrow.** Experiments follow AdaptEval (DomainBench and InstructBench) as in TLM, but do not include ReasoningBench, leaving uncertainty about the method’s behavior on reasoning-heavy evaluations.
5.	**Reproduction gap relative to TLM.** Given that official TLM checkpoints are publicly available, please clarify any cases where reproduced results fall below TLM.
6.	**Computational cost.** Please report latency and memory cost to assess feasibility in realistic serving settings.
7.	**Effectiveness on quantized LLMs.** Are the gains preserved for common INT8/INT4 deployments?
8.	**Bibliography quality issues.** While minor in isolation, the number of bibliographic errors raises concerns about the manuscript’s overall rigor. For example: (i) the author list and venue for Test-time Learning for Large Language Models are incorrect; (ii) the author list for Efficient Test-time Model Adaptation without Forgetting is incorrect.

**Questions:**

see Weaknesses

---

> ### Author Response · Authors · 2025-11-26
> **Response to Reviewer FMEh 1/3**
>
> We thank the reviewer for the careful reading and detailed feedback. We appreciate that the reviewer finds the paper clear and well structured, and agrees that test-time adaptation for LLMs is useful in real deployments. In the revised PDF, we have incorporated several changes; all modifications are highlighted in **blue**, and updated table captions are also shown in **blue** for easier inspection.
>
> > **Limited novelty over TLM. The paper directly reuses input perplexity minimization from prior TLM and mainly adds an output entropy term. Conceptually, this reads as an incremental extension that couples two known self-supervised signals rather than a fundamentally new objective.**
>
> **Response:** We agree that TLM is an important and high-impact baseline, and our IDA component builds on its key insight that prompt perplexity is a strong label-free signal under shift. Our main point is that prompt-side alignment alone is not sufficient: we add an explicit output-side objective (entropy minimization with a reverse-KL “trust region”) to shape the next-token distribution while preventing collapse/drift, and we introduce a *Dynamic Importance Weighting* rule to keep the two signals comparable across instances and steps (fixed weights are unstable in practice due to scale mismatch). Finally, we present two deployment modes (*Static-Ref* / *Dynamic-Ref*), where *Static-Ref* enables single-forward-pass adaptation via caching. Empirically, the added pieces are not cosmetic: the new OCS-only ablation and the full synergy results show that the output-side signal contributes non-trivially beyond TLM, and the weighting/stabilization are needed for robust training.
>
> > **Overstated “Contributions.” Of the three listed contributions, #2 (“consistent performance gains”) and #3 (“extensive empirical analysis/ablations”) summarize results and diagnostics rather than introduce additional methodological innovations. This weakens the crispness of the claimed contributions.**
>
> **Response:** Thank you for pointing this out. We revised the contribution list in the Introduction to be more precise, and we added an extra item based on the new rebuttal-stage analyses. In particular, we now emphasize that beyond performance, our new prefix/entropy analyses help explain *how* test-time adaptation changes model behavior and why short prefixes (e.g., k=4) can be sufficient.
>
> > **Unsubstantiated claim around L191–193. The statement that “Input Distribution Adaptation reduces input perplexity but does not ensure coherent or confident generation; models may still exhibit high predictive entropy or drift” is plausible but lacks a formal theoretical justification or a targeted empirical test explicitly verifying this claim.**
>
> **Response:** Thank you for the suggestion. To directly test whether reducing input perplexity (IDA/TLM) necessarily yields generation with less entropy, we conduct a targeted output-entropy analysis.
>
> Concretely, we use Qwen-2.5-7B and randomly sample 100 prompts per dataset on DomainBench (4 datasets) and InstructBench (3 datasets). For each prompt, we compute the model’s average next-token predictive entropy over a short generated prefix and then average the entropy across the 100 samples. The results below show that SyTTA consistently and substantially reduces output entropy across all datasets.
>
> | Dataset       | Entropy (Base) $\downarrow$ | Entropy (TLM) $\downarrow$ | Entropy (SyTTA) $\downarrow$ |
> |---------------|------------------------------:|------------------------------:|--------------------------------:|
> | Agriculture   | 0.47 | 0.42 | **0.09** |
> | GeoSignal     | 0.40 | 0.71 | **0.14** |
> | GenMedGPT     | 0.40 | 0.59 | **0.17** |
> | Wealth        | 0.47 | 0.82 | **0.15** |
> | Dolly         | 0.30 | 0.66 | **0.10** |
> | Alpaca-GPT4   | 0.40 | 0.66 | **0.23** |
> | InstructWild  | 0.46 | 0.61 | **0.15** |
>
> Overall, this supports our motivation: IDA/TLM can reduce input mismatch, but does not reliably control decoding-time uncertainty, whereas SyTTA’s output-side shaping explicitly lowers predictive entropy in the generated prefix.

---

> ### Author Response · Authors · 2025-11-26
> **Response to Reviewer FMEh 2/3**
>
> > **Benchmark coverage is narrow. Experiments follow AdaptEval (DomainBench and InstructBench) as in TLM, but do not include ReasoningBench, leaving uncertainty about the method’s behavior on reasoning-heavy evaluations.**
>
> **Response:** Thank you for the suggestion. We added **Appendix A.6 (Evaluation on Reasoning Tasks)** with a new ReasoningBench evaluation (from AdaptEval): GSM8K and MetaMathQA (math), and LogiQA (logic). For reasoning, we report Exact Match (EM) instead of ROUGE, by extracting a canonical final answer from both the reference and the model output.
>
> We find that reasoning indeed behaves differently from domain QA: very short prefixes can be less stable because early tokens may not yet determine the final solution pattern. Therefore we sweep prefix length in the efficient *Static-Ref* setting with $k\in\{1,4,16,32,64,128\}$ (full table in Appendix A.6). The summary below reports the best SyTTA (over $k$) compared with the base model and a strong TTA baseline (TLM), averaged over the three reasoning datasets.
>
> | Backbone | Base Avg. EM | TLM Avg. EM | SyTTA (*Static-Ref*) best Avg. EM |
> |---|---:|---:|---:|
> | LLaMA-3.2-3B | 65.46 | **73.71** | 73.62 ($k$=1) |
> | LLaMA-3.1-8B | 71.93 | **76.13** | 75.96 ($k$=1) |
> | Qwen-2.5-7B | 66.15 | 67.34 | **73.21** ($k$=128) |
> | Qwen-2.5-14B | 68.15 | 72.22 | **76.73** ($k$=128) |
>
> Overall, SyTTA improves EM substantially on the Qwen backbones (especially on MetaMathQA), showing that the framework transfers beyond domain QA and instruction following. On the LLaMA backbones, SyTTA is competitive but slightly below TLM on average, and very long prefixes can be less stable for the smaller LLaMA-3.2-3B, consistent with the reviewer’s concern that reasoning regimes can differ.
>
> We also include a small mode comparison on reasoning (Appendix A.6): for Qwen-2.5-7B at k=64, *Static-Ref* is better than *Dynamic-Ref* and is faster in practice due to avoiding repeated decoding in the reference path.
>
> | Qwen-2.5-7B @ k=64 | GSM8K | MetaMathQA | LogiQA |
> |---|---:|---:|---:|
> | *Static-Ref* | 85.74 | 67.68 | 61.64 |
> | *Dynamic-Ref* | 84.78 | 59.28 | 59.36 |
>
> > **Reproduction gap relative to TLM. Given that official TLM checkpoints are publicly available, please clarify any cases where reproduced results fall below TLM.**
>
> **Response:** Thank you for raising this. We respect TLM and used it as our primary reference point. In our reproduction, we observed that (i) even the base model scores can differ from those reported in TLM due to differences in inference frameworks, GPU models, etc., and (ii) when we follow a clean reimplementation aligned with the paper description, our reproduced TLM results are generally reasonable and often competitive. Importantly, on backbones such as Qwen-2.5-7B, we provide a direct comparison of the Base→TLM improvement magnitudes between our reproduction and the official paper:
>
> | Dataset      | Base (Ours) | TLM (Ours) | Δ (Ours) | %Δ (Ours) | Base (Official) | TLM (Official) | Δ (Official) | %Δ (Official) |
> | ------------ | ----------: | ---------: | -------: | --------: | --------------: | -------------: | -----------: | ------------: |
> | Agriculture  |        9.43 |      11.23 |     1.80 |     19.1% |            9.81 |          16.52 |         6.71 |         68.4% |
> | GeoSignal    |       22.03 |      26.21 |     4.18 |     19.0% |           26.49 |          30.81 |         4.32 |         16.3% |
> | GenMedGPT    |       12.51 |      29.67 |    17.16 |    137.2% |           13.13 |          23.94 |        10.81 |         82.3% |
> | Wealth       |       23.88 |      28.13 |     4.25 |     17.8% |           27.39 |          33.11 |         5.72 |         20.9% |
> | Dolly        |       27.05 |      31.05 |     4.00 |     14.8% |           31.21 |          31.77 |         0.56 |          1.8% |
> | Alpaca-GPT4  |       38.17 |      43.08 |     4.91 |     12.9% |           44.39 |          46.08 |         1.69 |          3.8% |
> | InstructWild |       27.77 |      30.76 |     2.99 |     10.8% |           28.66 |          34.82 |         6.16 |         21.5% |
> | **Average**  |   **22.98** |  **28.59** | **5.61** | **33.1%** |       **25.87** |      **31.01** |     **5.14** |     **30.7%** |
>
> We observe that our implementation achieves an average relative improvement of **%Δ = 33.1%**, which is slightly higher than the official paper’s **30.7%**, suggesting that we are not intentionally under-reporting the baseline. We will further clarify our reproduction setup in the revised PDF.

---

> ### Author Response · Authors · 2025-11-26
> **Response to Reviewer FMEh 3/3**
>
> > **Computational cost. Please report latency and memory cost to assess feasibility in realistic serving settings.**
>
> **Response:** We clarify that the serving-time cost comes from **generation**. After adaptation, inference is the same as the base model except that we attach a small LoRA module (rank 8 on q_proj and v_proj). This adds only a small number of extra parameters and a negligible increase in activations, so the memory cost and latency remain very close to the base model. In practical deployments, the LoRA weights can also be merged into the base weights, making inference effectively identical to the base model in both latency and memory usage.
>
> In addition, we measured **training-time** latency and memory on a single Nvidia H100. For this experiment we use Qwen-2.5-7B on the Agriculture dataset. Wall-clock time is the full adaptation run, and VRAM usage is measured at the 100th sample.
>
> | Method                    | Time (min:sec) | VRAM (MB) |
> | ------------------------- | -------------: | --------: |
> | TLM                       |           7:31 |     16295 |
> | SyTTA (*Dynamic-Ref*) $k$=4  |          15:29 |     16091 |
> | SyTTA (*Dynamic-Ref*) $k$=16 |          32:23 |     16209 |
> | SyTTA (*Static-Ref*) $k$=4   |           9:57 |     16091 |
> | SyTTA (*Static-Ref*) $k$=16  |          10:03 |     16205 |
>
> From these measurements, SyTTA adds only a small amount of extra training time compared with TLM when using *Static-Ref*: a rough increase for $k=4$ , while VRAM stays almost unchanged. In contrast, *Dynamic-Ref* costs grow roughly in proportion to $k$, which matches our forward-pass complexity analysis. The very similar VRAM numbers across all methods indicate that the main overhead is compute rather than memory, and that SyTTA does not require a larger memory footprint than TLM. Together, these results support *Static-Ref* as the most practical configuration: it keeps memory usage flat, keeps training latency close to TLM, and is relatively insensitive to $k$, while still providing most of the accuracy gains.
>
> > **Effectiveness on quantized LLMs. Are the gains preserved for common INT8/INT4 deployments?**
>
> **Response:** We ran a focused quantization study on Qwen-2.5-7B with INT4 AWQ (see revised PDF, Appendix A.7). Overall, SyTTA under INT4 is less stable than in our full-precision setting: some configurations still improve over the base model, but others degrade noticeably. In particular, we observed failure modes such as degenerate repetition (for example, repeatedly restating the question), suggesting that our test-time updates can be sensitive to aggressive quantization noise.
>
> Below are the INT4 AWQ results for Qwen-2.5-7B (ROUGE-Lsum × 100):
>
> | Method | Agriculture | GeoSignal | GenMedGPT | Wealth | Domain Avg. | Dolly | Alpaca-GPT4 | InstructWild | Instruct Avg. |
> |---|---:|---:|---:|---:|---:|---:|---:|---:|---:|
> | Base | 9.70 | 22.17 | 12.77 | 23.85 | 17.13 | 28.06 | 37.82 | 27.65 | 31.17 |
> | TLM | 13.55 | **29.10** | 27.57 | **29.31** | **24.88** | **32.80** | 39.12 | _32.99_ | **34.97** |
> | SyTTA (*Dynamic-Ref*) $k=4$ | 16.65 | 21.56 | **28.61** | 19.68 | 21.62 | 30.15 | **41.88** | 21.58 | 31.20 |
> | SyTTA (*Dynamic-Ref*) $k=16$ | 5.21 | 20.00 | 20.55 | 4.64 | 12.60 | 28.03 | 40.25 | 17.41 | 28.56 |
> | SyTTA (*Static-Ref*) $k=4$ | _18.67_ | 23.96 | _28.17_ | _24.39_ | _23.80_ | _30.45_ | _41.21_ | 21.56 | 31.07 |
> | SyTTA (*Static-Ref*) $k=16$ | **18.88** | _24.79_ | 16.96 | 17.14 | 19.45 | 29.05 | 38.55 | **34.60** | _34.07_ |
>
> These results suggest that SyTTA’s gains are not reliably preserved under aggressive INT4 quantization: some settings still help (often *Static-Ref* with smaller $k$), but others can collapse (notably *Dynamic-Ref* with longer prefixes). Improving quantization-tolerance (INT8/INT4) for test-time adaptation is an important direction for future work.
>
>
> > **Bibliography quality issues. While minor in isolation, the number of bibliographic errors raises concerns about the manuscript’s overall rigor. For example: (i) the author list and venue for Test-time Learning for Large Language Models are incorrect; (ii) the author list for Efficient Test-time Model Adaptation without Forgetting is incorrect.**
>
> **Response:** Thank you for the careful checks. We have thoroughly reviewed and corrected the bibliography to ensure author lists and venues are accurate. We apologize for these mistakes and appreciate the feedback that helped us improve the manuscript’s rigor.

---

### Official Review · Reviewer_iemY · 2025-10-31

**Soundness:** 2
**Presentation:** 2
**Contribution:** 3
**Rating:** 4
**Confidence:** 2

**Summary:**

The paper proposes a technique for test-time LLM adaptation (SyTTA) that relies on minimizing the input prompt perplexity on the one hand, and learning a 4-16 token output prefix on the other hand. The two objectives are weighted dynamically.

**Strengths:**

- Consistent gains across multiple model families and datasets are demonstrated
- Clear writing style

**Weaknesses:**

Since the two driving forces here seem to be input distribution adaptation and output confidence shaping, the two obvious ablations are doing only one of them. While the first one (input distribution adaptation) is provided in form of the TLM baseline, an ablation with isolated output confidence shaping is missing.

Having ROUGE-Lsum as the single metric in the main paper is limiting. Table 4 in the appendix does contain BERTScore results, but the gains are less conclusive.

I don't have a background in test time adaptation, so this may be my ignorance. But I have a hard time buying the motivation behind the output confidence shaping. According to the authors, it "introduce an output-oriented objective that regularizes the next-token distribution" (L194). But IIUC the algorithm is actually the opposite of traditional regularization as it makes the model accumulate more probability mass on its own prediction.

Figs. 3-5 are too tiny, especially Fig. 5. I feel that this has become more common recently, but imo it should be a reason for desk-rejection if there is absolutely no chance of reading it on a print-out.

Minor comments:
The Hu et al. (2025) citation is for ICLR, but it should be ICML.

**Questions:**

- The approach seems quite expensive in terms of test-time compute - can you say more about the compoutational complexity?

---

> ### Author Response · Authors · 2025-11-26
> **Response to Reviewer iemY 1/3**
>
> We thank the reviewer for the careful reading and constructive feedback. We are glad that the reviewer finds our writing clear and our gains consistent across multiple model families and datasets. In the revised PDF, we have incorporated several changes; all modifications are highlighted in **blue**, and updated table captions are also shown in **blue** for easier inspection.
>
> > **(Weakness) Since the two driving forces here seem to be input distribution adaptation and output confidence shaping, the two obvious ablations are doing only one of them. While the first one (input distribution adaptation) is provided in form of the TLM baseline, an ablation with isolated output confidence shaping is missing.**
>
> **Response:** We agree, and we added an **OCS-Only** ablation to the main results table (Table 2) for every model and dataset. **OCS-Only** applies only our output confidence shaping loss $\mathcal{L}_{\text{OCS}}$, without input distribution adaptation (IDA).
>
> Across models and datasets, **OCS-Only** often improves over the base model, but it is not consistently strong by itself and can be comparable to (or worse than) TLM on some settings. In contrast, the full SyTTA (joint **IDA + OCS** with our dynamic weighting) is consistently better, supporting that the strongest gains come from combining the two components rather than using confidence shaping alone.
>
> | Backbone | DomainBench Avg. (Base / TLM / OCS-Only / SyTTA) | InstructBench Avg. (Base / TLM / OCS-Only / SyTTA) |
> |---|---:|---:|
> | LLaMA-3.2-3B | 16.48 / 23.20 / 19.28 / **24.06** | 30.23 / 30.03 / 30.88 / **36.93** |
> | LLaMA-3.1-8B | 16.51 / 24.96 / 20.04 / **26.43** | 30.99 / 33.22 / 29.03 / **37.34** |
> | Qwen-2.5-7B | 16.96 / 23.81 / 23.55 / **27.00** | 31.00 / 34.96 / 35.74 / **37.99** |
> | Qwen-2.5-14B | 18.23 / 25.37 / 23.76 / **27.96** | 31.84 / 35.61 / 33.87 / **38.24** |
>
> These results show that confidence shaping alone is helpful but not enough: IDA and OCS address different parts of test-time shift, and their combination yields the most reliable improvements.
>
>
> > **(Weakness) Having ROUGE-Lsum as the single metric in the main paper is limiting. Table 4 in the appendix does contain BERTScore results, but the gains are less conclusive.**
>
> **Response:** We agree that relying on a single metric is limiting. The earlier BERTScore table was less conclusive mainly because our first submission had a mismatch in the BERTScore evaluation setup between SyTTA and some baselines. We have **recomputed and updated Appendix Table 34** with a consistent BERTScore setting across all methods, and the updated results show that SyTTA achieves state-of-the-art performance on the vast majority of model–dataset pairs. We also added extra analysis and case studies (Appendix A.5, A.8, A.9) to show that SyTTA is not simple format imitation, but a real adaptation that changes content and reasoning behavior, which are as follows: We first include results for Qwen-2.5-7B (one backbone) due to rebuttal space; the full set of backbones is in the revised PDF (Appendix A.5).
>
> **Audit results (Qwen-2.5-7B; 50 samples per dataset).** Each cell is `Base factuality / SyTTA factuality`. (Faithfulness follows similar trends; see Appendix A.5 for full details.)
>
> **DomainBench**
>
> | Dataset | *Dynamic-Ref* $k=4$ | *Dynamic-Ref* $k=16$ | *Static-Ref* $k=4$ | *Static-Ref* $k=16$ |
> |---|---|---|---|---|
> | Agriculture | 0.56 / 0.80 | 0.76 / 0.86 | 0.82 / 0.76 | 0.64 / 0.80 |
> | GenMedGPT | 0.68 / 0.46 | 0.78 / 0.60 | 0.80 / 0.50 | 0.86 / 0.70 |
> | GeoSignal | 0.52 / 0.74 | 0.68 / 0.72 | 0.52 / 0.70 | 0.56 / 0.64 |
> | Wealth | 0.76 / 0.96 | 0.92 / 0.98 | 0.90 / 0.98 | 0.74 / 0.94 |
>
> **InstructBench**
>
> | Dataset | *Dynamic-Ref* $k=4$ | *Dynamic-Ref* $k=16$ | *Static-Ref* $k=4$ | *Static-Ref* $k=16$ |
> |---|---|---|---|---|
> | Alpaca-GPT4 | 0.96 / 0.94 | 0.76 / 0.92 | 0.74 / 0.90 | 0.72 / 0.92 |
> | Dolly | 0.56 / 0.70 | 0.70 / 0.74 | 0.62 / 0.82 | 0.80 / 0.76 |
> | InstructWild | 0.78 / 0.96 | 0.84 / 0.96 | 0.74 / 1.00 | 0.78 / 1.00 |
>
> **ReasoningBench**
>
> | Dataset | *Dynamic-Ref* $k=4$ | *Dynamic-Ref* $k=16$ | *Static-Ref* $k=4$ | *Static-Ref* $k=16$ |
> |---|---|---|---|---|
> | GSM8K | 0.88 / 0.92 | 0.84 / 0.90 | 0.82 / 0.88 | 0.82 / 0.90 |
> | LogiQA | 0.58 / 0.60 | 0.68 / 0.56 | 0.64 / 0.62 | 0.66 / 0.66 |
> | MetaMath | 0.84 / 0.86 | 0.88 / 0.90 | 0.82 / 0.88 | 0.82 / 0.88 |
>
> **Takeaways:** Across most domain/instruction/reasoning datasets, SyTTA improves or maintains factuality compared to the base model, suggesting that the ROUGE gains are not mainly stylistic. We also observe similar trends on faithfulness (agreement with the benchmark reference); please see Appendix A.5 for the full faithfulness tables and judge prompt.

---

> ### Author Response · Authors · 2025-11-26
> **Response to Reviewer iemY 2/3**
>
> Below is a compact snapshot of the key takeaways of the case studies:
>
> | Dataset (type)                    | What Base does wrong                                                                | What SyTTA changes (main effect)                                                                                                                                                           |
> | --------------------------------- | ----------------------------------------------------------------------------------- | ------------------------------------------------------------------------------------------------------------------------------------------------------------------------------------------ |
> | Agriculture (DomainBench)         | Misreads **CAD** as *Computer-Aided Design* and answers about engineering software  | Anchors to the **agriculture disease** context early; fixes the macro domain mismatch (Software → Agriculture). (It can still miss the exact disease expansion, which we note in the PDF.) |
> | GeoSignal (DomainBench)           | Treats “Getchellite” as non-existing and refuses (“not widely recognized”)          | Anchors to geology and gives a meaningful, domain-grounded answer (recognizes it as a mineral and provides plausible provenance information).                                              |
> | GenMedGPT (DomainBench)           | High-entropy hedging: long list of many diagnoses, weak “doctor” persona            | Reduces uncertainty and gives a concise, more decisive diagnosis-style response aligned with the symptom cluster.                                                                          |
> | Wealth (DomainBench)              | Starts with *Computational Fluid Dynamics* before self-correcting to finance        | Immediately anchors **CFD** to *Contract for Difference* and stays in the financial context.                                                                                               |
> | Dolly (InstructBench; context QA) | “Knowledge leakage”: adds plot details not in the given passage                     | Stays within the provided context and avoids adding unsupported details.                                                                                                                   |
> | MetaMathQA (ReasoningBench)       | Uses an inefficient brute-force path (listing factors) and collapses mid-generation | Switches to a stable strategy (prime factorization/LCM reasoning) and completes a correct solution.                                                                                        |
>
> Across these cases, the recurring pattern is that SyTTA improves the *first-step framing* of generation: it anchors the answer in the correct domain context early (acronyms, long-tail terms, persona), improves faithfulness to the provided context when required, and stabilizes the reasoning strategy on math problems. The full outputs and analysis are in Appendix A.8 of the revised PDF.
>
> > **(Weakness) I don't have a background in test time adaptation, so this may be my ignorance. But I have a hard time buying the motivation behind the output confidence shaping. According to the authors, it "introduce an output-oriented objective that regularizes the next-token distribution" (L194). But IIUC the algorithm is actually the opposite of traditional regularization as it makes the model accumulate more probability mass on its own prediction.**
>
> **Response:** Thank you for pointing this out. Here, “output-oriented objective” means we reduce the entropy of the next-token distribution along a short prefix, making the distribution slightly sharper (higher confidence) than base. We agree that our original wording could be confusing if “regularization” is understood in terms like L2 regularization. In our method, confidence sharpening is paired with a "reverse-KL-to-base" term that keeps the adapted distribution anchored to the base model, preventing drift/collapse. We have revised the corresponding text for clarity in the main text.

---

> ### Author Response · Authors · 2025-11-26
> **Response to Reviewer iemY 3/3**
>
> > **(Weakness) Figs. 3-5 are too tiny, especially Fig. 5. I feel that this has become more common recently, but imo it should be a reason for desk-rejection if there is absolutely no chance of reading it on a print-out.**
>
> **Response:** We apologize for this oversight. We have increased the font size in **Figures 3–5** and enlarged **Figure 5** to improve readability when printed.
>
> > **(Minor) The Hu et al. (2025) citation is for ICLR, but it should be ICML.**
>
> **Response:** Thank you for catching this. We have corrected the venue to **ICML** and have checked all references to ensure that such mistakes are corrected. We apologize for the mistake.
>
> > **(Question) The approach seems quite expensive in terms of test-time compute - can you say more about the compoutational complexity?**
>
> **Response:** We clarify that the compute has two parts: adaptation (test-time training) and generation (deployment inference).
>
> | Method | Forward passes (adaptation) |
> |---|---:|
> | TENT / EATA | $(k+1)\cdot\lvert\mathcal{D}\rvert$ |
> | TLM | $2\cdot\lvert\mathcal{D}\rvert$ |
> | SyTTA (*Dynamic-Ref*) | $(k+1)\cdot\lvert\mathcal{D}\rvert$ |
> | **SyTTA (*Static-Ref*)** | **$\lvert\mathcal{D}\rvert$** |
>
>
> For adaptation, we report the dominant cost in **Table 1** as the number of forward passes per sample. This is the main difference across methods, because all compared gradient-based baselines still require one backward pass for each update step, so the key gap is how many forward evaluations are needed to form the test-time objective. In particular, our *Dynamic-Ref* costs $(k+1)|D|$ (with small $k\in[4,16]$), which is the same as baselines like TENT and EATA, while our *Static-Ref* uses **one** forward pass per sample ($|D|$), which is much fewer than baselines. This is achieved by caching the base-model prefix outputs and log-probabilities for the KL term, which avoids repeated decoding and further reduces adaptation overhead in practice.
>
> For generations, inference is the same as the base model, except that we attach a small LoRA module. This adds a very small overhead, and we can also merge the LoRA weights into the base weights, making the inference identical to that of the base model in practice.

---

### Official Review · Reviewer_rHcJ · 2025-11-01

**Soundness:** 3
**Presentation:** 3
**Contribution:** 3
**Rating:** 8
**Confidence:** 3

**Summary:**

This paper proposes SYTTA (Synergistic Test‑Time Adaptation), a label‑free test‑time adaptation framework for LLMs that couples

(i) input‑side perplexity minimization (IDA) with
(ii) output‑side confidence shaping via entropy minimization plus a reverse‑KL “trust region” (OCS).

A lightweight Dynamic Importance Weighting scheme balances the two signals on the fly.
Two deployment modes are offered: Static‑Ref (compute/caches a short base‑model prefix once) and Dynamic‑Ref (updates while generating the prefix).
Experiments on LLaMA‑3.1/3.2 and Qwen‑2.5 (7B/14B) show consistent ROUGE‑Lsum improvements, often with k = 4‑token prefixes and LoRA updates to q_proj and v_proj.

**Strengths:**

- Static‑Ref’s single forward pass per sample (during adaptation) is a practical and well‑motivated design. The cohort‑level, transductive “question‑only” setting matches common multi‑tenant deployments and makes the compute constraints explicit
- Principled way to sharpen predictions without collapse and to anchor updates near the base model. The mode‑seeking property of reverse‑KL makes it a reasonable “trust‑region” surrogate for generation.
- Analysis provides actionable choices and relates gains to model post‑training intensity (Qwen vs LLaMA), which will help practitioners.

**Weaknesses:**

- ROUGE/BERTScore/BLEU are surface overlap metrics. For domain QA they correlate imperfectly with factual correctness and safety. The paper lacks human or verifier‑based factuality/faithfulness evaluation (even sampled) and error analysis to ensure gains aren’t mainly stylistic (e.g., matching domain phrasing) rather than correct content.
- Does not report out‑of‑cohort generalization (e.g., adapt on batch A, test on disjoint batch B from the same target distribution), nor sensitivity to cohort size or streaming/batch arrival patterns
- All tasks are text‑generation QA/instruction following. Prior TTA results suggest entropy signals behave differently in code, math, or structured reasoning tasks with verifiers

**Questions:**

- Use paired bootstrap for ROUGE/BLEU and report 95% CIs.
- Since domains include health/finance, show a toxicity/hallucination sanity check (or cite one) to confirm KL+entropy do not over‑sharpen into confident but wrong outputs.
- Have you tried adaptive k (e.g., stop once average entropy drops below a threshold)? This may retain the benefits of k=4 while saving tokens when the signal is even earlier.
- Why do you restrict the lora to just q_proj and v_proj? Does more modules not yield gains?
- Can you add small‑scale human evaluation or automatic verifier checks (where feasible) to ensure that ROUGE improvements reflect correct content, not just stylistic alignment? Even random 100‑sample audits on Agriculture/Wealth would be informative.

---

> ### Author Response · Authors · 2025-11-26
> **Response to Reviewer rHcJ 1/5**
>
> We thank the reviewer for the thorough review and encouraging feedback. We are glad that the reviewer finds SyTTA practical (especially *Static-Ref*), the reverse-KL stabilization well motivated, and the analysis actionable. In the revised PDF, we have incorporated several changes; all modifications are highlighted in **blue**, and updated table captions are also shown in **blue** for easier inspection.
>
> > **ROUGE/BERTScore/BLEU are surface overlap metrics. For domain QA they correlate imperfectly with factual correctness and safety. The paper lacks human or verifier-based factuality/faithfulness evaluation (even sampled) and error analysis to ensure gains aren’t mainly stylistic (e.g., matching domain phrasing) rather than correct content.**
> >
> > **Since domains include health/finance, show a toxicity/hallucination sanity check (or cite one) to confirm KL+entropy do not over-sharpen into confident but wrong outputs.**
> >
> > **Can you add small-scale human evaluation or automatic verifier checks (where feasible) to ensure that ROUGE improvements reflect correct content, not just stylistic alignment? Even random 100-sample audits on Agriculture/Wealth would be informative.**
>
> **Response:** We agree that ROUGE/BLEU/BERTScore are surface-overlap metrics and can miss factuality and safety issues, especially in health/finance. To address this directly, we added **Appendix A.5: External Factuality, Faithfulness, and Safety Audit** in the revised PDF. We employ an LLM-as-a-judge protocol (judge: `DeepSeek-V3.2-Exp`) to audit random samples across datasets and backbones, ensuring that the evaluation is not based on token overlap. For each backbone, compute mode (*Dynamic-Ref* / *Static-Ref*), and prefix length $k\in\{4,16\}$, we randomly sample 50 examples per dataset and evaluate Base vs. SyTTA. The judge sees *(question, benchmark reference, model answer)* and outputs three binary labels: **Factuality** (mostly correct by expert knowledge), **Faithfulness** (mostly consistent with the benchmark reference), and **Safety** (no clearly harmful/hateful/illegal/dangerous content). For space, we report (i) the factuality rate for Base and SyTTA, and (ii) the number of unsafe outputs for SyTTA; faithfulness follows similar trends and is included in the revised PDF.
>
> Below, we include results for Qwen-2.5-7B (one backbone) due to space constraints; the full set of backbones is presented in the revised PDF (Appendix A.5).
>
> **Audit results (Qwen-2.5-7B; 50 samples per dataset).** Each cell is `Base factuality / SyTTA factuality / #Unsafe(SyTTA)`.
>
> **DomainBench**
>
> | Dataset | *Dynamic-Ref* $k=4$ | *Dynamic-Ref* $k=16$ | *Static-Ref* $k=4$ | *Static-Ref* $k=16$ |
> |---|---|---|---|---|
> | Agriculture | 0.56 / 0.80 / 0 | 0.76 / 0.86 / 0 | 0.82 / 0.76 / 0 | 0.64 / 0.80 / 0 |
> | GenMedGPT | 0.68 / 0.46 / 0 | 0.78 / 0.60 / 0 | 0.80 / 0.50 / 0 | 0.86 / 0.70 / 0 |
> | GeoSignal | 0.52 / 0.74 / 0 | 0.68 / 0.72 / 0 | 0.52 / 0.70 / 0 | 0.56 / 0.64 / 0 |
> | Wealth | 0.76 / 0.96 / 0 | 0.92 / 0.98 / 0 | 0.90 / 0.98 / 0 | 0.74 / 0.94 / 0 |
>
> **InstructBench**
>
> | Dataset | *Dynamic-Ref* $k=4$ | *Dynamic-Ref* $k=16$ | *Static-Ref* $k=4$ | *Static-Ref* $k=16$ |
> |---|---|---|---|---|
> | Alpaca-GPT4 | 0.96 / 0.94 / 0 | 0.76 / 0.92 / 1 | 0.74 / 0.90 / 1 | 0.72 / 0.92 / 0 |
> | Dolly | 0.56 / 0.70 / 0 | 0.70 / 0.74 / 0 | 0.62 / 0.82 / 0 | 0.80 / 0.76 / 0 |
> | InstructWild | 0.78 / 0.96 / 0 | 0.84 / 0.96 / 0 | 0.74 / 1.00 / 0 | 0.78 / 1.00 / 0 |
>
> **ReasoningBench**
>
> | Dataset | *Dynamic-Ref* $k=4$ | *Dynamic-Ref* $k=16$ | *Static-Ref* $k=4$ | *Static-Ref* $k=16$ |
> |---|---|---|---|---|
> | GSM8K | 0.88 / 0.92 / 0 | 0.84 / 0.90 / 0 | 0.82 / 0.88 / 0 | 0.82 / 0.90 / 0 |
> | LogiQA | 0.58 / 0.60 / 0 | 0.68 / 0.56 / 0 | 0.64 / 0.62 / 0 | 0.66 / 0.66 / 0 |
> | MetaMath | 0.84 / 0.86 / 0 | 0.88 / 0.90 / 0 | 0.82 / 0.88 / 0 | 0.82 / 0.88 / 0 |
>
> **Takeaways:** On most domain/instruction/reasoning datasets, SyTTA improves or maintains factuality, and unsafe outputs remain very low (including on Wealth and GenMedGPT, where unsafe is 0 in this audit). This sanity check suggests that KL+entropy shaping is not mainly producing stylistic domain phrasing while drifting into unsafe content. To further address the “stylistic alignment only” concern, we also expanded **Appendix A.8** with case studies showing substantive behavior changes (domain anchoring, reduced context leakage, and strategy correction), not just phrasing changes, with side-by-side comparisons across four DomainBench datasets, one InstructBench dataset, and one ReasoningBench dataset. To facilitate comparison, all cases utilize the same backbone (Qwen-2.5-7B) and present Base vs. SyTTA under the same prompt, so the differences arise from test-time adaptation rather than changes to the backbone. You are very welcome to read Appendix A.8 for the full outputs; below is a compact snapshot of the key takeaways.

---

> ### Author Response · Authors · 2025-11-26
> **Response to Reviewer rHcJ 2/5**
>
> | Dataset (type)                    | What Base does wrong                                                                | What SyTTA changes (main effect)                                                                                                                                                           |
> | --------------------------------- | ----------------------------------------------------------------------------------- | ------------------------------------------------------------------------------------------------------------------------------------------------------------------------------------------ |
> | Agriculture (DomainBench)         | Misreads **CAD** as *Computer-Aided Design* and answers about engineering software  | Anchors to the **agriculture disease** context early; fixes the macro domain mismatch (Software → Agriculture). (It can still miss the exact disease expansion, which we note in the PDF.) |
> | GeoSignal (DomainBench)           | Treats “Getchellite” as non-existing and refuses (“not widely recognized”)          | Anchors to geology and gives a meaningful, domain-grounded answer (recognizes it as a mineral and provides plausible provenance information).                                              |
> | GenMedGPT (DomainBench)           | High-entropy hedging: long list of many diagnoses, weak “doctor” persona            | Reduces uncertainty and gives a concise, more decisive diagnosis-style response aligned with the symptom cluster.                                                                          |
> | Wealth (DomainBench)              | Starts with *Computational Fluid Dynamics* before self-correcting to finance        | Immediately anchors **CFD** to *Contract for Difference* and stays in the financial context.                                                                                               |
> | Dolly (InstructBench; context QA) | “Knowledge leakage”: adds plot details not in the given passage                     | Stays within the provided context and avoids adding unsupported details.                                                                                                                   |
> | MetaMathQA (ReasoningBench)       | Uses an inefficient brute-force path (listing factors) and collapses mid-generation | Switches to a stable strategy (prime factorization/LCM reasoning) and completes a correct solution.                                                                                        |
>
> Across these cases, the recurring pattern is that SyTTA improves the *first-step framing* of generation: it anchors the answer in the correct domain context early (acronyms, long-tail terms, persona), improves faithfulness to the provided context when required, and stabilizes the reasoning strategy on math problems. The full outputs and analysis are in Appendix A.8 of the revised PDF.

---

> ### Author Response · Authors · 2025-11-26
> **Response to Reviewer rHcJ 3/5**
>
> > **All tasks are text-generation QA/instruction following. Prior TTA results suggest entropy signals behave differently in code, math, or structured reasoning tasks with verifiers.**
>
> **Response:** Thank you for raising this point. In the revised PDF, we added **Appendix A.6 (Evaluation on Reasoning Tasks)** with a new ReasoningBench evaluation (from AdaptEval): GSM8K and MetaMathQA (math), and LogiQA (logic). For reasoning, we report Exact Match (EM) instead of ROUGE, by extracting a canonical final answer from both the reference and the model output.
>
> We find that reasoning indeed behaves differently from domain QA: very short prefixes can be less stable because early tokens may not yet determine the final solution pattern. Therefore, we sweep prefix length in the efficient *Static-Ref* setting with $k\in\{1,4,16,32,64,128\}$ (full table in Appendix A.6). The summary below reports the best SyTTA (over $k$) compared with the base model and a strong TTA baseline (TLM), averaged over the three reasoning datasets.
>
> | Backbone | Base Avg. EM | TLM Avg. EM | SyTTA (*Static-Ref*) best Avg. EM |
> |---|---:|---:|---:|
> | LLaMA-3.2-3B | 65.46 | **73.71** | 73.62 ($k$=1) |
> | LLaMA-3.1-8B | 71.93 | **76.13** | 75.96 ($k$=1) |
> | Qwen-2.5-7B | 66.15 | 67.34 | **73.21** ($k$=128) |
> | Qwen-2.5-14B | 68.15 | 72.22 | **76.73** ($k$=128) |
>
> Overall, SyTTA improves EM substantially on the Qwen backbones (especially on MetaMathQA), showing that the framework transfers beyond domain QA and instruction following. On the LLaMA backbones, SyTTA is competitive but slightly below TLM on average, and very long prefixes can be less stable for the smaller LLaMA-3.2-3B, consistent with the reviewer’s concern that reasoning regimes can differ.
>
> We also include a small mode comparison on reasoning (Appendix A.6): for Qwen-2.5-7B at k=64, *Static-Ref* outperforms *Dynamic-Ref* and is faster in practice due to avoiding repeated decoding in the reference path.
>
> | Qwen-2.5-7B @ k=64 | GSM8K | MetaMathQA | LogiQA |
> |---|---:|---:|---:|
> | *Static-Ref* | 85.74 | 67.68 | 61.64 |
> | *Dynamic-Ref* | 84.78 | 59.28 | 59.36 |
>
> > **Use paired bootstrap for ROUGE/BLEU and report 95% CIs.**
>
> **Response:** Thank you for the suggestion. In the revised PDF, we add **Appendix A.4: Paired Bootstrap Analysis of ROUGE-Lsum**, where we compute 95% confidence intervals for the ROUGE-Lsum gains using paired bootstrap with 5,000 resamples for each model–dataset–mode–$k$ configuration. In each resample, we draw examples with replacement, compute the mean difference $\Delta = \text{ROUGE-Lsum}(\text{SyTTA}) - \text{ROUGE-Lsum}(\text{Base})$, and then summarize the empirical distribution of $\Delta$.
>
> For example, on Qwen-2.5-7B (*Dynamic-Ref*, $k=4$) we obtain:
>
> * **Agriculture:** $\Delta = 11.15$, 95% CI [10.84, 11.47]
> * **GenMedGPT:** $\Delta = 17.24$, 95% CI [16.91, 17.55]
> * **Wealth:** $\Delta = 5.80$, 95% CI [5.44, 6.19]
>
> On the same backbone in *Static-Ref* mode, all DomainBench and InstructBench datasets also have strictly positive $\Delta$ with CIs that do not cross zero. More broadly, for both Qwen-2.5 models (7B and 14B) and for both modes, every configuration in Tables A.4 (a–d) shows $\Delta > 0$ in all 5,000 bootstrap samples, so the 95% CIs are strictly above zero.
>
> For the LLaMA models, the pattern is similar: almost all configurations have positive and significant gains. The main edge case is Dolly, $k=16$, *Dynamic-Ref*, where SyTTA is statistically tied with the base model:
>
> * LLaMA-3.2-3B: $\Delta = -0.12$, CI [-0.43, 0.18]
> * LLaMA-3.1-8B: $\Delta = 0.08$, CI [-0.24, 0.40]
>
> Since these intervals include zero, we treat this particular setting as neutral; all other *Dynamic-Ref* $k=16$ settings and all *Static-Ref* settings have positive and significant $\Delta$.
>
> Due to space constraints, we apply the paired bootstrap to **ROUGE-Lsum**, our primary text-level metric, and provide full tables for all backbones, modes, and prefix lengths in **Appendix A.4**.

---

> ### Author Response · Authors · 2025-11-26
> **Response to Reviewer rHcJ 4/5**
>
> > **Does not report out-of-cohort generalization (e.g., adapt on batch A, test on disjoint batch B from the same target distribution), nor sensitivity to cohort size or streaming/batch arrival patterns**
>
> **Response:** Thank you for this suggestion. We have added an out-of-cohort study in **Appendix A.10** (“Out-of-cohort Generalization”). For Qwen-2.5-7B, we select 5,000 examples per dataset from DomainBench and InstructBench, and split them into two disjoint halves: the first 2,500 examples are used solely for test-time adaptation, and the last 2,500 examples are used solely for evaluation. For ReasoningBench, we follow the same protocol and report Exact Match (EM).
>
> | Benchmark (out-of-cohort) | Metric | Base | Best SyTTA | Gain |
> |---|---:|---:|---:|---:|
> | DomainBench (4 ds avg.) | ROUGE-Lsum ($\times 100$) | 17.00 | 26.98 | **+9.98** |
> | InstructBench (3 ds avg.) | ROUGE-Lsum ($\times 100$) | 31.03 | 37.59 | **+6.56** |
> | ReasoningBench (3 ds avg.) | EM | 66.36 | 67.92 | **+1.56** |
>
> The new tables in Appendix A.10 show that SyTTA remains effective under this stricter out-of-cohort setting: on DomainBench, SyTTA improves the average ROUGE-Lsum by 9.98 points over the base model; on InstructBench, the average gain is 6.56; and on ReasoningBench, SyTTA improves the average EM by 1.56. These gains are evident across all domains and reasoning tasks, indicating that the improvements are not limited to the within-cohort reuse of the same questions.
>
> > **Why do you restrict the lora to just q\_proj and v\_proj? Does more modules not yield gains?**
>
> **Response:** We follow common practice in LLM fine-tuning work and restrict LoRA to the attention projections, specifically q_proj and v_proj, because these layers have a significant impact on how information flows across tokens and are known to be effective and stable adaptation targets. In our pilot experiments, we also attempted to enable LoRA on more modules (attention + FFN), but observed almost no average improvement and, in some datasets, a clear degradation, while the cost increased substantially. For example, on Qwen-2.5-7B with LoRA rank 8 defaultly, switching from “LoRA on Attn(Q,V) only” to “LoRA on all attention and FFN modules” increases the LoRA trainable parameters (and corresponding branch compute) by about 8×. Given the lack of consistent gains and test-time efficiency constraints, we chose the more focused and lightweight configuration for q_proj and v_proj only.

---

> ### Author Response · Authors · 2025-12-01
> **Response to Reviewer rHcJ 5/5**
>
> > **Have you tried adaptive k (e.g., stop once average entropy drops below a threshold)? This may retain the benefits of k=4 while saving tokens when the signal is even earlier.**
>
> **Response:** Thank you for the suggestion. We did not include an adaptive-$k$ variant in the current submission. We agree that it is a natural extension, but setting a fixed entropy threshold is not straightforward because the entropy scale and its decay pattern vary across datasets (and also vary with prompt format and backbone), making a single universal threshold brittle.
>
> We further implemented adaptive-$k$ for the *Dynamic-Ref* pipeline and ran a sweep on Qwen-2.5-7B. The stopping rule has two parameters: $\tau$ (running-average token entropy threshold) and $k_{\max}$ (hard cap on per-sample generated tokens). Table below reports ROUGE-Lsum (×100) on DomainBench and InstructBench for fixed ($k$) and for several ($k_{\max},\tau$) choices.
>
>
> **Per-dataset results (ROUGE-Lsum ×100)**
>
> | Dataset|$k=4$ | $k=16$ | AK $k_{max}=16$, $\tau$=0.025 | AK $k_{max}=16$, $\tau$=0.1 | AK $k_{max}=16$, $\tau$=0.4 | AK $k_{max}=16$, $\tau$=1.6 | AK $k_{max}=64$, $\tau$=0.1 | AK $k_{max}=64$, $\tau$=0.4 | AK $k_{max}=64$, $\tau$=1.6 |
> | ----------------------- | --------: | --------: | ------------------: | ----------------: | ----------------: | ----------------: | ----------------: | ----------------: | ----------------: |
> | Agriculture|20.58 |21.14 |  21.35 |21.56 |17.70 |12.60 |18.40 |17.51 |12.62 |
> | GeoSignal  |29.42 |28.81 |  29.09 |29.27 |28.84 |27.00 |29.26 |28.82 |26.02 |
> | GenMedGPT|29.74 |26.83 |  28.18 |28.07 |30.21 |29.94 |28.31 |30.51 |29.91 |
> | Wealth|29.69 |30.25 |  30.32 |30.33 |29.69 |28.87 |29.95 |29.63 |28.21 |
> | **DomainBench — avg**   | **27.36** | **26.76** |**27.24** |  **27.31** |  **26.61** |  **24.60** |  **26.48** |  **26.62** |  **24.19** |
> | Dolly |36.56 |35.93 |  35.85 |36.52 |37.23 |35.32 |36.46 |36.26 |36.07 |
> | Alpaca-GPT4|43.33 |43.13 |  43.18 |43.23 |43.33 |43.27 |43.12 |43.18 |43.17 |
> | InstructionWild |32.95 |33.32 |  33.93 |33.51 |32.15 |30.73 |33.65 |32.08 |30.87 |
> | **InstructBench — avg** | **37.62** | **37.46** |**37.65** |  **37.75** |  **37.57** |  **36.44** |  **37.74** |  **37.18** |  **36.70** |
> | **All 7 — avg**  | **31.75** | **31.34** |**31.70** |  **31.78** |  **31.31** |  **29.67** |  **31.31** |  **31.14** |  **29.55** |
>
> ---
>
> In short, adaptive-$k$ sometimes matches or improves over fixed settings on individual datasets, but its behavior is dataset dependent and sensitive to both $\tau$ and $k_{\max}$; averaged over DomainBench and InstructBench, the gains are small and not a consistent win over fixed ($k=4$). We also note that we only enable adaptive-$k$ in *Dynamic-Ref*. In *Static-Ref*, we rely on highly optimized decoding frameworks (e.g., vLLM) that expose only top-$k$ log probabilities; their entropy estimates are biased relative to full-vocab log probabilities, and stopping decisions based on those biased entropies can be unreliable.
>
> Because adaptive-$k$ incurs additional tuning costs (two hyperparameters rather than one) and does not yield a clear, consistent average improvement, we maintain a fixed value ($k=4$) as a simple, low-cost default; we have included these details in **Appendix A.12**.

---

### Meta-Review · Area_Chair_x3zM · 2026-01-09

**Summary:**

This paper worked on label‑free test‑time adaptation, and presents a framework SyTTA which use a tailored  Dynamic Importance Weighting rule to balance  two uncertainty signals including input-side perplexity and output-side predictive entropy arise under distribution shift. While the merits of the paper (e.g., addresses a relevant and timely problem; good performance, clear and well-structured writing) appreciated by reviewers, there are several key weaknesses that concern the reviewers (and myself). Although the authors addressed some of these concerns during the rebuttal, major weaknesses remain.

**Reviewer Concerns:**

This is a borderline case, after a quick examination of the paper and a careful review of all rebuttals, I found that I share the same concerns raised by the reviewers, as summarized below:

- Reviewer rHcJ raises concern about *“the  factuality/faithfulness evaluation”*. While the authors partially addressed this issue by adding an LLM-as-a-judge audit in the rebuttal, human evaluation would be more convincing.
- Reviewer iemY raises concerns:
    1. *“ROUGE-Lsum as the single metric, gains are less conclusive”*. I am skeptical about the authors’ rebuttal response:*“The earlier BERTScore table was less conclusive mainly because our first submission had a mismatch in the BERTScore evaluation setup between SyTTA and some baselines.”*
    2. “*the motivation behind the output confidence shaping*”. Although the authors attempted to address this concern, I find that they did not sufficiently clarify or provide evidence explaining why or how output confidence shaping is helpful. This concern also echoes reviewer FMEh’s comment regarding *“unsubstantiated claims”.*
- Reviewer FMEh raises concern about *“theoretical justification/explicitly verification of the proposed method*” While the authors added experiments in the rebuttal showing that SyTTA reduces entropy, they did not explain/analyze why or how entropy reduction leads to performance improvements. For example, TLM appears to have relatively high entropy (sometimes even higher than the base model), but still achieves significantly better performance (66.88 vs. 69.84), and is comparable to the proposed method (69.84 vs. 71.98).
- Finally, reviewers express consensus concern about the narrow benchmark coverage. Although the authors added additional experiments (e.g., ReasoningBench) in the rebuttal and revised version, the results on reasoning-heavy benchmarks do not seem to support the authors’ claim that the proposed method that *only needs four extra tokens for test-time adaptation*. For instance, on the LLaMA series, the performance is worse than TLM, and on the Qwen series, 128 tokens are required.

I encourage the author to address the issues listed above, which would make the work more impactful and further solid.

**Reviewer Scores:**

After carefully reviewing the reviews and the rebuttal, it is possible that Reviewer 9aHm may increase their rating from 6 to 8. However, I do not believe this change should be considered a decisive factor for accepting the paper. The most substantive concerns stem from Reviewers FMEh and iemY. After reading the detailed rebuttal, I remain unconvinced that the authors’ responses adequately address these core issues, and to change their original scores.

---

### Decision · Program_Chairs · 2026-01-26

Reject